# Aberrant gene activation in synovial sarcoma relies on SSX specificity and increased PRC1.1 stability

Nezha S. Benabdallah[1], Vineet Dalal[1], R. Wilder Scott [2], Fady Marcous[1], Afroditi Sotiriou[1], Felix K. F. Kommoss [1,3], Anastasija Pejkovska[1], Ludmila Gaspar[1], Lena Wagner[1], Francisco J. Sánchez-Rivera [4], Monica Ta[5], Shelby Thornton[5], Torsten O. Nielsen [5], T. Michael Underhill [2] & Ana Banito [1]✉

The SS18-SSX fusion drives oncogenic transformation in synovial sarcoma by bridging SS18, a member of the mSWI/SNF (BAF) complex, to Polycomb repressive complex 1 (PRC1) target genes. Here we show that the ability of SS18-SSX to occupy H2AK119ub1-rich regions is an intrinsic property of its SSX C terminus, which can be exploited by fusion to transcriptional regulators beyond SS18. Accordingly, SS18-SSX recruitment occurs in a manner that is independent of the core components and catalytic activity of BAF. Alternative SSX fusions are also recruited to H2AK119ub1-rich chromatin and reproduce the expression signatures of SS18-SSX by engaging with transcriptional activators. Variant Polycomb repressive complex 1.1 (PRC1.1) acts as the main depositor of H2AK119ub1 and is therefore required for SS18-SSX occupancy. Importantly, the SSX C terminus not only depends on H2AK119ub1 for localization, but also further increases it by promoting PRC1.1 complex stability. Consequently, high H2AK119ub1 levels are a feature of murine and human synovial sarcomas. These results uncover a critical role for SSX-C in mediating gene deregulation in synovial sarcoma by providing specificity to chromatin and further enabling oncofusion binding by enhancing PRC1.1 stability and H2AK119ub1 deposition.

Sarcomas are a group of cancers arising in soft tissues or bone that disproportionately affect children and young adults. Like other pediatric cancers, many types of sarcoma display a low mutational burden and are driven by dominant fusion oncoproteins involving chromatin-associated regulators and transcription factors[1].

Synovial sarcoma, one of the more common soft tissue tumors in young patients, is characterized by the in-frame fusion of the mammalian switch/sucrose nonfermentable (mSWI/SNF or BAF) chromatin remodeling complex subunit SS18 to an SSX family member, whereby the last eight amino acids of SS18 are replaced by the C-terminal 78 amino

[1]Soft Tissue Sarcoma Research Group, Hopp Children's Cancer Center, Heidelberg (KiTZ), German Cancer Research Center (DKFZ), Heidelberg, Germany. [2]Department of Cellular and Physiological Sciences, Faculty of Medicine, University of British Columbia, Vancouver, BC, Canada. [3]Institute of Pathology, University of Heidelberg, Heidelberg, Germany. [4]Cancer Biology and Genetics Program, Memorial Sloan Kettering Cancer Center, Sloan Kettering Institute, New York, NY, USA. [5]Department of Pathology and Laboratory Medicine, Vancouver Coastal Health Research Institute and Faculty of Medicine, University of British Columbia, Vancouver, BC, Canada. ✉e-mail: a.banito@kitz-heidelberg.de

acids of SSX1, SSX2 or, rarely, SSX4 (refs. [2],[3]). Biochemical and proteomic studies have shown that SS18-SSX integration into BAF evicts the tumor suppressor subunit SMARCB1 (also known as BAF47 or hSNF5) from the complex via competition of SSX with SMARCB1 for binding to the nucleosome acidic patch[4–6]. This led to the view that alteration of BAF composition is a crucial step in synovial sarcoma tumorigenesis[4].

We previously showed that SS18-SSX1 co-occupies noncanonical PRC1.1 target sites[7], and recent work has demonstrated that SSX displays a strong affinity for H2AK119ub1-modified nucleosomes, a mark deposited by PRC1 (ref. [6]). The Polycomb repressive system plays a crucial role in regulating gene expression in all eukaryotes. It consists of two protein complexes: PRC1 and PRC2. PRC1 is composed of a catalytic core made up of RING finger proteins 1 and 2 (RING1A and RNF2/RING1B) plus one of six Polycomb group RING finger (PCGF) proteins, which monoubiquitylate histone H2A at lysine 119 (H2AK119ub1). Noncanonical PRC1 complexes containing PCGF1/3/5/6 are responsible for the majority of H2AK119ub1 deposition and gene repression[8–10]. Polycomb domains are formed by the subsequent recruitment of PRC2, which monomethylates, dimethylates and trimethylates histone H3 at lysine 27 (H3K27me1, H3K27me2 and H3K27me3, respectively), further recruiting canonical PRC1 complexes containing PCGF2 or PCGF4 (refs. [11–14]). The co-occupancy of Polycomb group proteins at specific chromatin sites results in the repression of key developmental genes[15]. In synovial sarcoma, H2AK119ub1-modified nucleosomes provide an interface for SS18-SSX, resulting in the rewiring of an altered BAF complex to Polycomb targets, leading to their aberrant activation and resulting in the oncogenic gene expression signatures characteristic of synovial sarcoma[6,7,16,17].

While disruption of normal BAF complex function is central in synovial sarcoma, studies in mice have shown that SMARCB1 loss is not required for SS18-SSX-driven tumorigenesis, generating instead tumors with epithelioid sarcoma features[16,18]. Moreover, the recent discovery of SSX fusions in synovial sarcoma with alternative activators, such as EWSR1 and MN1 (ref. [19]), also raises questions about the requirement for direct BAF complex deregulation for all synovial sarcomas, prompting further investigation into the characteristics of the SSX tail.

Here, we demonstrate that the SSX C terminus is responsible for the presence of SS18-SSX at its specific targets via an interaction with H2AK119ub1 independently of SS18 and BAF. We show that the new SSX fusions, EWSR1-SSX1 and MN1-SSX1, share the same transcriptional signature as SS18-SSX1 and that their presence at H2AK119ub1-rich regions depends solely on SSX. While BAF complexes are critical in SS18-SSX-driven synovial sarcomas, we show that EWSR1 and MN1 activate gene expression via a mechanism that can be independent of BAF presence. Therefore, a more general view of synovial sarcoma emerges in which SSX-C serves as an anchor for recruitment and mislocalization of transcriptional activators to H2AK119ub1-rich chromatin domains. Furthermore, we uncover a feedback loop in which the SSX-C binds to and enhances H2AK119ub1 by stabilizing PRC1.1 complex presence on

chromatin. This results in acquisition of high H2AK119ub1 levels during synovial sarcoma tumorigenesis, enabling further oncofusion binding and potentiating its oncogenic activity.

## Results

### PRC1.1 controls global H2AK119ub1 and SS18-SSX occupancy

We previously showed that SS18-SSX1 co-occupies KDM2B/PRC1.1 target sites and that lysine demethylase 2B (KDM2B) suppression disrupts SS18-SSX chromatin occupancy, triggering proliferative arrest and a fibroblast-like morphology[7]. However, the chromatin environments bound by SS18-SSX/PRC1.1 that are rich in H2AK119ub1 are also co-occupied by other chromatin regulators. Several variant PRC1 complexes can deposit H2AK119ub1, which is recognized and bound by PRC2. This leads to H3K27me3 deposition, which in turn results in canonical PRC1 recruitment[11–14]. To dissect the hierarchy of SS18-SSX targeting at Polycomb sites, we first assessed whether KDM2B, which mediates the recruitment of PRC1.1 via its ZF-CxxC domain[15,20,21], is sufficient to recruit SS18-SSX onto chromatin. To this end, we took advantage of a previously described artificial targeting approach in which KDM2B is fused to the methyl binding domain (MBD) of methyl-CpG binding domain protein 1 (MBD1), leading to its retargeting to regions of densely methylated DNA such as pericentromeric heterochromatin[22,23] (Fig. 1a,b). Additionally, a critical residue in the ZF-CxxC DNA binding domain of KDM2B (ref. [24]) is mutated so that the MBD-fused protein can only bind methylated DNA (MBD-KDM2B$^{K643A}$, referred to as MBD-KDM2B). MBD fused to luciferase (MBD-Luc) was used as control to assess specific targeting (Fig. 1b). We first confirmed the correct tethering of the MBD-fused proteins to heterochromatin using immunofluorescence in a human synovial sarcoma cell line (HS-SY-II) harboring endogenously hemagglutinin (HA)-tagged SS18-SSX1 (ref. [7]). We observed a specific co-localization of the MBD constructs, marked by a V5 tag, to heterochromatin protein 1 (HP1) foci (Extended Data Fig. 1a). As expected, MBD-KDM2B, but not MBD-Luc, was able to recruit the PRC1.1 components BCL6 corepressor (BCOR) and PCGF1, resulting in H2AK119ub1 deposition (Fig. 1c and Extended Data Fig. 1b). Most importantly, MBD-KDM2B was sufficient to recruit SS18-SSX1 (Fig. 1d). To dissect the requirement of PRC2 in PRC1.1-mediated SS18-SSX recruitment, we knocked out components of both complexes using CRISPR–Cas9-directed mutagenesis[25–27] (Extended Data Fig. 1c). Remarkably, depleting the PRC1.1 subunits BCOR or PCGF1, but not the PRC2 components embryonic ectoderm development (EED) or enhancer of zeste homolog 2 (EZH2), significantly reduced SS18-SSX1 recruitment and H2AK119ub1 deposition mediated by MBD-KDM2B (Fig. 1e and Extended Data Fig. 1d). Moreover, here, MBD-KDM2B tethering did not lead to recruitment of PRC2 components nor H3K27me3 accumulation (Extended Data Fig. 1e,f), suggesting that SS18-SSX recruitment is independent of PRC2 presence. Together, these results show that SS18-SSX1 targeting can be initiated by KDM2B, relies on an intact PRC1.1 complex, and is independent from PRC2 activity.

**Fig. 1 | PRC1.1 regulates SS18-SSX recruitment independently of PRC2.**
**a**, MBD-mediated targeting to methylated CpG. Here, KDM2B is redirected to methylated CpG via the MBD. **b**, Schematic of MBD-Luc and MBD-KDM2B fusions containing a V5 tag. MBD-KDM2B contains the histone demethylase domain (JmjC) and a mutated CxxC. **c**, Left: immunofluorescence in HS-SY-II cells of the MBD constructs (V5, magenta) with BCOR (green) and H2AK119ub1 (cyan). Arrowheads point to the MBD foci. Scale bars, 5 µm. Right: percentage of BCOR or H2AK119ub1 foci overlapping a V5 focus in $n = 3$ (MBD-Luc) or $n = 4$ (MBD-KDM2B) biological replicates. Data represent the mean and the s.e.m. **d**, Left: immunofluorescence for V5 (magenta) and SS18-SSX1 (HA, cyan). Right: percentage of HA (SS18-SSX1) foci overlapping a V5 focus. Data represent the mean ± s.e.m. in $n = 5$ biological replicates. **e**, Left: immunofluorescence of MBD-KDM2B (V5, magenta) in the presence of different sgRNAs (eGFP background fluorescence) with SS18-SSX1 (HA, cyan) in HS-SY-II-Cas9 cells. Right: percentage of HA (SS18-SSX1) foci overlapping a V5 focus in $n = 2$ (sgBCOR, sgEZH2), $n = 3$ (sgPCGF1, sgEED) or $n = 4$ (sgCTRL) biological replicates. Data represent the

mean ± s.e.m. $P$ values determined by unpaired one-tailed $t$-test between groups (**$P = 0.003$; ***$P = 0.0005$). **f**, Heatmaps of H2AK119ub1 calibrated ChIP (purple) and SS18-SSX scaled HA CUT&RUN signals (blue) in HS-SY-II-Cas9 cells expressing empty sgRNA as control (sgEV) or targeting $PCGF1$ (sgPCGF1). Both heatmaps represent signals over H2AK119ub1 peaks ($n = 11,099$) called using H2AK119ub1 CUT&RUN in HS-SY-II (Extended Data Fig. 1h). Rows correspond to ±10-kb regions across the midpoint of each enriched region, ranked by increasing signal. **g**, H2AK119ub1 and SS18-SSX corresponding score distributions. **h**, Gene tracks for H2AK119ub1 and SS18-SSX at the $SHH$, $IGF2$, $FGF4$-$FGF3$ and $WNT7B$ loci. **i**, $k$-means clustering of H2AK119ub1 and SS18-SSX $\log_2$ ratio of sgPCGF1 over sgEV. **j**, Salt extraction assay displaying SS18-SSX1 levels by western blot in HS-SY-II-Cas9 cells expressing an empty vector (EV) or sgRNAs against $PCGF1$ or $PCGF3$. **k**, Percentage of total SS18-SSX per salt extraction fractions. Data represent the mean ± s.e.m. of $n = 3$ biological replicates. $P$ values determined by paired one-tailed $t$-test between groups (*$P = 0.03$).

While PRC1.1 is sufficient to initiate SS18-SSX recruitment, several PRC1 complexes can deposit H2AK119ub1. This raises the question of whether PRC1.1 inhibition alone is able to deplete the mark in synovial sarcoma cells, resulting in the loss of SS18-SSX at its target sites. To assess the effect of PRC1.1 inactivation, we knocked out *PCGF1* in HS-SY-II and SYO-1 cells (harboring *SS18-SSX1* and *SS18-SSX2* fusions, respectively) and used cleavage under targets and release using nuclease (CUT&RUN)[28] to assess global changes in H2AK119ub1

and SS18-SSX1/2 chromatin binding. *PCGF1* knockout led to a global decrease in H2AK119ub1 deposition alongside a strong reduction in SS18-SSX1/2 chromatin occupancy in both cell lines, illustrating the pivotal role of variant PRC1.1 in SS18-SSX chromatin maintenance (Fig. 1f–h and Extended Data Fig. 1g–i). To assure a fair comparison between experimental conditions, this result was further verified using H2AK119ub1 calibrated chromatin immunoprecipitation (cChIP)[29] in HS-SY-II (Fig. 1f–h). Moreover, sites exhibiting the highest depletion

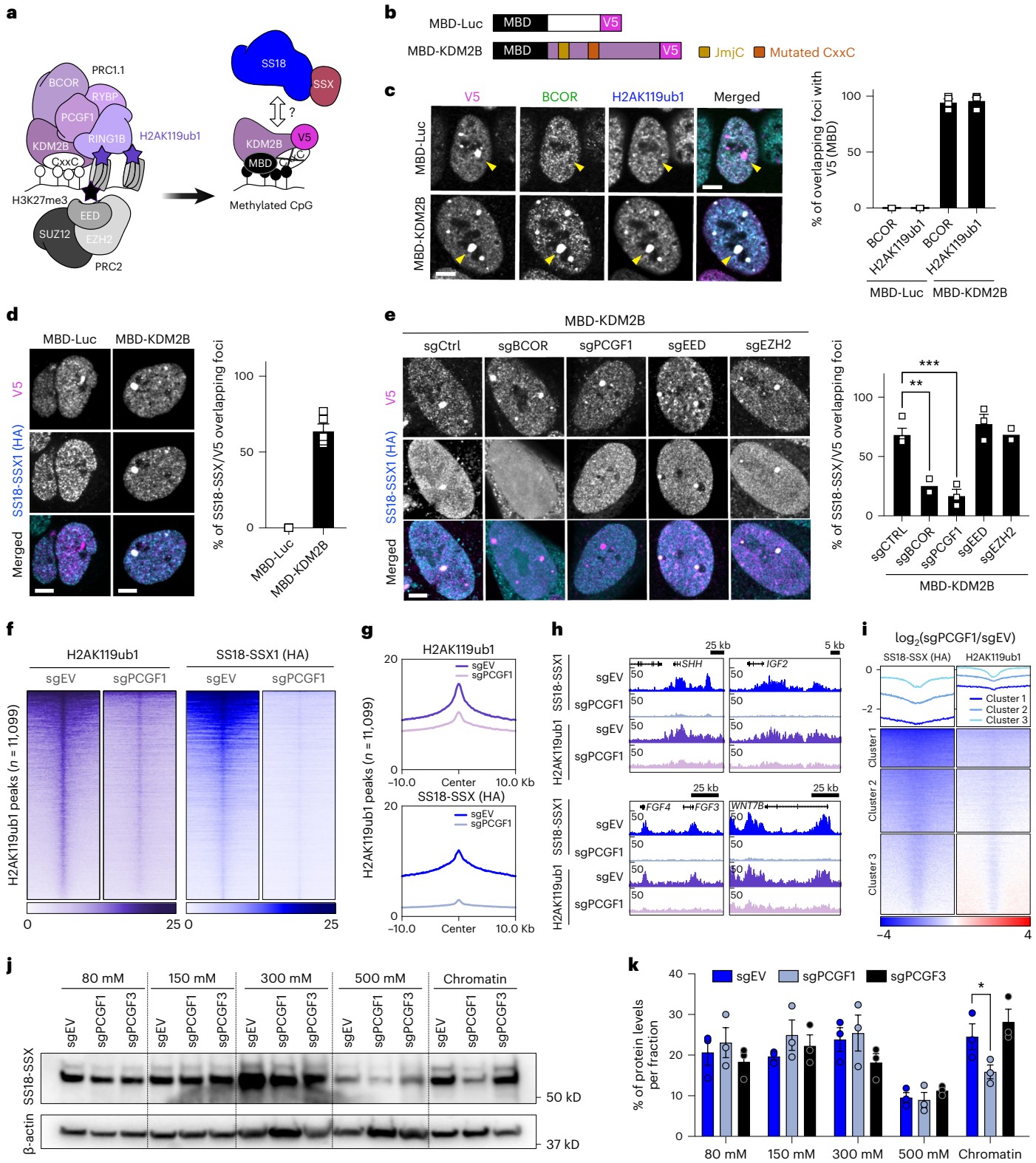

in SS18-SSX binding due to *PCGF1* knockout also exhibited the highest losses in H2AK119ub1, supporting a direct link between these two processes (Fig. 1i).

To further assess if the effect of PRC1.1 is specific, we depleted PCGF3, a member of PRC1.3 and a known dependency in synovial sarcoma (https://depmap.org/portal). As expected, removing either PCGF1 or PCGF3 drastically inhibited synovial sarcoma cell proliferation (Extended Data Fig. 1j, k). We performed chromatin salt extraction in both HS-SY-II and SYO-I synovial sarcoma cell lines to compare the global action of PCGF1 versus PCGF3 on SS18-SSX chromatin binding. Whereas removal of PCGF1 diminished SS18-SSX presence on chromatin, PCGF3 removal did not (Fig. 1j,k). Hence, although PCGF3 is essential for synovial sarcoma maintenance, our results indicate that it is not required for SS18-SSX global chromatin binding, suggesting an alternative role for PRC1.3 in this context. Our data show that PRC1.1 acts as the main depositor of H2AK119ub1 in synovial sarcoma cells and is therefore needed for SS18-SSX chromatin binding.

## SSX-C determines fusion occupancy independently of BAF

Previous studies suggest a model in which introduction of the SSX tail to the BAF complex via SS18 induces changes in complex composition and conformation, allowing its redistribution to H2AK119ub1-rich genomic regions[6]. However, the SSX family of testis-specific proteins have additionally been shown to associate with various members of the Polycomb group complex[30,31], and SSX1 has recently been found to occasionally be fused to partners other than SS18 in synovial sarcoma patient samples[19]. This raises the question of whether the ability to bind Polycomb target genes enriched in H2AK119ub1 is an intrinsic property of SSX proteins that can be exploited by fusion to other transcriptional regulators. To uncouple SS18 and SSX-dependent activities, we started by mapping protein domains in SS18-SSX that are essential for tumor maintenance. We performed a CRISPR–Cas9 knockout screen using a gene-tiling single guide RNA (sgRNA) library covering the entire SS18 and SSX1 coding sequences (Fig. 2a). In this assay, sgRNAs targeting DNA sequences coding for essential protein domains often result in a more significant dropout, as even small in-frame insertion–deletion mutations (indels) in these regions are likely to affect protein function and cell fitness[27,32]. We screened for critical SS18-SSX1 domains in HS-SY-II and used ProTiler to map CRISPR knockout hypersensitive (CKHS) regions[33]. In line with a key role for SS18-containing BAF complexes in these cells, sgRNAs targeting SS18 were generally depleted, with the exception of those targeting a region that is not present in the shorter isoform of SS18 (amino acids 295–325). However, a clear CKHS region was identified at the SSX C terminus corresponding to the highly conserved SSX repression domain (SSXRD)[34,35] (Fig. 2b). These results suggest that this region, consisting of the last 34 amino acids of SS18-SSX1, is the most critical for its oncogenic function.

To explore the intrinsic ability of the SSXRD in specific chromatin binding, we generated constructs containing enhanced green fluorescent protein (eGFP) fused to the SSX1 C-terminal region present in SS18-SSX1, with or without an SSXRD deletion (SSX-C^ΔRD and SSX-C, respectively) or the SSXRD alone. SS18 and SS18-SSX1 eGFP fusions were used as controls (Extended Data Fig. 2a). When expressed in the human embryonic kidney cell line HEK293T, eGFP-SS18 exhibited both nuclear and cytoplasmic localization. In contrast, eGFP-SS18-SSX1 and SSX-C were exclusively detected in the nucleus in an SSXRD-dependent manner (Extended Data Fig. 2b). This supports the presence of a nuclear localization signal in SSXRD (ref. 31) and a role in mediating chromatin interaction. Sequential salt extractions further showed that SSXRD-containing GFP fusions, but not eGFP-SSX-C^ΔRD, are predominantly present in the chromatin fraction, confirming that the SSX1 C terminus strongly binds chromatin via this domain (Extended Data Fig. 2c, d). To identify factors that contribute to SSX-C/SSXRD chromatin binding, we studied their common interactome through eGFP co-immunoprecipitation followed by mass spectrometry (Extended Data Fig. 2e and Supplementary Table 1). Noticeably, histones were highly represented in both SSX-C and SSXRD top interactors, with higher enrichment than PRC1 or PRC2 components. These results indicate that SSX-C alone can bind chromatin via a direct histone interaction, consistent with recent biochemistry studies that showed the ability of SS18-SSX to bind the nucleosome acidic patch, with a preference for H2AK119ub1-modified nucleosomes conferred by the last five amino acids (EEDDE) of the SSXRD (refs. 6,35). Accordingly, a mutant lacking this region (SSX-C^E184*) lost the specific co-localization of SSX to H2AK119ub1-rich Barr bodies in HEK293T (Fig. 2c). We further confirmed the preferential interaction of SSX-C with H2AK119ub1-modified nucleosomes in live cells using NanoBRET, a protein–protein interaction assay based on bioluminescence resonance energy transfer (BRET)[36,37]. We detected an interaction of the SSX-C (Halo-SSX-C) when co-expressed with histone H2A fused to NanoLuc luciferase (NLuc-H2A) which was dependent on the SSXRD domain and was diminished in a mutant lacking the last five amino acids (SSX-C^E184*). Most importantly, expression of a mutant H2A that cannot be ubiquitinated (NLuc-H2A^K118K119R)[38] decreased the ability of SSX-C to interact with the nucleosome in vivo (Extended Data Fig. 2f, g).

These results confirm that H2AK119ub1 plays an active role in specifying the chromatin occupancy of SS18-SSX mediated by SSXRD, as previously described[6]. However, to understand if SSX-C alone is sufficient to reproduce SS18-SSX binding patterns, we performed chromatin immunoprecipitation with high-throughput sequencing (ChIP–seq) of eGFP-SSX-C overexpression in HS-SY-II cells. This revealed a clear overlap with previously identified SS18-SSX/KDM2B bound regions, which was abolished in the absence of the SSXRD domain (Fig. 2d). This result suggests that SS18-SSX chromatin binding patterns are a consequence of SSX-C specificity regardless of the SS18 fusion partner. Notably, like SS18-SSX, SSX-C co-localizes at

**Fig. 2 | SSX C terminus directs SS18-SSX chromatin binding independently of BAF. a**, Layout of CRISPR–Cas9 knockout tiling screen. **b**, Mapping of CKHS regions in SS18-SSX1 using ProTiler based on log₂(fold change) (LFC) in representation of sgRNAs targeting *SS18-SSX1* in HS-SY-II. The CKHS region is highlighted in dark red and corresponds to the SSXRD Pfam sequence (PF09514). **c**, Immunofluorescence of HEK293T cells expressing eGFP constructs (cyan) stained for H2AK119ub1 (magenta). Images are representative of three biological replicates. Yellow arrowheads indicate the Barr body. Scale bar, 5 μm. **d**, Left: heatmaps for SS18-SSX1 (endogenously HA tagged) and KDM2B ChIP–seq from ref. 7 and HA ChIP in HS-SY-II cells expressing HA-eGFP fused to SSX-C or SSX-C^ΔRD over SS18-SSX1 peaks (n = 26,805). Rows correspond to ±10-kb regions across the midpoint of each HA-enriched region, ranked by increasing signal. Right: gene tracks for SS18-SSX1, KDM2B and HA ChIP–seq at the *SHH* and *FGF4-FGF3* loci. **e**, Left: CUT&RUN heatmaps in HS-SY-II cells for SS18-SSX1 (endogenously HA tagged) and HA-eGFP fused to SSX-C without and with SS18-SSX depletion mediated by shRNA. Heatmaps represent CUT&RUN signals over H2AK119ub1

peaks (n = 11,099). Rows correspond to ±10-kb regions across the midpoint of each HA-enriched region, ranked by increasing signal. Right: CUT&RUN gene tracks at the *SHH* and *FGF4-FGF3* loci. **f**, Left: heatmaps for HA CUT&RUN in KHOS-240S cells expressing HA-SS18, HA-SS18-SSX1 or HA-eGFP-SSX-C. Heatmaps represent CUT&RUN signals over all HA peaks (n = 58,843). Rows correspond to ±10-kb regions across the midpoint of each HA-enriched region, ranked by increasing signal. Right: CUT&RUN gene tracks at the *SHH* and *FGF4-FGF3* loci. **g**, Immunofluorescence of HEK293T cells expressing eGFP-SS18-SSX1 (cyan) treated with DMSO (top) or 500 nM ACBI1 (bottom) stained for BAF subunits (magenta) SMARCA2 (BRM, left), ARID1A (middle) or SMARCC1 (right). Images are representative of two biological replicates. **h**, Immunofluorescence of MBD constructs (V5, cyan) and SS18-SSX1 (HA), SMARCA2 or SMARCC1 (magenta) in HS-SY-II cells not treated (MBD-Luc) or treated with DMSO or 500 nM ACBI1 (MBD-KDM2B). **i**, Percentage of SS18-SSX1 (HA), SMARCA2 or SMARCC1 foci overlapping a V5 focus. Data represent the mean of two biological replicates.

H2Aub-rich regions when overexpressed in HEK293T, further indicating that SSX determines the fusion binding profile independently of SS18 (Fig. 2c). To uncouple SSX-C specificity from SS18-SSX-mediated

BAF complex deregulation, we compared SSX-C binding patterns in HS-SY-II sarcoma cells in the presence or absence of SS18-SSX via inducible short hairpin RNA (shRNA) knockdown (Extended

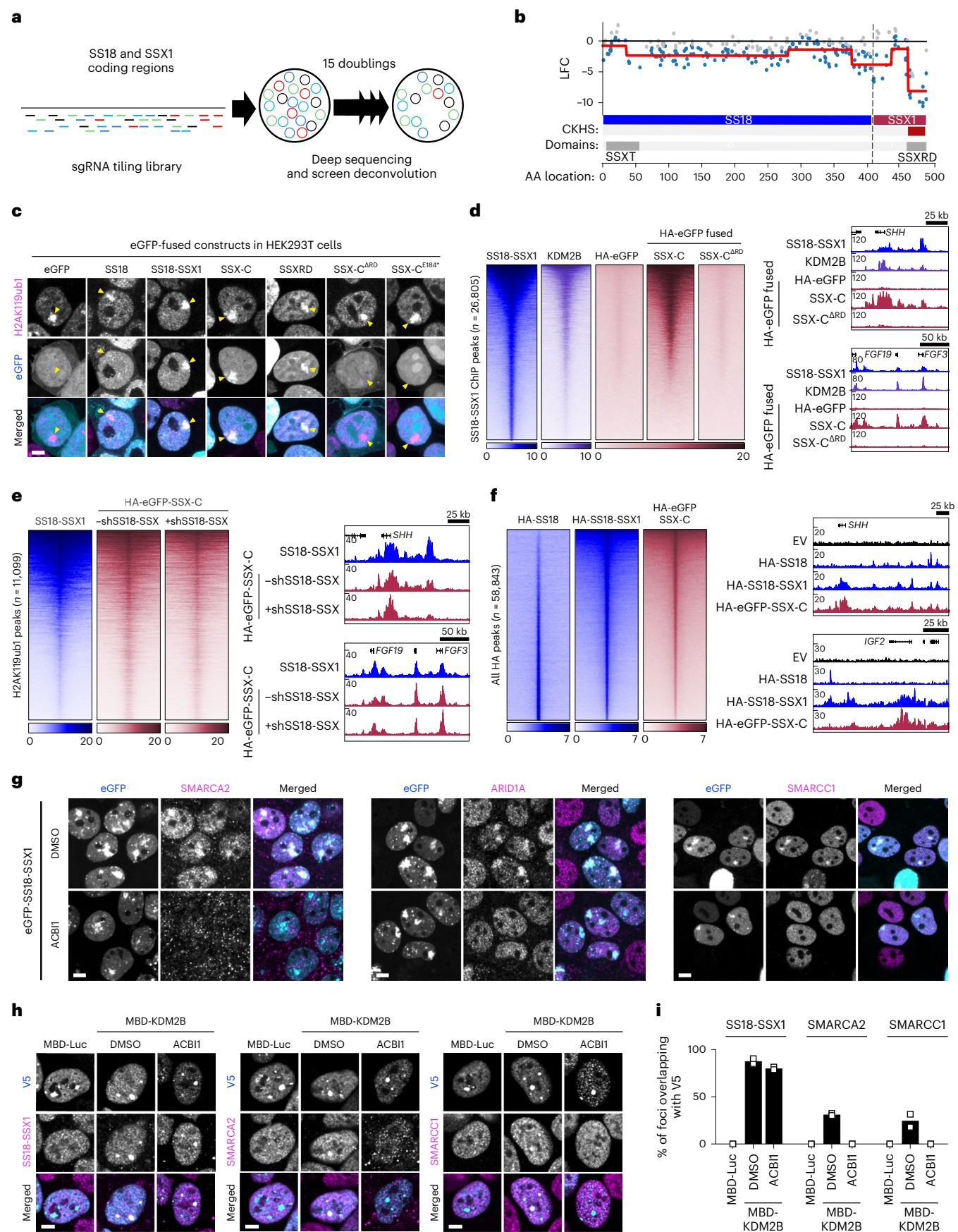

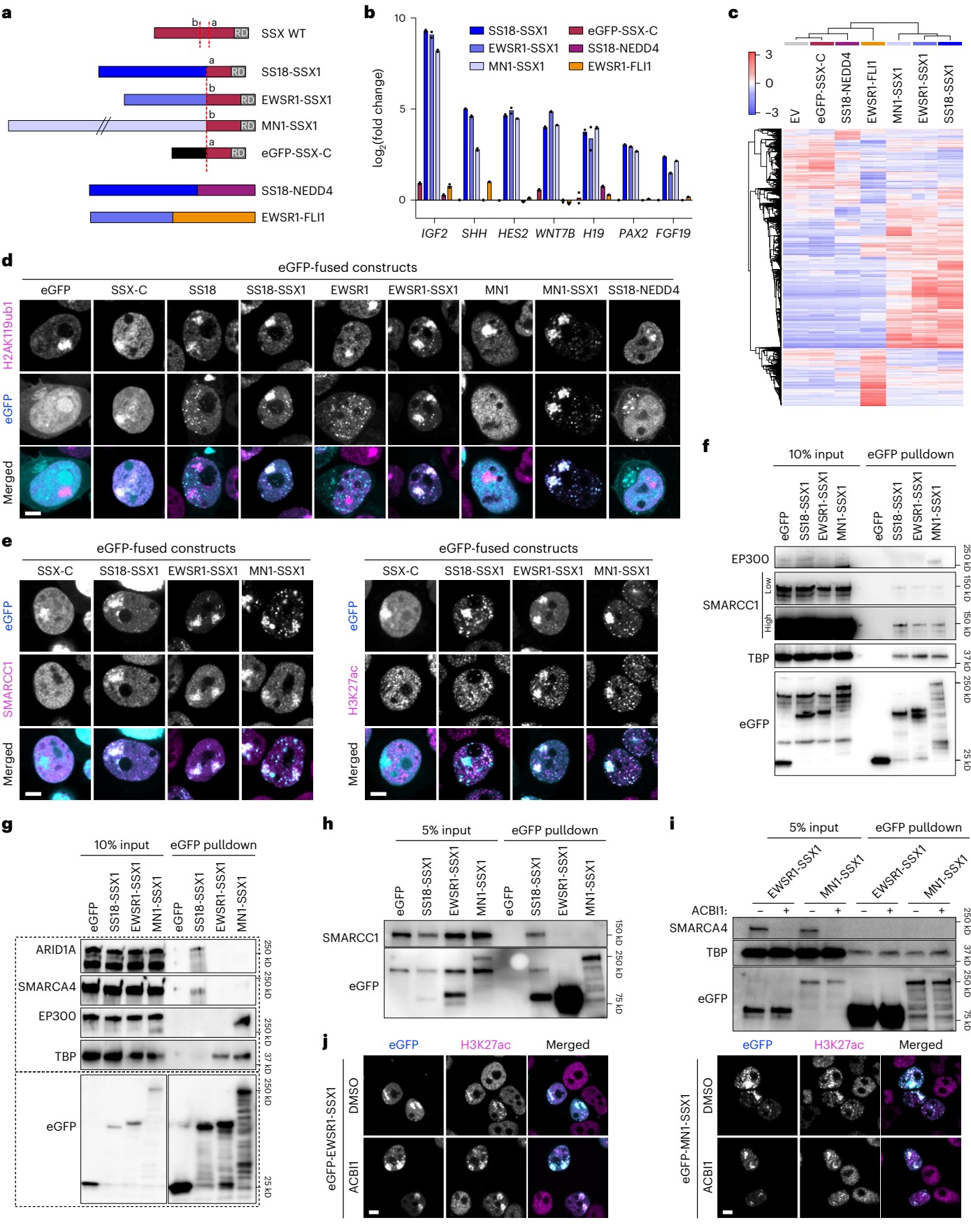

**Fig. 3 | New SSX fusions drive similar gene signature via alternative activators. a**, Schematic representing cloned constructs. SS18-SSX1 and SSX-C contain the canonical breakpoint 'a', whereas EWSR1-SSX1 and MN1-SSX1 exhibit an alternative breakpoint 'b'. WT, wild type. **b**, The $\log_2$-transformed fold change of FPKM values in hMSCs expressing the new fusion constructs and controls at synovial sarcoma signature genes. Data represent the mean of two biological replicates. **c**, RNA-seq heatmap showing the 1,000 most variable genes with a cutoff $z$-score of 4. **d**, Immunofluorescence of HEK293T cells expressing eGFP constructs (cyan) stained for H2AK119ub1 (magenta). Images are representative of three biological replicates. Scale bar, 5 µm. **e**, SMARCC1 (left) or H3K27ac (right) immunofluorescence of HEK293T cells expressing the indicated eGFP constructs. Bottom panels display merge channels with eGFP (cyan) and SMARCC1 or H3K27ac (magenta). Images are representative of three biological replicates. **f**, Co-immunoprecipitation pulling down on eGFP in HEK293T cells expressing eGFP constructs representing one replicate. **g**–**i**, Co-immunoprecipitation pulling down on eGFP in HEK293T cells expressing eGFP constructs with harsher chromatin shearing conditions. All co-immunoprecipitations were repeated in two independent replicates. **j**, H3K27ac immunofluorescence of HEK293T cells expressing eGFP-EWSR1-SSX1 (left) or eGFP-MN1-SSX1 (right) treated with DMSO (top) or 500 nM ACBI1 (bottom) for SMARCA2 (BRM, left), ARID1A (middle) or SMARCC1 (right). Merge channels display eGFP (cyan) and BAF H3K27ac (magenta) overlays. Images are representative of three biological replicates.

Data Fig. 2h). SSX-C chromatin binding at H2AK119ub-rich regions remained unchanged upon fusion knockdown (Fig. 2e), indicating that SSX-C specificity is independent of the presence of an altered BAF complex. To further confirm this, we profiled SS18, SS18-SSX and SSX-C in an SS18-SSX-negative human osteosarcoma cell line (KHOS-240S). SS18 and SSX bound distinct chromatin regions, with SS18-SSX overall occupancy correlating more strongly with that of SSX-C (Fig. 2f and Extended Data Fig. 2i). Moreover, removal of the BAF complex ATPases SMARCA2/4 and PBRM1 using the ACBI1 PROTAC degrader[39,40] (Extended Data Fig. 2j) did not affect SS18-SSX localization at H2Aub-rich regions in HEK293T cells (Fig. 2g). Importantly, although ACBI1 treatment specifically resulted in the depletion of SMARCA4 without affecting the levels of other BAF complex members (Extended Data Fig. 2j), it abolished the relocalization of BAF subunits SMARCC1 and ARID1A to Barr bodies (Fig. 2g). These results are consistent with the modular assembly of BAF complexes where SS18 is recruited to the complex via its ATPase module[5], and show that inhibition of the catalytic activity of BAF in synovial sarcoma results in the loss of SS18-mediated BAF complex recruitment to H2Aub-rich regions. This was further confirmed using the MBD recruitment assay in synovial sarcoma cells. Again, ACBI1 treatment abolished BAF complex recruitment as shown by the lack of the SMARCC1 core subunit at MBD-KDM2B foci. Still, de novo SS18-SSX recruitment to these regions was unaffected by the absence of the BAF complex (Fig. 2h,i). Together, our results demonstrate that the SSX-C terminus, via its SSXRD, confers specificity to H2AK119ub1-rich regions in the genome and mediates SS18-SSX binding independently of SS18 and the BAF complex.

**Novel SSX fusions activate a synovial sarcoma gene signature**

That SSX-C binding patterns remain unchanged regardless of the presence of an altered BAF complex suggests that SSX-C specificity to H2AK119ub1-rich regions could be exploited by fusion to other partners. The recently identified alternative SSX fusion partners that can replace SS18 in synovial sarcoma involve the transcriptional activators EWSR1 and MN1 (ref. 19). We sought to investigate if these alternative partners can substitute the function of BAF in activating a synovial sarcoma gene signature. First, we expressed *EWSR1-SSX1* and *MN1-SSX1* in human mesenchymal stem cells (hMSCs) alongside *SS18-SSX1*. For comparison, *EWSR1-FLI1* (pathognomonic of Ewing sarcoma[41]) and *SS18-NEDD4* (which has been found in one case described as a primary renal synovial sarcoma, two cases of myxoid morphology and in an epithelioid sarcoma[42,43]) were also expressed in hMSCs (Fig. 3a). While EWSR1-FLI1 and SS18-NEDD4 led to distinct gene expression changes, all SSX1-containing fusions clustered together and resulted in a specific upregulation of Polycomb target genes characteristic of a synovial sarcoma gene signature[7] (Fig. 3b, c and Supplementary Table 2). Accordingly, all SSX fusions retained the ability to localize to Barr bodies enriched in H2AK119ub1 as shown in HEK293T cells. The fusion partners on their own and SS18-NEDD4 exhibited a diffuse nuclear pattern, further showing that specificity is conferred by the SSX1 tail regardless of its fusion partner (Fig. 3d).

In line with previous studies reporting an interaction of the BAF complexes with EWSR1 and MN1 (refs. 44,45), we observed that all SSX fusions resulted in rewiring of the BAF to Barr bodies (Fig. 3e). However, only EWSR1-SSX1 and MN1-SSX1, but not SS18-SSX, led to the deposition of H3K27ac (Fig. 3e and Extended Data Fig. 3a), indicating that the new SSX fusions use alternative routes to deregulate Polycomb target genes. Indeed, while all fusions were able to pull down SMARCC1, as well as TATA-binding protein (TBP), MN1-SSX specifically interacted with EP300 (Fig. 3f). This is in line with previous studies demonstrating a synergistic effect of EP300 and MN1 as transcriptional co-activators[46]. Using more stringent chromatin shearing conditions for immunoprecipitation, we observed a specific interaction of BAF complex subunits with SS18-SSX1, while EP300 and TBP immunoprecipitated with MN1-SSX and both EWSR1-SSX and MN1-SSX, respectively (Fig. 3g,h). Notably, interaction of EWSR1-SSX or MN1-SSX1 with TBP was not affected by ACBI1 treatment (Fig. 3i). Recruitment of both fusions to Barr bodies or consequent H3K27ac deposition was also not affected (Fig. 3j and Extended Data Fig. 3b). These results indicate that the deposition of H3K27ac by alternative SSX fusions is mediated by strong interactions with transcriptional activators such as EP300 and TBP, but does not rely on BAF activity. Accordingly, gene activation

**Fig. 4 | SSX-C increases PRC1.1 stability, thus reinforcing H2AK119ub1 levels and SS18-SSX occupancy. a**, The $\log_2$-transformed fold change of FPKM values in hMSCs expressing the new fusion constructs and controls for *BCOR* mRNA levels. Data represent the mean of two biological replicates. **b**, Western blot of whole cell extracts of HS-SY-II cells expressing shRNA against SS18-SSX over a time course of 0–72 h of doxycycline (DOX) induction. Blot is representative of four biological replicates. **c**, Quantitative PCR (qPCR) displaying $\log_2$-transformed fold change of mRNA levels normalized by *GAPDH* HS-SY-II cells expressing shRNA against SS18-SSX over a time course of 0–72 h of doxycycline induction. Data are relative to time 0 and represent the mean of two biological replicates. **d**, Immunofluorescence in HEK293T cells (left) or hMSCs (right) expressing the indicated eGFP-fused constructs with nuclei stained with DAPI, eGFP signals and H2AK119ub1 stainings. Scale bars, 20 µm. Images are representative of two independent replicates throughout the figure. **e**, Immunofluorescence against BCOR and H2AK119ub1 in hMSCs expressing eGFP-fused constructs. **f**, Quantification of BCOR and H2AK119ub1 fluorescence ratio in high versus low eGFP in hMSCs. Data represent the mean of two biological replicates. *P* values determined by ratio of paired one-tailed *t*-test between groups (*$P$ = 0.047 for BCOR and *$P$ = 0.02 for H2AK119ub1). **g**, Top: sequential chromatin washes assay using 150 mM salt buffer in uninduced control (Ctrl) or eGFP-SSX-C-expressing HEK293T cells. BCOR, PCGF1 or β-actin as loading control was detected by western blot. Bottom: quantification of the protein distribution for BCOR, PCGF1 or β-actin in the various washes. Data represent the percentage of total protein levels of one replicate. **h**, Left: immunofluorescence against SS18 in HS-SY-II cells expressing the indicated eGFP-fused constructs. Right: quantification of the SS18 fluorescence ratio in high versus low eGFP cells. Data represent the mean of two biological replicates. *P* values determined by ratio of paired one-tailed *t*-test between groups (*$P$ = 0.03). **i**, Hematoxylin and eosin (H&E) and immunohistochemical staining for inhibin-α, SSX and H2AK119ub1 in human testis. Scale bar (top), 40 µm. Bottom panel displays insets of the areas marked by dashed lines in the top panel. Scale bar (bottom), 20 µm.

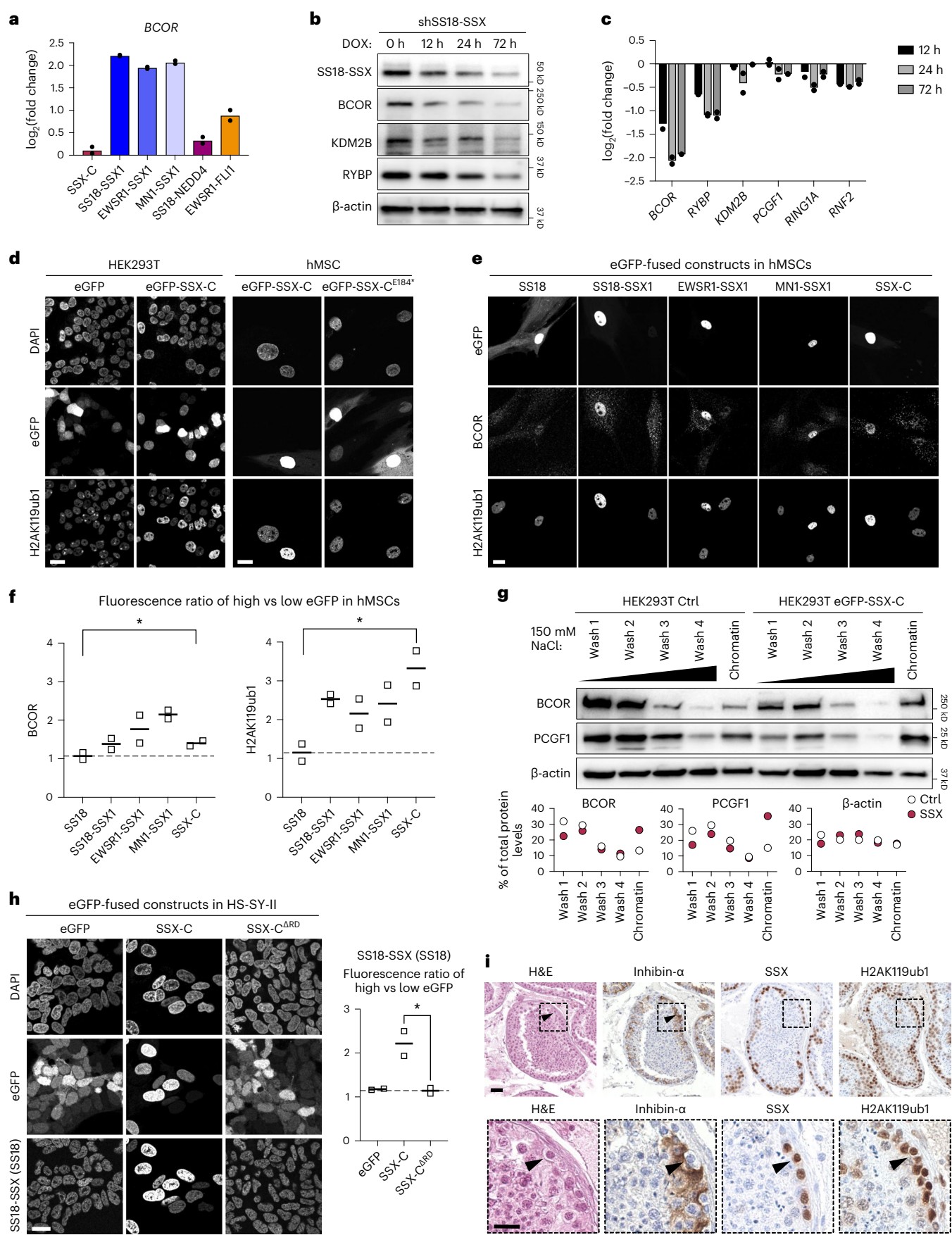

by EWSR1-SSX1 or MN1-SSX1 was not affected by ACBI1 treatment (Extended Data Fig. 3c). Together, our results show that the induction of Polycomb target genes that define a synovial sarcoma signature can be achieved by the recruitment of transcriptional co-activators as a result of fusion of SSX-C to different partners.

### SSX-C reinforces H2AK119ub1 via increased PRC1.1 stability

Consistent with a critical role for PRC1.1 in depositing H2AK119ub1 in synovial sarcoma, its subunit BCOR is upregulated in synovial sarcoma tumor samples[47,48]. In fact, all SSX-containing fusions resulted in increased *BCOR* expression, and indeed *BCOR* is a direct target of both SS18-SSX and SSX-C (Fig. 4a and Extended Data Fig. 4a). Reciprocally, in publicly available data, SS18-SSX knockdown in HS-SY-II (SS18-SSX1) and SYO-I (SS18-SSX2) synovial sarcoma lines led to a concomitant decrease in *BCOR* (ref. 16) (Extended Data Fig. 4b). This suggests an interplay between SSX fusions and PRC1.1 regulation. However, although inducible *SS18-SSX* knockdown readily affected the protein levels of several PRC1.1 members, it did not greatly affect the mRNA levels of all of them (Fig. 4b,c and Extended Data Fig. 4c), indicating additional regulation at the protein level. Since SSX-C does not act as a transcriptional activator (Fig. 3b,c), but directly interacts with chromatin, we hypothesized that it could augment PRC1.1 protein levels by increasing stabilization of the complex on chromatin. To assess this, we overexpressed eGFP-SSX-C in HS-SY-II, HEK293T and hMSC cellular contexts and measured its effect on BCOR and H2A119ub1 levels. eGFP-SSX-C expression in synovial sarcoma cells led to higher BCOR and H2AK119ub1 levels in a manner that correlated with eGFP reporter levels. The same was not observed when expressing an eGFP-only control or an SSX-C mutant lacking the SSXRD domain, where H2AK119ub1 or BCOR staining remains homogeneous regardless of the amount of construct in the cell (Fig. 4d and Extended Data Fig. 4d, e). Similarly, SSX-C overexpression in mesenchymal stem cells recapitulated the increase in BCOR and H2AK119ub1 levels, and indeed all SSX-containing fusions had the same effect (Fig. 4e,f and Extended Data Fig. 4f). Of note, overexpression of SSX-C alone did not induce *BCOR* transcription, indicating that SSX fusions, via their C-terminal tail, also regulate PRC1.1 at the protein level (Fig. 4a). Sequential chromatin washes in HEK293T cells and chromatin salt extractions in HS-SY-II cells showed that SSX-C expression increases the presence of the PRC1.1 proteins BCOR and PCGF1 in the chromatin fraction while decreasing their presence in more soluble fractions (Fig. 4g and Extended Data Fig. 4g,h). These results show that SSX-C alone is able to increase total H2AK119ub1 levels in part by stabilizing PRC1.1 presence on chromatin. Notably, by increasing PRC1.1 stability and H2AK119ub1 levels, SSX-C overexpression also affected SS18 levels, which serve as a proxy for SS18-SSX1 in synovial sarcoma cells (Fig. 4h). Again, SSX-C acts on the protein level, as it does not bind the *SS18* promoter or increase *SS18* mRNA levels (Extended Data Fig. 4i, j). This indicates that in enhancing H2AK119ub1, SSX-C is also able to reinforce fusion binding. Together, these results demonstrate that SSX fusions promote PRC1.1 activity via transcriptional and SSX-C-mediated mechanisms.

Given that SSX-C alone has the ability to both recognize and further induce H2AK119ub1, we reasoned that this could reflect a role of SSX proteins in their physiological context. To explore this, we investigated whether wild-type SSX1 levels are associated with H2AK119ub1 in human testis where the SSX1 protein is normally expressed. Publicly available single-cell RNA sequencing data from human testis show that *SSX1* is mainly expressed in spermatogonial stem cells, differentiating spermatogonia and in early spermatocytes, but not in other testicular cell types (Extended Data Fig. 4k)[49]. Immunohistochemical staining of human testis revealed that H2AK119ub1 levels are not homogeneous, but rather are particularly high in cells around the outer edge of the seminiferous tubules next to the basal lamina that correspond to spermatogonia (inhibin-α-negative cells) where SSX1 is also specifically

detected (Fig. 4i). These results suggest that the physiological role of SSX proteins is also linked to PRC1 function.

### High levels of H2AK119ub1 are a feature of synovial sarcoma

The above in vitro results uncovered a link between SSX-C and PRC1.1 and suggest that high levels of H2AK119ub1 are acquired during tumorigenesis to further enable SS18-SSX binding. To assess if the SS18-SSX oncoprotein promotes H2AK119ub1 in vivo, we took advantage of a synovial sarcoma mouse model in which *SS18-SSX2* expression is conditionally induced in *Hic1*-positive mesenchymal progenitors[50,51] (Fig. 5a). Similar to our observations in cell culture, SS18-SSX-positive tumor cells (marked by GFP) specifically exhibited high levels of H2AK119ub1 when compared with normal muscle (Extended Data Fig. 5a–c). Moreover, increased levels of H2AK119ub1 were clearly detected at earlier time points following SS18-SSX induction, as early as 5 weeks after induction and with a steady increase that was concomitant with the time course of tumor formation. Similarly, BCOR levels increased during this time course, again pointing to increased expression and stability of PRC1.1 in response to fusion expression (Fig. 5b–e). These results indicate that SS18-SSX activation induces BCOR and H2AK119ub1 deposition early during murine tumorigenesis.

Lastly, we reasoned that if this autoregulatory feedback loop has a role in human sarcomagenesis, increased levels of H2AK119ub1 would be a feature of human synovial sarcomas. To test this, we performed H2AK119ub1 immunohistochemistry on a synovial sarcoma tissue microarray of 37 patient samples. H2AK119ub1 exhibited stronger nuclear staining in synovial sarcomas than in other sarcomas and normal tissues, including skeletal muscle (Fig. 5f,g). Consistent with an autoregulatory feedback loop in which SS18-SSX increases H2AK119ub1 to promote its own binding and stability, we observed a positive correlation between H2AK119ub1 staining and staining using SSX-specific or SS18-SSX-specifc antibodies (Extended Data Fig. 5d). These results show that SS18-SSX activity is also associated with enhanced H2AK119ub1 in human synovial sarcoma and suggest an autoregulatory mechanism in which the oncofusion can potentiate its own chromatin binding and therefore its oncogenic activity.

## Discussion

Our study addresses the molecular mechanism underlying SS18-SSX chromatin recruitment in synovial sarcoma. We confirm that H2AK119ub1 is important for SS18-SSX specific chromatin targeting[6], and further show that in synovial sarcoma, PRC1.1 is central in establishing H2AK119ub1 deposition and orchestrating oncofusion protein occupancy and maintenance, with PCGF1 removal leading to global erosion in SS18-SSX binding. These results support a role for PRC1.1 as the main contributor of genome-wide H2AK119ub1 deposition as observed in mouse embryonic stem cells[8], and suggest that other variant PRC1 complexes may have alternative roles in synovial sarcoma.

We demonstrate that the most critical domain of SS18-SSX1 for synovial sarcoma cell maintenance is at the SSX-C terminus, where only 34 amino acids are sufficient to determine binding patterns of the oncofusion protein on chromatin. This highlights the critical role of the SSXRD domain in the precise recruitment of SS18-SSX at specific synovial sarcoma gene targets. Indeed, the SSX1 tail alone can reproduce the genome-wide occupancy of SS18-SSX1 in an SSXRD-dependent manner, and de novo oncofusion recruitment occurs independently of the BAF complex. This is consistent with the recent finding that some synovial sarcomas harbor translocations in which SSX is fused not to SS18, but rather to alternative partners including EWSR1 and MN1 (ref. 19). Such occurrences support the notion that the SSXRD domain, by mediating recruitment of transcriptional activators to induce Polycomb target genes during sarcomagenesis, is the key determinant of a synovial sarcoma signature, and that direct deregulation of the mSWI/SNF (BAF) complex through SS18 is not essential to all synovial

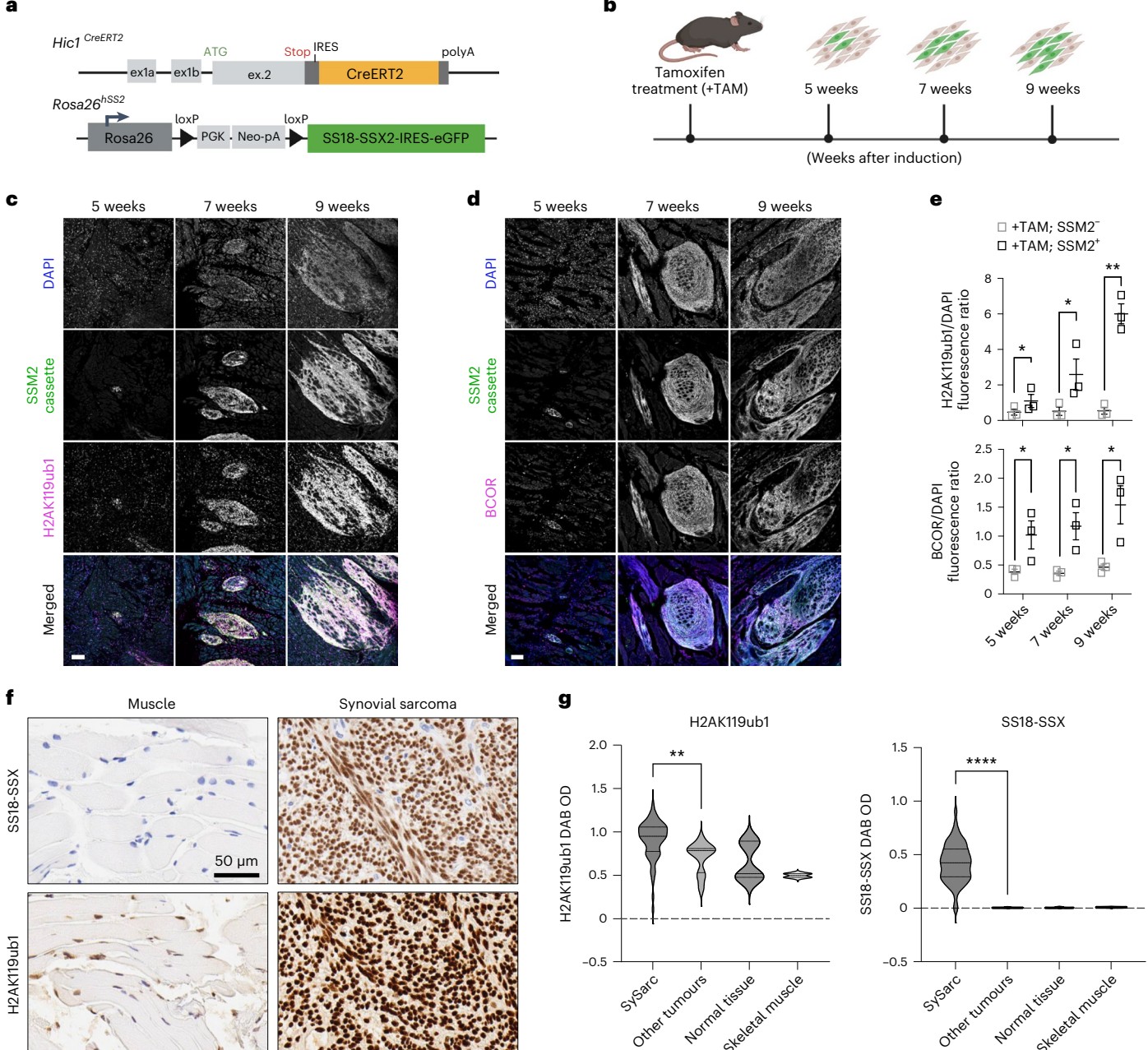

**Fig. 5 | High levels of H2AK119ub1 are acquired during synovial sarcoma development. a**, Overview of the *Hic1^CreERT2* knock-in allele[50] and of the *Rosa26-hSSM2* allele (Rosa26^hSS2)[51] for conditional induction of SS18-SSX2 in Hic1-expressing mesenchymal progenitors. Upon tamoxifen treatment, CreERT2 mediates recombination between the two LoxP sites in SSM2 mice, thereby removing the transcriptional stop signal and allowing transcription of SS18-SSX2⁻IRES-EGFP from the endogenous ROSA26 promoter. **b**, Illustration of the timeline for the tissue sample collection of samples analyzed in **c** and **d**. Eight-week-old mice were treated with tamoxifen, and tongue muscle tissues were collected at 5, 7 and 9 weeks after induction. **c,d**, Immunofluorescence of *Hic1^creERT2/creERT2*; *Rosa26^SSM2/SSM2*, Cre-positive mouse tongue tissue at 5, 7 or 9 weeks after induction. The cells are stained for DAPI, SSM2 (eGFP) and H2AK119ub1 (**c**) or BCOR (**d**). Scale bars, 100 μm. **e**, Quantification of H2AK119ub1 (top) and BCOR (bottom) signal intensity normalized to DAPI signal intensity in three biological replicates (three different mice) in tamoxifen treated mice (+TAM) expressing or not expressing the SSM2 cassette (human SS18-SSX2) and showing

normal tongue muscle (+TAM; SSM2⁻) adjacent to synovial sarcoma tumors (+TAM; SSM2⁺). *P* values determined by paired one-tailed *t*-test between groups (from left to right, *P = 0.04, *P = 0.04, **P = 0.002 (H2AK119ub1); *P = 0.03, *P = 0.04, *P = 0.04 (BCOR)). **f**, Immunohistochemical staining for H2AK119ub1 on a tissue microarray of human surgical excised tissue specimens (left, skeletal muscle; right, synovial sarcoma). Scale bar, 50 μm. **g**, Quantification of H2AK119ub DAB signal intensity across 37 synovial sarcomas (sample cores in duplicate), other sarcomas (one case each of epithelioid sarcoma, sarcomatoid mesothelioma, Ewing sarcoma, sarcomatoid renal cell carcinoma, clear cell sarcoma, dedifferentiated liposarcoma and myxoid liposarcoma) and normal tissues (normal skeletal muscle, ovarian stroma, breast glandular tissue and testis controls). Quantification for the two skeletal muscle samples is also shown separately in the graph. All samples were stained in parallel on the same formalin-fixed, paraffin-embedded tissue microarray slide. *P* values determined by Mann−Whitney *U*-test between groups (**P = 0.001 (H2AK119ub1), ****P < 0.0001 (SS18-SSX)).

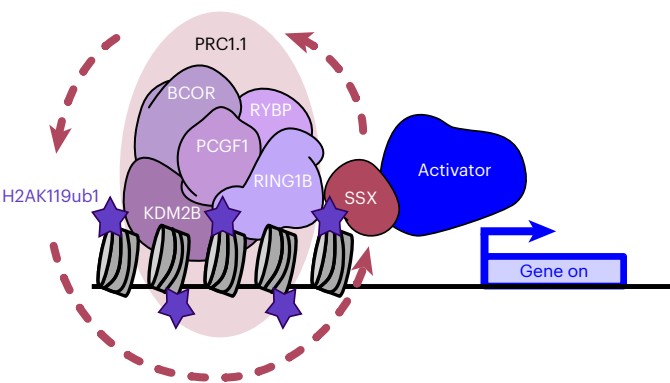

**Fig. 6 | Model.** Model depicting the strong interplay between SSX and H2AK119ub1. SSX-C interacts with regions rich in ubiquitinated H2A on lysine K119 and therefore determines oncofusion chromatin occupancy. SSX-C binding further enhances H2AK119ub1 levels, reinforcing the presence of the SSX fusion on chromatin. Aberrant activation of Polycomb target genes is mediated by the recruitment of different transcriptional activators via their SSX-C domain.

## Online content

sarcomas. A limitation of our study is the use of ectopic expression of these new fusions for mechanistic studies, which may not reproduce the physiological levels observed in tumors. Future work will be needed to generate patient-derived cell lines or murine models in which the molecular activity of these alternative SSX fusions can be studied in more detail.

Our data also reveal an interplay between SS18-SSX and PRC1.1 activity leading to a positive feedback loop that results in increased H2AK119ub1 in murine and human synovial sarcomas. Two distinct mechanisms mediate this interplay. On one hand, SS18-SSX binds to and positively regulates the transcriptional level of PRC1.1 gene *BCOR*. On the other hand, the SSX-C terminus induces an increase in H2AK119ub1 by stabilizing PRC1.1 complex protein levels and chromatin binding. In increasing H2AK119ub1 levels, SS18-SSX and other SSX fusions are able to further promote the mark that they recognize, a process that will increase their presence on chromatin (Fig. 6). This model is in agreement with a previous study showing that RYBP chromatin occupancy is increased by SS18-SSX expression in murine mesenchymal stem cells[52]. The feedback loop that we identify is also reminiscent of the role of RYBP in the PRC1 complex, where it both promotes interactions within the complex leading to increased complex stability[29] and recognizes and binds H2AK119ub1-modified nucleosomes to further promote H2AK119ub1 deposition[53]. This work further highlights the central role that PRC1 activity, and its derivate H2AK119ub1 histone mark, plays in driving and sustaining synovial sarcoma and supports inhibition of PRC1.1 as a potential therapeutic strategy. These findings are also important in light of a putative role for full-length wild-type SSX family proteins, which have been reported to be expressed in synovial sarcomas[3,54], in further promoting oncofusion protein activity. Moreover, SSX proteins are cancer-testis antigens that are abnormally present in various cancers such as melanoma, breast cancer and prostate cancer[55,56]. Therefore, the interplay between SSX-C and H2AK119ub1 could affect a wider range of other malignancies. It remains to be determined if H2AK119ub1 levels are increased in SSX-positive cancers and whether they play an oncogenic role.

Our study describes a central role for PRC1.1-deposited H2AK119ub1 in driving synovial sarcoma, thus highlighting a key role for this complex beyond cell fate decisions and development, which is further supported by the occurrence of main driving genetic events involving *BCOR* in several pediatric tumors[57–62]. Further studies will uncover the extent to which 'PRC1-dependent' tumors share molecular characteristics and circuitries that could be exploited therapeutically.

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

## Methods

### Cell culture

Human synovial sarcoma cell lines HS-SY-II (RRID: CVCL_8719)[63] and SYO-1 (RRID: CVCL_7146)[64] were obtained from their original source laboratories. Human osteosarcoma KHOS-240S (RRID: CVCL_2544) and human embryonic kidney HEK293T (RRID: CVCL_0063) cell lines were purchased from the American Type Culture Collection (ATCC). HEK293GP cells used for retrovirus production were obtained from Takara Bio (631458). Cells were cultured in DMEM (Gibco) supplemented with 10% fetal bovine serum (FBS) and penicillin-streptomycin. The human telomerase reverse transcriptase (hTERT)-immortalized adipose-derived mesenchymal stem cell line ASC52telo was purchased from ATCC (SCRC-4000) and was cultured in MesenPRO RS Medium (Gibco, 12746-012) supplemented with L-glutamine (Sigma-Aldrich, G7513-100ML) at a final concentration of 2 mM. The *Drosophila* SG-4 cell line used for calibrated ChIP was provided by A. Feldmann (DKFZ) and maintained in Schneider's *Drosophila* medium (Thermo Fisher Scientific, 21720024) supplemented with 10% FBS and penicillin-streptomycin. The SMARCA2/SMARCA4 degrader ACBI1 was purchased from MedChemExpress (2375564-55-7), resuspended in dimethylsulfoxide (DMSO) and kept at −80 °C. Cells were treated for 72 h with 500 nM ACBI1.

### Plasmid cloning

All constructs cloned in this study can be found in Supplementary Table 3.

MBD-V5 constructs were cloned into pLV-EF1a-IRES-Neo (Addgene, 85139). Luciferase was amplified by PCR from pT3-EF1a-NrasG12V-GFP-P2A-Luc2 (a gift from S.W. Lowe's laboratory), KDM2B was amplified from pUC19-hKDM2B (Sino Biological, HG20918-U), and the ZF-CxxC mutant was generated with PCR using mismatched primers (Q5). The MBD sequence was amplified using pENTR-MBD1 (ref. 65) (Addgene, 47057) as a template. The assembly was designed and performed in a single step adding the MBD, complementary DNAs and V5-NLS using NEBuilder HiFi DNA Assembly.

sgRNAs for CRISPR knockout were designed using the tool from Sanjana Lab (http://guides.sanjanalab.org) and cloned as previously described[27,66] (see Supplementary Table 4 for sgRNA sequences). In brief, sgRNAs were cloned by annealing two DNA oligos and ligating into a BsmB1-digested pLKO1-puro-U6-sgRNA-eGFP. Transformation was carried into Stbl3 bacteria.

For the SSX fusion vectors, cDNA of EWSR1, MN1 and NEDD4 were obtained from the DKFZ cDNA clone repository and assembled with an HA tag at the amino terminus and SSX at the C terminus into a MSCV-PGK-Puro backbone in a single step using NEBuilder HiFi DNA Assembly (New England Biolabs, E2621). EWSR1-FLI1 cDNA was a gift from T. Grünewald.

eGFP-fused constructs were cloned into pLV-EF1a-IRES-Neo lentiviral backbone[67] (Addgene, 85139) containing a neomycin selection cassette. cDNAs were adapted from the MSCV-HA-PGK-Puro plasmids[7].

NanoBRET plasmids pHTN-HaloTag-CMV-neo (Promega, G7721) and pNLF1-N-CMV-Hygro (Promega, N1351) were obtained from Promega. Histone H2A cDNAs were amplified by PCR from pCDNA3.1-Flag-H2A and pCDNA3.1-Flag-H2A K118-119R (ref. 38) (Addgene, 63560 and 63564).

### Virus production and transduction

For lentivirus production, $1 \times 10^6$ HEK293T cells were transfected with 3 μg of constructs and helper vectors (2.5 μg of psPAX2 and 0.9 μg of VSV-G). For retroviral infection, $10 \times 10^6$ HEK293GP cells containing a gag-pol insertion were transduced with 20 μg of MSCV vectors and 2.5 μg of VSV-G. Transfection of packaging cells was performed using polyethyleneimine (Polysciences, 23966-2) by mixing with DNA in a 3:1 ratio. Viral supernatants were collected 48 h after transfection, filtered through a 0.45-μm filter and supplemented with 4 μg ml⁻¹ polybrene

(Sigma) before adding to target cells. Downstream experiments using sgRNAs for knockouts were performed 10 days after sgRNA induction (CUT&RUN) or 12 days after knockout (immunofluorecence). Downstream experiments using overexpression of eGFP or MBD constructs (salt extraction, imaging, nuclear co-immunoprecipitation for mass spectrometry, and RNA sequencing (RNA-seq)) were performed 3–7 days after induction and will be specified for each technique. Downstream experiments using shSS18-SSX knockout were performed 3 days after doxycycline induction (CUT&RUN and western blots).

### Generation of Cas9 stable cell lines

For stable expression, HS-SY-II and SYO-1 synovial sarcoma cell lines were transduced with lentiCas9-Blast (ref. 66) (Addgene, 52962) and selected using 20 μg ml⁻¹ blasticidin to generate stable Cas9-expressing cell lines. Cells were subsequently transduced with sgRNAs. After 3 days of infection, cells were selected with 2 μg ml⁻¹ puromycin.

### Whole cell protein extracts and western blotting

Cells grown in 6-well plates were collected and washed in PBS. Cell pellets were incubated with RIPA buffer (Cell Signaling Technology) supplemented with protease inhibitors (Roche) for 30 min and cleared by centrifugation (15 min, >21,000*g*, 4 °C). Protein lysates were quantified using a BCA Protein Assay (Pierce). Lysates were then denatured in 2x Laemmli at 95 °C for 5 min, then run in Mini-PROTEAN Precast Gels (Bio-Rad) and transferred onto membranes using Trans-Blot Turbo. Membranes then were blocked in 5% milk in TBST. Western blots were visualized using an Amersham Imager 680.

### Immunofluorescence staining

Between $0.5 \times 10^6$ and $1 \times 10^6$ cells were seeded 6 days after induction in 6-well plates containing coverslips. Cells were fixed the following day with 4% paraformaldehyde for 10 min. Permeabilization was performed using Triton X (0.1% in PBS) for 12 min, followed by incubation with blocking solution (1% BSA, 0.1% gelatin fish in PBS) for 1 h. Incubation with the primary antibody was performed in blocking buffer at room temperature (20–22 °C) for 1 h. Cells were washed, incubated with secondary antibodies for 1 h, and mounted in VECTASHIELD Antifade Mounting Medium containing 4′,6-diamidino-2-phenylindole (DAPI; Vector Laboratories). For four-color immunofluorescence using V5-555 antibody (Invitrogen), after the secondary antibody, cells were washed and incubated with V5-555 for 1 h prior to mounting. Antibodies used are listed in Supplementary Table 5.

### Image capture and processing

Confocal images were acquired on a Leica TCS SP5 inverted confocal microscope using an HCX PL APO 63x/1.40-0.60 Oil Lbd BL objective, and a single z-stack was captured. Samples were imaged using 405, 488, 561, 594 and 633 nm laser lines using sequential mode in the Leica Application Suite software. For illustration, samples were imaged using a 512×512 format at a speed of 100 Hz using line averaging at 4 with a zoom factor of 11 for a single nucleus or 5 when showing three or four nuclei. Images were then smoothed and adjusted for brightness and contrast using the ImageJ/Fiji software.

### MBD assay quantification

Images were acquired using a 512×512 format at a speed of 700 Hz with a zoom factor of 1.7. Between 50 and 100 foci were counted per replicate, each MBD focus was selected, and only co-occurring foci were counted.

### Calibrated native ChIP for H2AK119ub1

The protocol for calibrated ChIP sequencing was modified from ref. 8. In brief, human synovial sarcoma cell line HS-SY-II was used to determine the change in H2AK119ub1 status when *PCGF1* was knocked out. The experiment was performed in biological triplicate. *Drosophila* cell

line SG-4 was used as the spike-in cell line. HS-SY-II cells were transduced with either the empty vector plasmid or with sgRNA targeting the *PCGF1* gene. The cells were cultured for 10 days, and then $10 \times 10^6$ cells were collected; $2 \times 10^6$ SG-4 cells were mixed with the collected cells. The cells were washed with ice-cold lysis buffer (10 mM Tris-HCl at pH 8, 10 mM NaCl, 3 mM $MgCl_2$, 0.1% NP-40, 5 mM sodium butyrate, 5 mM $N$-ethylmaleimide) to extract the nuclei. The nuclei were then digested using 100 U of MNase (Fermentas, EN0191) at 37 °C for 5 min in MNase digestion buffer (10 mM Tris-HCl at pH 8.0, 10 mM NaCl, 3 mM $MgCl_2$, 0.1% NP-40, 0.25 M sucrose, 3 mM $CaCl_2$, 10 mM sodium butyrate, 10 mM $N$-ethylmaleimide, 1× PIC (Roche)), followed by an addition of 4 mM EDTA to stop the digestion. All the buffers were supplemented with protease inhibitor and NEM (inhibitor of deubiquitinase enzymes). After centrifugation, the supernatant was retained and incubated at 4 °C overnight with 5 μl of anti-H2AK119ub1 (Cell Signaling Technology, D27C4). Next, 30 μl of Protein A/G Magnetic Beads (Thermo Fisher) were added for the pull down and incubated at 4 °C for 2 h. To elute the DNA, beads were incubated in 1% SDS, 0.1 M NaHCO₃ at 24 °C for 30 min. DNA was purified using a ChIP DNA Clean & Concentrator kit (Zymo Research).

## CUT&RUN

Chromatin profiles of endogenous SS18-SSX1/2, H2AK119ub1 in human synovial sarcoma cells and HA occupancy in osteosarcoma or synovial sarcoma cells expressing MSCV-HA-eGFP-SS18-PGK-Puro, MSCV-HA-eGFP-SS18-SSX1-PGK-Puro and MSCV-HA-eGFP-SSX-C-PGK-Puro were assayed using a CUTANA ChIC/CUT&RUN Kit (EpiCypher, 14-1048) following the manufacturer's protocol. In brief, 1 million cells (HS-SY-II, SYO-I or KHOS-240S) were collected per sample and bound to activated Concanavalin A Magnetic Beads. Beads were then incubated at 4 °C overnight with 1:50 dilution of antibodies per sample. Chromatin digestion was performed at 4 °C for 2 h. Digestion was then stopped by chelating $Ca^{2+}$ ions in a buffer containing *Escherichia coli* DNA for spike-in. DNA fragments were then released in solution after incubation at 37 °C for 10 min on a ThermoMixer at 500 r.p.m. DNA fragments were then purified using a CUTANA DNA Purification Kit (EpiCypher).

## Crosslinked ChIP

Chromatin occupancy of HS-SY-II cells expressing MSCV-HA-eGFP-PGK-Puro, MSCV-HA-eGFP-SSX-C-PGK-Puro or MSCV-HA-eGFP-SSX-C$^{\Delta RD}$-PGK-Puro was performed following selection with 2 μg ml$^{-1}$ puromycin and collected 6 days following transduction. HS-SY-II cells were prefixed for 20 min with 1.5 mM ethylene glycol bis(succinimidyl succinate) (Thermo Scientific) and then fixed with 1% formaldehyde for 15 min; the crosslinking reaction was stopped by adding 125 mM glycine. Cells were washed twice with cold PBS and lysed in swelling buffer (150 mM NaCl, 1% v/v Nonidet P-40, 0.5% w/v deoxycholate, 0.1% w/v SDS, 50 mM Tris pH 8, 5 mM EDTA) supplemented with protease inhibitors. Cell lysates were sonicated using a Covaris E220 sonicator to generate fragments less than 400 base pairs (bp). Sonicated lysates were centrifuged and incubated at 4 °C overnight with HA tag (Abcam, 9110). Immunocomplexes were recovered by incubation with 30 μl of Protein A/G Magnetic Beads at 4 °C for 2 h. Beads were sequentially washed twice with RIPA buffer and finally TE buffer.

## Library preparation

DNA fragments obtained after ChIP or CUT&RUN were quantified using a Qubit dsDNA HS Assay Kit (Invitrogen). Five nanograms of DNA were used for library preparation using a NEBNext Ultra II DNA Library Prep Kit for Illumina (New England Biolabs, E7645S), SPRIselect beads (Beckman Coulter, B23317) and NEBNext Multiplex Oligos for Illumina (New England Biolabs, Set 1 E7335S and Set 2 E7500S). ChIP libraries were prepared following New England Biolab's guidelines (New England Biolabs, E7645S), and CUT&RUN libraries were prepared following

the CUTANA ChIC/CUT&RUN Kit adapted protocol. Both libraries were done without size selection with an input of 5 ng. ChIP libraries were sequenced as 75 bp single-read on an Illumina NextSeq 550 platform on High-Output. SS18-SSX and H2AK119ub1 CUT&RUN libraries were sequenced as 75 bp paired-end reads on the Illumina NextSeq 550 platform on Mid-Output. HA-eGFP-SS18, HA-eGFP-SS18-SSX1 and HA-eGFP-SSX-C CUT&RUN libraries and native H2AK119ub1 calibrated ChIP were sequenced as 50 bp paired-end reads on a NovaSeq 6000 SP.

## Calibrated native H2AK119ub1 ChIP analysis

Sequenced reads were mapped using Bowtie 2 to the human genome build T2T-CHM13 using options–local–very-sensitive-local, and to the dm6 genome (*Drosophila*). PCR duplicates were removed using the Rmdup tool. Downsampling of reads for each sample was done based on the formula from ref. [8]:

$$\text{Downsampling factor}: \ \alpha \times 1/N(\text{ChIP SpikeIn})$$
$$\times N(\text{Input SpikeIn})/N(\text{Input HSSY})$$

where $\alpha$ is a coefficient applied for all of the files normalized together so that the value of the largest downsampling factor equals 1. $N$(ChIP SpikeIn) is the total number of reads aligned to the dm6 in the immunoprecipitation sample; $N$(Input SpikeIn) is the total number of reads aligned to dm6 in the corresponding Input; $N$(Input HSSY) is the total number of reads aligned to the T2T genome in the corresponding Input sample. The downsampled replicates were then combined using the pileup function from MACS2 ($q$-value, 0.05), and bigWig files were generated with the ucsc-wigtobigwig tool. Data are shown in Fig. [1].

## CUT&RUN analysis

Paired-end reads were aligned to the newly released human genome build T2T-CHM13 and *E. coli* K12, MG1655 reference genome using Bowtie 2 (with options for T2T-CHM13: –local–very-sensitive-local–no-unal–no-mixed–no-discordant–phred33-I10 -X 700; and for K12: –end-to-end–very-sensitive–no-overlap–no-dovetail–no-mixed–no-discordant–phred33-I10 -X 700). To internally calibrate our CUT&RUN experiments, we used the exogenous *E. coli* genome to quantitatively compare the genomic profiles as previously described[68]. We first calculated the percentage of spike-in reads in total reads that aligned uniquely ($N_x$). We then normalized the samples using a scaling factor so that the *E. coli* spike-in signal was set to be equal across all samples. We used the sample displaying the smallest percentage of *E. coli* reads ($N_{min}$) to downscale all other conditions using the same constant to calculate our scaling factor:

$$\text{Scaling factor for sample } x = N_{min}/N_x$$

Genome coverage files were generated using bamCoverage[69] with 50 bp bins, no normalization and scaled (–scaleFactor). For H2AK119ub1 ($n = 11,099$) and SS18-SSX2 ($n = 27,686$) peak calling, the MACS2 callpeak function was used on the aligned BAM files and IgG was used as control (with '–nomodel,' '–qvalue 0.01,' '–broad' options, '–keep-dup all'). For HA peak calling in KHOS-240S, HA-SS18, HA-SS18-SSX1 and HA-eGFP-SSX1 were combined in MACS2 to compute all of the HA peaks ($n = 58,843$). A heatmap of Spearman correlation coefficients was generated using deepTools multiBigwigSummary and plotCorrelation[69]. H2AK119ub1 CUT&RUN is shown in Extended Data Fig. 1.

## ChIP–seq analysis

SS18-SSX1 (HA) input (SRR6451607), SS18-SSX1 (HA) immunoprecipitation (SRR6451595), KDM2B input (SRR6451587) and KDM2B immunoprecipitation (SRR6451586) were obtained from the Gene Expression Omnibus (GEO) under accession number GSE108926. Raw reads were trimmed for quality and Illumina adapter sequences using trim-galore, then aligned to the human genome assembly hg38

using Bowtie 2 (refs. [70],[71]) (with the '–very-sensitive' option). ChIP signals were normalized to their respective inputs using the pileup function from MACS2 (refs. [72],[73]) using corresponding input for background normalization. To visualize ChIP–seq tracks, normalized bigWig files were generated with the ucsc-wigtobigwig tool. HA-SS18-SSX1 peaks ($n = 26,805$) were generated with the MACS2 function (with '–nomodel,' '–qvalue 0.05,' '–broad' options) and normalized to input.

## ChIP and CUT&RUN data visualization
Genome tracks were visualized using UCSC Genome Browser (https://genome.ucsc.edu). For heatmaps and metaplot profiles, read densities of the various immunoprecipitations were centered around peak signals with a ±10-kilobase (kb) window from peak center and binned with 50 bp using the computeMatrix and plotProfile/plotHeatmap functions from deepTools (ref. [69]).

## Tracking of indels by decomposition (TIDE) analysis
Genomic DNA was extracted from Cas9 infected cells expressing sgRNA and parental cells using a DNeasy Blood & Tissue Kit (QIAGEN) following the manufacturer's protocol. The region targeted with sgRNA was amplified using the relevant primers (Supplementary Table 4) and purified using a PCR purification kit (QIAGEN). Following Sanger sequencing of the PCR amplicons, sequences were analyzed using the TIDE website (http://shinyapps.datacurators.nl/tide) to calculate the percentage of insertions and deletions and assess sgRNA efficiency[74].

## Cell competition assays
HS-SY-II and KHOS-240S Cas9 cells were transduced with an empty plasmid (empty vector) or a plasmid containing sgRNA targeting *PCGF1*. Infections were done with a virus dilution of 1:10 to obtain an infection efficiency of around 70–80%. Infected cells become GFP$^+$ due to the backbone of the sgRNA. The cells were then cultured over a period of 25 days, and the percentage of GFP$^+$ cells was measured using a Fortessa fluorescence-activated cell sorting (FACS) machine. Data were analyzed using FlowJo software.

## Chromatin salt extraction and sequential chromatin washes
Chromatin salt extraction was adapted from ref. [75]. Approximately $10 × 10^6$ cells were collected and washed twice in PBS. Cell pellets were then washed in a series of chromatin salt extraction buffers containing 0.1% Triton X, 300 mM sucrose, 1 mM MgCl$_2$, 1 mM EGTA, 10 mM PIPES and NaCl at increasing concentrations: 80 mM, 150 mM, 300 mM and 500 mM. All buffers were supplemented with protease inhibitors (Protease Inhibitor Tablets, Roche). Cell pellets were resuspended and incubated in 50 µl of chromatin salt extraction buffer at room temperature for 10 min and pelleted at $2,000 × g$ for 5 min. The supernatant was transferred to a new tube and supplemented with 2x Laemmli (Invitrogen) and kept on ice after denaturation at 95 °C for 5 min. For the chromatin extraction after the last 500 mM wash, pellets were resuspended in 500 mM NaCl chromatin salt extraction buffer supplemented with 2x Laemmli. The chromatin sample was then denatured at 95 °C for 5 min and sonicated. Chromatin samples were then centrifuged at full speed for 5 min to get rid of the DNA debris and transferred to a new tube. Sequential chromatin washes were performed similarly, but the cells and the chromatin were washed at a constant salt concentration of 150 mM for four washes; the chromatin fraction was then sonicated as above. Samples were then used for western blotting. The signal intensity in the various salt fractions was measured using the maximum intensity of a square containing the band in the ImageJ/Fiji software. The total protein level was calculated using the sum of the maximum intensity as a proxy. Each intensity or salt fraction was then represented as a percentage of total protein levels.

## CRISPR–Cas9 gene-tiling screen
sgRNA library cloning and screen deconvolution were performed as previously described[76],[77]. In brief, sgRNAs targeting the entire coding

sequence of SS18 and SSX1 were designed using Benchling (https://benchling.com) and cloned into pLKO-U6-sgRNA-improved-EF1s-GFP-P2A (gifted by D.F. Tschaharganeh). A total of 211 sgRNAs were designed spanning the length of isoform 1 of SS18 (NT 010966), and 90 sgRNAs targeting isoform 1 of SSX1 (NT 011568). Additionally, 200 safe sgRNAs were added as negative controls; these guides target the nongenic region of genome[78]. Stable Cas9-expressing cell lines were transduced to about 30% efficiency. After 3 days of infection, cells were selected with 2 µg ml$^{-1}$ puromycin. Cells were passaged with the number of cells kept at 3,000 times the size of the library, that is, at least $1.56 × 10^6$ cells were passaged. After 15 population doublings, the cells were collected and their genomic DNA was extracted using the phenol extraction method. The region spanning the sgRNA was amplified using custom primers. Amplicons were sent for next generation sequencing using NextSeq 550 SR 75 HO. Files were demultiplexed, and counts were mapped on the library using the MAGeCK tool. To identify individual regions that are more important for cell survival, we used ProTiler to identify CKHS regions.

## Live imaging
Approximately 30,000 HEK293T cells transduced with the various eGFP constructs were seeded in an 8-well chamber slide (µ-Slide 8 Well high, ibidi). Cells were then imaged within the next 48 h using the Leica TCS SP5 inverted confocal microscope with the HCX PL APO 63x/1.40-0.60 Oil Lbd BL objective; a single z-stack was captured. DNA was stained 30 min prior to image acquisition using NucBlue Live ReadyProbes Reagent (Hoechst 33342) (Invitrogen).

## Nuclear immunoprecipitation
Approximately $5 × 10^7$ cells were collected and washed twice in PBS. Nuclei isolation, nuclear fraction digestion and collection were performed using a Nuclear Complex Co-IP Kit (Active Motif, 54001). For classical immunoprecipitation and immunoprecipitation submitted to mass spectrometry analysis, chromatin shearing was performed on ice for 90 min. For harsher conditions used in Fig. 3, chromatin shearing was performed at 37 °C for 10 min. Twenty-five microliters per immunoprecipitation of GFP-Trap Magnetic Agarose beads (ChromoTek) were washed twice in 1X IP Low Buffer supplemented with protease inhibitor and PMSF following the manufacturer's guidelines (Active Motif, 37511). Two hundred microliters of nuclear extracts were incubated with the GFP-Trap beads at 4 °C for 1 h. Beads were then washed three times in 1X IP Low Buffer and resuspended in 50 µl of 2x Laemmli, then boiled at 95 °C for 10 min.

## Mass spectrometry
Following the final wash of nuclear immunoprecipitation, beads were resuspended in 100 µl of 1% SDS. Beads were denatured at 95 °C for 5 min, and the supernatant was submitted for mass spectrometry at the EMBL Proteomics Core Facility. Data analysis was performed by the Facility. The raw output files of IsobarQuant (protein.txt – files) were processed using the R programming language. Only proteins that were quantified with at least two unique peptides were considered for the analysis. Raw signal-sums (signal_sum columns) were first cleaned for batch effects using limma (ref. [79]) and further normalized using variance stabilization normalization[80]. Different normalization coefficients were estimated for control conditions in order to maintain the lower observed abundance.

## Histone acid extraction
Approximately $1 × 10^6$ cells were collected and washed twice in PBS. Cells were resuspended in 100 µl of PBS + 0.5% Triton X and incubated on ice for 10 min. After centrifugation at $6,500g$ at 4 °C for 10 min, nuclei were washed a second time in 100 µl of PBS + 0.5% Triton X. Nuclear pellets were then resuspended in 25 µl of 0.2 N HCl. Histones were released overnight at 4 °C, and DNA debris was pelleted at

6,500 × $g$ at 4 °C for 10 min. Histone acid extracts were neutralized with 2.5 µl of 2 M NaOH. After 2x Laemmli addition and denaturation at 95 °C for 5 min, samples were loaded onto a western blot gel.

## NanoBRET

NanoBRET Protein:Protein Interaction assay was performed following the manufacturer's conditions (Promega, N1662). Approximately 0.5×10^6 HEK293T cells were plated the day before transfection in a 12-well plate. Two micrograms of HaloTag plasmid (empty, SS18, SS18-SSX, SSX-C, SSX-C^ΔRD or SSX-C^E184*) + 0.2 µg of NanoLuc plasmid (H2A WT, H2A^K118K119R) were transfected using polyethylenimine by mixing with DNA in a 3:1 ratio. Forty-eight hours after transfection, cells were counted and adjusted to a final concentration 2 × 10^6 cells per ml. Cells were passed in a 96-well white plate. For each condition, 10 µl (20,000 cells) were seeded in four different wells. Each well was supplemented with 90 µl of Opti-MEM I Reduced Serum Medium, no phenol red (Gibco, 11058-021) containing 4% FBS with either 100 nM HaloTag NanoBRET 618 Ligand (+ligand, experimental samples in two technical replicates) or 0.1% DMSO final concentration (−ligand, no-acceptor controls in two technical replicates). The next day, 72 h after transfection, 25 µl of 5x NanoBRET Nano-Glo in Opti-MEM I Reduced Serum Medium was added on all of the wells. Measurements of NanoBRET bioluminescent donor emission (460 nm) and acceptor emission (618 nm) were performed within 10 min of substrate addition using a PHERAstar Microplate Reader (BMG Labtech) with 450 nm and 620 nm filters. NanoBRET calculations were done using the followings steps. The raw NanoBRET ratio (BU) was obtained by dividing the acceptor emission value (620 nm) by the donor emission value (450 nm) for each sample. BU values were then converted to milliBRET units (mBU) by multiplying each raw BRET value by 1,000. The final BRET ratio (mBU) displayed in the figures is calculated for each biological replicate by subtracting the mean of the two experimental replicates (+ligand) with the mean of the two no-ligand control replicates (−ligand).

## RNA extraction and qPCR

RNA was prepared using an RNeasy Mini Kit (QIAGEN) according to the manufacturer's protocol and including the DNase I (QIAGEN) treatment. cDNA was synthesized from purified RNA with RevertAid Reverse Transcriptase (Thermo Scientific) primed with random hexamers. qPCR was carried on the Roche LightCycler 480 Real-Time PCR System using Power SYBR Green PCR Master Mix (Applied Biosystems). The real-time thermal cycler was programmed as follows: 15 min Hotstart; 44 PCR cycles (95 °C for 15 s, 55 °C for 30 s, 72 °C for 30 s). Primers are listed in Supplementary Table 4.

## RNA-seq analysis

RNA libraries were prepared at the DKFZ Genomics and Proteomics Core Facility and was sequenced on a NovaSeq 6000 Paired-End 100 S4. RNA-seq reads were aligned to the human genome assembly hg19, and a fragments per kilobase million (FPKM) count matrix was generated using featureCounts (ref. 81). Data analysis of replicate clustering (principal component analysis), heatmaps of the 5,000 most variable genes, and differential expression analysis were performed using iDEP (http://bioinformatics.sdstate.edu/idep93)[82].

## eGFP-based imaging quantification of protein levels

To directly compare the staining signal of H2AK119ub1, BCOR or SS18, we took advantage of the eGFP reporter as a proxy for construct expression (Extended Data Fig. 6). We used side-by-side comparison using GFP-negative (or very low expressing cells) versus GFP-positive cells. The calculation was done using ImageJ/Fiji software by first isolating the nuclei using Li thresholding on their DAPI signal and region of interest selection. Next, we measured the signal intensity for each channel: DAPI (405 nm), eGFP (488 nm), and either red channel (594 nm) and/or far read (647 nm) when applicable and kept the mean. Then, for each nucleus, the signal intensities were normalized to DAPI, which should be constant across different nuclei. eGFP-high and eGFP-low (or negative) cell populations were distinguished based on a threshold of normalized eGFP (488 nm) intensity of >1 or <1, respectively. Finally, the ratio of the high versus low was used to display the change in signal intensity in the high-eGFP population (average of the corrected mean intensity for high eGFP / average of the corrected mean intensity for low eGFP). For each biological replicate, between 50 and 250 nuclei were analyzed.

## Human testis immunohistochemical imaging

For immunohistochemical analyses, formalin-fixed and paraffin-embedded tissue samples of non-neoplastic human testis were retrieved from the archives of the Institute of Pathology, University Hospital Heidelberg. Use of patient samples was approved by the ethics committee of the University of Heidelberg (S-442/2020). Sections of 4-µm thickness were cut and mounted on SuperFrost Plus Adhesion Slides (Thermo Scientific), followed by deparaffinization and heat-induced antigen retrieval (97 °C) in high pH buffer (pH 9) for 30 min. Primary monoclonal mouse antibodies for inhibin-α (ready-to-use, clone R1, Dako Omnis, Agilent), SSX (dilution 1:100) and H2A119ub1 (dilution 1:500) listed in Supplementary Table 5 were each incubated for 25 min. Visualization was performed using the ready-to-use POLYVIEW PLUS HRP (anti-mouse) reagent (Enzo Life Sciences). Sections were counterstained with hematoxylin.

## Human single cell testis atlas

t-distributed stochastic neighbor embedding (t-SNE) plots were obtained from the Human Testis Atlas Browser by Cairns Lab[83] (https://humantestisatlas.shinyapps.io/humantestisatlas1/). Data were acquired on young adults aged 17, 24 and 25 years.

## Mouse model for conditional SS18-SSX2 expression

The mouse model of synovial sarcoma used herein is based on the hSS2 model with a conditional SS18-IRES-eGFP allele knocked into the Rosa26 locus in a C57BL/6J background[50,51]. Mice were housed under standard conditions (12 h/12 h light/dark cycle) and provided food and water ad libitum. Animals were maintained in a controlled environment of 21–24 °C and 40–60% humidity, and experimental protocols were conducted in accordance with approved and ethical treatment standards of the Animal Care Committee at the University of British Columbia.

**Tissue processing and staining.** To enable detection of native eGFP expression in processed tissue samples, mice at clinical endpoint were humanely euthanized by intraperitoneal injection of Avertin (400 mg per kg (body weight)), and the tongues (containing tumor) were removed. Wild-type tongue samples were obtained from age-matched Cre-negative control animals. Dissected tongues were immersed in 2% paraformaldehyde fixative at 4 °C for 48 h. Samples were then washed three times for 30 min each in PBS and then immersed through a gradient of sucrose solutions from 10% to 50% at 4 °C for >4 h each before being embedded in cryomolds (Polysciences, 18646A) using OCT (Sakura Finetek, 4583) and frozen in an isopentane bath cooled by liquid nitrogen. Cryosections were cut (Leica, CM3050S) at a thickness of 20 µm and mounted onto Superfrost Plus slides (VWR, 48311-703). Slides were thawed at 37 °C for 30 min, washed three times for 10 min each in PBS and incubated for 1 h in PBS containing 10 mg ml^−1 sodium borohydride (Sigma, 213462) to quench autofluorescence. Following this treatment, slides were briefly washed with PBS and incubated in block solution containing 2.5% BSA (Sigma, A7030) and 2.5% goat serum (Gemini, 100-190) at room temperature for 90 min prior to incubation in primary antibody dissolved in block solution (1:100) at 4 °C overnight. Primary antibody solution was removed, and slides were washed three times for 5 min each in PBS before Alexa Fluor-conjugated

secondary antibodies were applied to the slides for 45 min. After secondary antibody incubation, three 5-min PBS washes were performed and sections were counterstained with DAPI (600 nM in PBS) for 5 min, then rinsed and mounted with Aqua-Poly/Mount (Polysciences, 18606).

**Image acquisition and quantification.** Confocal images were collected using a Nikon Ti-E inverted microscope with an A1R HD25 confocal scanning head and acquired in Nikon Elements software. For quantification, a single z-stack was selected and the image was first smoothed. Nuclei were detected using Li thresholding in ImageJ/Fiji software. Signal intensity for each selected nucleus was measured for the channels 405-DAPI, 488-eGFP (SSM2 cassette) and 647-H2AK119ub1. The ratio intensity of H2AK119ub1 over DAPI was calculated by dividing the 647 mean signal intensity over its corresponding 405 mean signal intensity.

## Human synovial sarcoma tissue microarray immunohistochemical imaging

Tissue microarray (TMA) construction from anonymized patient primary surgical excision specimens was performed under protocols H18-00524 and H18-02391, approved by the Clinical Research Ethics Board of the University of British Columbia and BC Cancer. H2AK119ub1 and SS18-SSX immunohistochemistry was performed on a 4-μm section of a formalin-fixed, paraffin-embedded human TMA consisting of 37 synovial sarcoma cases; one case each of epithelioid sarcoma, sarcomatoid mesothelioma, Ewing sarcoma, sarcomatoid renal cell carcinoma, clear cell sarcoma, dedifferentiated liposarcoma and myxoid liposarcoma; as well as normal skeletal muscle, ovarian stroma, breast glandular tissue and testis controls from Vancouver General Hospital. Cases were included as 0.6 mm patient sample cores in duplicate. The assays were run with the following conditions via a Leica BOND RX (Leica Biosystems). Heat-induced epitope retrieval was performed using citrate-based BOND Epitope Retrieval Solution 1 (Leica Biosystems) for 10 min, 10 min and 20 min, respectively. The primary antibodies H2AK119ub1 (Cell Signaling Technology, 8240) and SSX-SS18 (Cell Signaling Technology, 72364S) were incubated at ambient temperature at 1:400 for 30 min and 1:300 for 15 min, respectively. Staining was visualized using the BOND Polymer Refine Detection kit (Leica Biosystems, DS9800), which includes a 3,3′-diaminobenzidine (DAB) chromogen and hematoxylin counterstain. TMA virtual slide scans were then generated on a Leica Aperio AT2 (Leica Biosystems) at ×40 magnification. Each individual patient sample core was analyzed using HALO and HALO AI (Indica Labs), which required user annotated training data to develop an artificial intelligence segmentation network for nuclear identification. The TMA module was implemented to extract individual patient core images from the TMA whole slide scan. The Multiplex IHC module was trained to identify DAB staining using representative pixels for delineation from hematoxylin in order to determine average DAB nuclear optical density.

## Statistics and reproducibility

Details of the individual statistical analyses and tests, as well as the number of biological replicates, can be found in the respective figure legends. Statistical analysis was performed using Microsoft Excel and GraphPad Prism software.

## Reporting summary

Further information on research design is available in the Nature Portfolio Reporting Summary linked to this article.

## Data availability

HA-SS18-SSX1 and KDM2B ChIP sequencing data reanalyzed in Fig. 1 originate from GEO accession number GSE108929. The GEO accession number for ChIP–seq, CUT&RUN-seq and RNA-seq data reported in this paper is GSE205955. Source data are provided with this paper.

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

## Acknowledgements

We thank W. He from H. Xu's laboratory for help in running ProTiler analysis for the *SS18-SSX1* gene-tiling screen. We also thank S.W. Lowe,

D.F. Tschaharganeh (University Hospital Heidelberg), J. Zuber (IMP, Vienna) and S. Henikoff (Fred Hutchinson Cancer Research, Seattle) for sharing reagents and protocols; A. Feldmann (DKFZ) for assistance in calibrated ChIP analysis and *Drosophila* SG-4 cells; D. Talwar from the T. Dick group at the DKFZ for assistance in measurements for the NanoBRET assays; and EMBL Mass Spectrometry Facility members M. Rettel and F. Stein for sample processing and data analysis. We also thank R. Illingworth and C. Playfoot for their useful comments and discussion regarding the manuscript and members of the paediatric soft tissue sarcoma lab and the U54 Synovial Sarcoma consortium for feedback and fruitful discussions. This project has received funding from the European Research Council (ERC) under the European Union's Horizon 2020 research and innovation programme (grant agreement no. 805338) (A.B.) and from the National Institutes of Health/National Cancer Institute (NIH/NCI) U54CA231652 (T.O.N., T.M.U. and A.B.). T.O.N. and T.M.U. were additionally supported by grants from the Canadian Cancer Society (705615) and the Terry Fox Research Institute (1082). N.S.B. was supported by a DKFZ Postdoctoral Fellowship. F.J.S.-R. was supported by the MSKCC TROT program (5T32CA160001) and a GMTEC Postdoctoral Researcher Innovation Grant, and is an HHMI Hanna Gray Fellow. The funders had no role in study design, data collection and analysis, decision to publish or preparation of the manuscript.

## Author contributions

N.S.B. conceived the study; designed, performed and analyzed the experiments; and wrote the manuscript. V.D. generated the Cas9 cell lines, conducted the CRISPR–Cas9 screen, conducted the competition assays, assisted with the CUT&RUN experiments and performed and analyzed the calibrated native H2AK119ub1 ChIP. R.W.S. and T.M.U. provided the mouse model data. F.K.F.K. performed immunohistochemistry analysis of human testis. F.M., A.S., A.P., L.G. and L.W. assisted in experiments and reagent production. F.J.S.-R. assisted with CRISPR–Cas9 library cloning and screen deconvolution. M.T., S.T. and T.O.N. provided the analysis for the synovial sarcoma TMAs. A.B. conceived and coordinated the study and wrote the manuscript. All authors read and approved the final manuscript for publication.

## Funding

## Competing interests

The authors declare no competing interests.

## Additional information

**Extended data** is available for this paper at https://doi.org/10.1038/s41594-023-01096-3.

**Correspondence and requests for materials** should be addressed to Ana Banito.

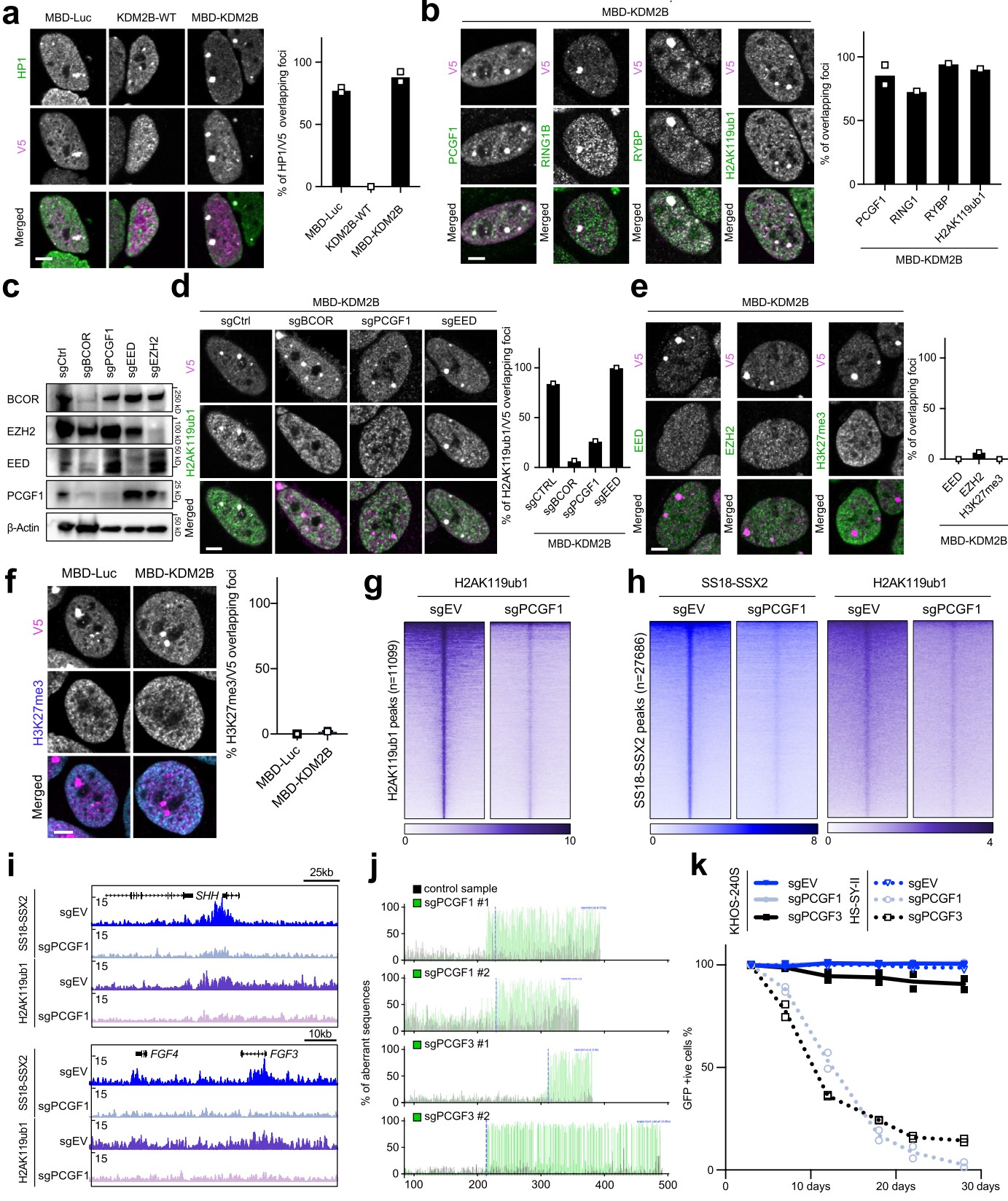

**Extended Data Fig. 1 | See next page for caption.**

**Extended Data Fig. 1 | PRC1.1 is suficient to initiate SS18-SSX recruitment.**
**a**) Left, Immunofluorescence for the MBD-Luc, MBD-KDM2B or for KDM2B-WT fused to a V5 tag (V5, magenta) and HP1 (green). Right, percentage of V5 foci overlapping HP1 foci. Data represents the mean of 2 biological replicates. Scale bars 5um throughout the figure. **b**) Left, Immunofluorescence of MBD-KDM2B (V5, magenta) with PCGF1, RING1B, RYBP and H2AK119ub1 (green). Right, percentage of foci overlapping a V5 foci in n = 2 (PCGF1, data represents the mean) or 1 biological replicate. **c**) Western Blot of HS-SY-II-Cas9 whole cell extracts expressing sgRNAs revealed using BCOR, EZH2, EED, PCGF1 or Beta-actin antibodies. **d**) Left, Immunofluorescence for MBD-KDM2B (V5, magenta) in the presence of different sgRNAs (resulting in eGFP background fluorescence) with H2AK119ub1 (green). Right, percentage of H2AK119ub1 foci overlapping V5 foci in one biological replicate. **e**) Left, Immunofluorescence of MBD-KDM2B (V5, magenta) with EED, EZH2 or H3K27me3 (green). Right, percentage of foci overlapping a V5 foci in one biological replicate. **f**) Left, Immunofluorescence for V5 (magenta) and H3K27me3 (cyan). Right, percentage of H3K27me3 foci overlapping with V5 foci in 2 biological replicates. Data represents the mean.

**g**) Heatmaps of H2AK119ub1 scaled CUT&RUN signals (purple) in HS-SY-II-Cas9 cells expressing empty sgRNA as control (sgEV) or targeting PCGF1 (sgPCGF1) over H2AK119ub1 peaks in HS-SY-II (n = 11099). Rows correspond to ±10-kb regions across the midpoint of each enriched region, ranked by increasing signal. **h**) Heatmaps of SS18-SSX2 (blue) or H2AK119ub1 (purple) scaled CUT&RUN signals in SYO-I-Cas9 cells expressing empty sgRNA as control (sgEV) or targeting PCGF1 (sgPCGF1) over SS18-SSX2 peaks in SYO-I (n = 27686). Rows correspond to ±10-kb regions across the midpoint of each enriched region, ranked by increasing signal. **i**) Gene tracks for H2AK119ub1 and SS18-SSX CUT&RUN signals in SYO-I-Cas9 at the *SHH* and *FGF4-FGF3* loci. **j**) Tracking of Indels by Decomposition (TIDE) assay displaying the percentage of aberrant sequences after Cas9 editing for 2 guides targeting PCGF1 and PCGF3 versus the wild-type sequence (control sample). **k**) Cell competition assay performed in the osteosarcoma cell line KHOS-240S-Cas9 (fusion negative control) or in the synovial sarcoma line HS-SY-II-Cas9 transduced with an empty sgRNA as control or with guides targeting PCGF1 and PCGF3.

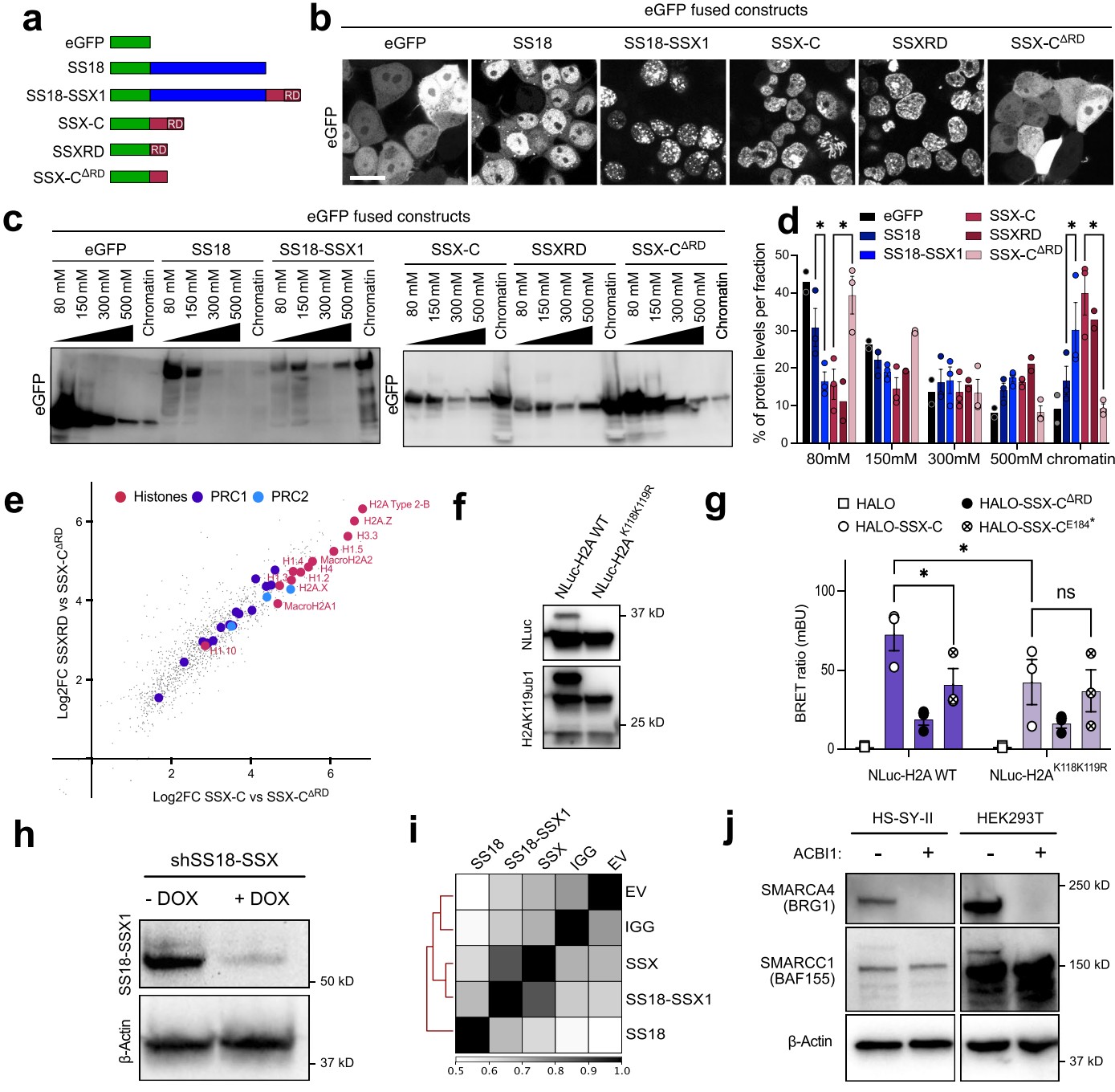

**Extended Data Fig. 2 | SSX-C binds chromatin via the SSRXD domain.**
**a)** Schematic of eGFP-fused constructs (green) for SS18, SS18-SSX1, SSX-C
(78aa of SSX1 present in the SS18-SSX1 fusion), SSXRD (last 34aa of SSX-C) or
SSX-CΔRD (SSX-C with a deletion of the SSXRD). **b)** Images representative of 2
independent live confocal imaging of the eGFP-fused constructs in HEK193T
cells. Scale bar 20um. **c)** Salt extraction assay displaying eGFP levels by western
blot in HEK293T expressing the various eGFP constructs. **d)** Percentage of total
eGFP-fused protein per salt extraction fractions. Data represents the mean
± S.E.M of n = 2 (eGFP, SSXRD) or n = 3 (SS18, SS18-SSX1, SSX-C and SSX-CΔRD)
biological replicates. Asterisks represent p-values of paired one-tailed t-test
between groups (from left to right, p = 0.02; p = 0.03; p = 0.03; p = 0.01). **e)** Log2
fold change correlation plot of eGFP-SSXRD and eGFP-SSX-C mass spectrometry
data following eGFP pull down in HS-SY-II cells. Data was normalized to eGFP-
SSX-CΔRD. **f)** Western blot of histone acid extracts from HEK293T cells transfected
with either Nluc-H2A or Nluc-H2A K118K119R revealed with NLuc, H2AK119ub1 and

H3 antibodies. Western was repeated for each replicate. **g)** BRET ratio (mBU) in
Nluc-H2A or Nluc-H2A K118K119R transfected HEK293T cells expressing empty vector
HALO, HALO-SSX-C, HALO-SSX-CΔRD or HALO-SSX-CE184*. Data represents the
mean ± S.E.M of n = 3 biological replicates. Asterisks represent p-values of paired
one-tailed t-test between groups (from left to right, p = 0.04; p = 0.01; ns=0.33).
**h)** Western Blot of whole cell extracts from HS-SY-II cells expressing shRNA
against SS18-SSX collected 72 h after no doxycycline (-DOX) or doxycycline
( + DOX) treatment. Blot revealed using SS18-SSX or Beta-actin antibodies. shRNA
knockdown was repeated in 3 independent experiments. **i)** Heatmap of Spearman
correlation coefficients from bigWig coverages computed over all HA peaks on
the KHOS-240S CUT&RUN. **j)** Western Blot of whole cell extracts from HS-SY-II
or HEK293T cells collected 72 h without or with 500 nM ACBI1 treatment. Blot
revealed using SMARCA4 (BRG1), SMARCC1 (BAF155) and Beta-actin antibodies.
Western blot was repeated in 2 independent experiments.

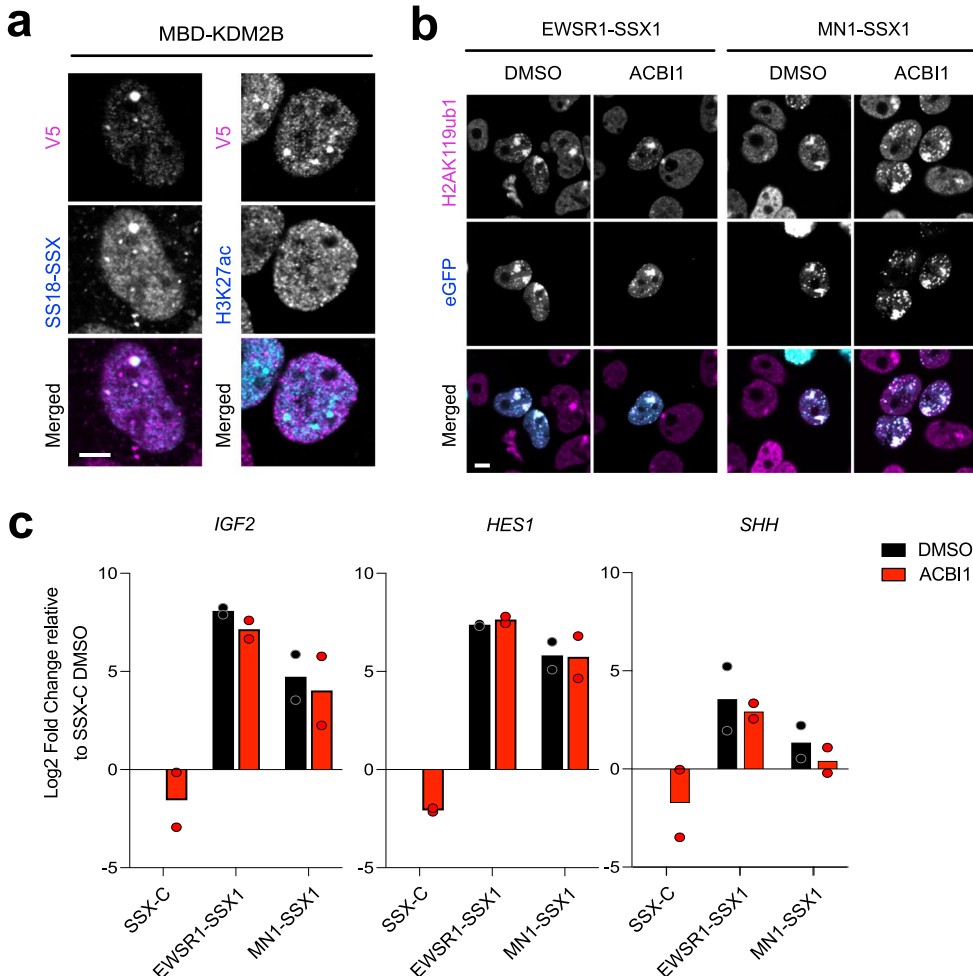

**Extended Data Fig. 3 | Alternative SSX fusions activate gene expression independently of BAF. a)** Immunofluorescence of MBD-KDM2B (V5, magenta) and H3K27ac (cyan) in HS-SY-II cells. Images are representative of 3 independent replicates. Scale bars indicate 5um throughout the figure. **b)** H2AK119ub1 immunofluorescence of HEK293T cells expressing eGFP-SS18-SSX1, eGFP- EWSR1-SSX1 or eGFP-MN1-SSX1 treated with DMSO (left) or 500 nM ACBI1 (right). Bottom panel displays merge channels with eGFP (cyan) and H2AK119ub1 (magenta). Images are representative of 1 replicate. **c)** qRT-PCR in hMSC expressing SSX-C, EWSR1-SSX1 or MN1-SSX 72 h after DMSO or 500 nM ACBI1 treatment. Data represents the mean of n = 2 biological replicates.

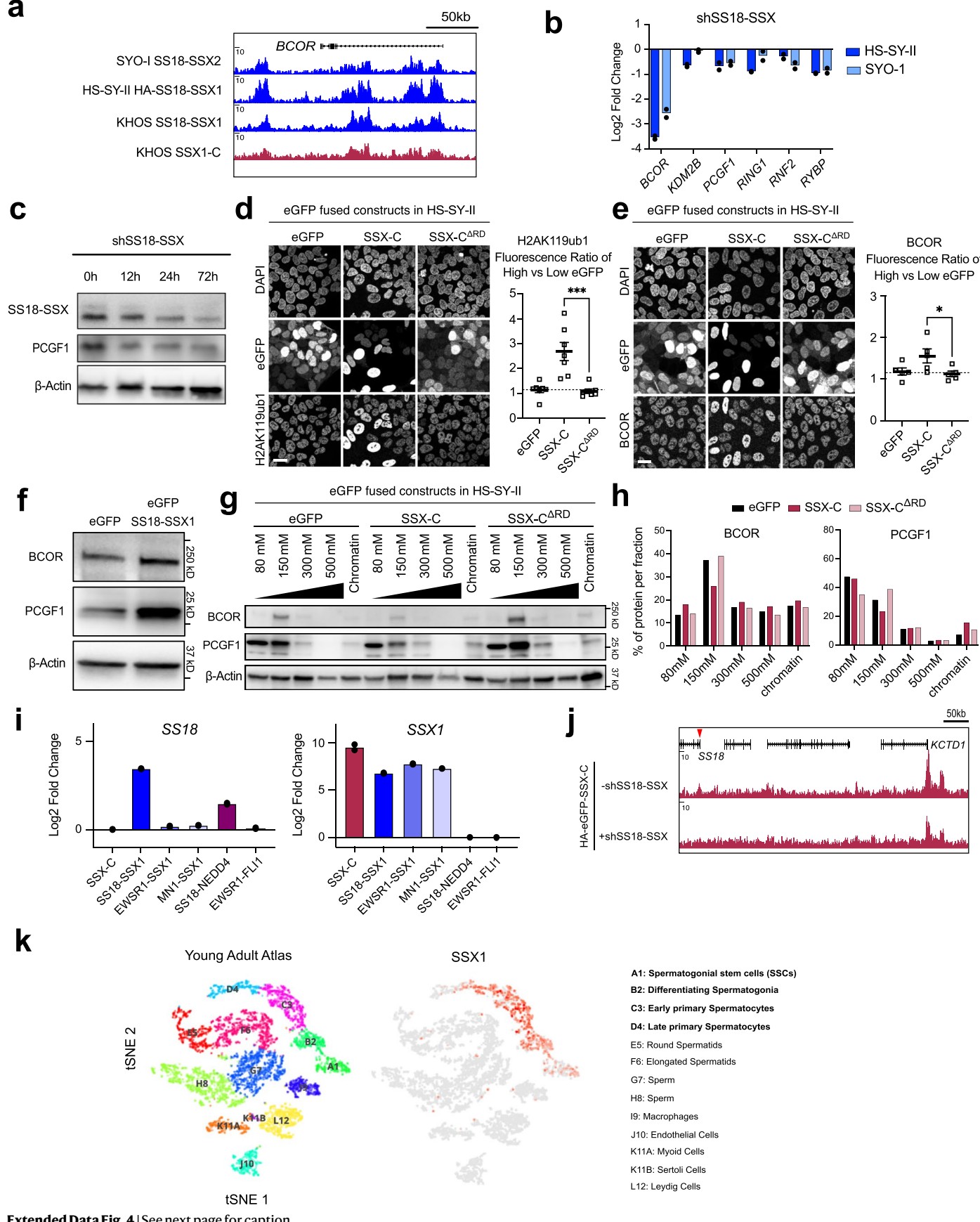

**Extended Data Fig. 4** | See next page for caption.

**Extended Data Fig. 4 | SSX-C enhances H2AK119ub1 by stabilizing PRC1.1 levels. a**) Gene tracks for SS18-SSX and SSX CUT&RUN at the *BCOR* locus. **b**) Log2 Fold change of RPKM values from RNA sequencing in HS-SY-II and SYO-I cells after knockdown of SS18-SSX compared to shCtrl cells. Data from McBride et al., 2018 represents the mean of two biological replicates. **c**) Western blot of whole cell extracts of HS-SY-II cells expressing shRNA against SS18-SSX over a time-course of 0 h to 72 h doxycycline induction. Blot represents one replicate. **d**) **e**) Left, Immunofluorescence against H2AK119ub1/BCOR in HS-SY-II expressing eGFP-fused constructs. Scale bars indicate 20um throughout the figure. Right, quantification of H2AK119ub1/BCOR fluorescence ratio in high versus low eGFP cells. Data represents the mean ± S.E.M of n = 7 (H2AK119ub1) or n = 5 (BCOR) biological replicates. Asterisks represent p-values of paired one-tailed t-test between groups (p = 0.0008 (H2AK119ub1), p = 0.03 (BCOR)). **f**) Western blot of whole cell extracts from hMSCs expressing eGFP or eGFP-SS18-SSX1. Blot represents one replicate. **g**) Salt extraction assay in HS-SY-II expressing eGFP, eGFP-SSX-C and eGFP-SSX-C$^{\Delta RD}$. Proteins were detected by western blot using with BCOR, PCGF1 or Beta-actin (loading control) antibodies. **h**) Quantification of the protein distribution in the various fractions of the salt extraction for BCOR or PCGF1. Data represents the percentage of total protein levels in one replicate. **i**) Log2 Fold change of FKPM values in mesenchymal stem cells (hMSCs) expressing the new fusion constructs and controls for *SS18* or *SSX1* mRNA levels. Data represents the mean of two biological replicates. **j**) CUT&RUN Gene tracks in HS-SY-II cells expressing HA-eGFP fused to SSX-C without and with SS18-SSX depletion mediated by shRNA over the *SS18* to the *KCTD1* loci. Red arrowhead marks the *SS18* promoter. **k**) Left, tSNE and clustering analysis of combined single-cell transcriptome data from human testes (n = 6490) from (Guo et al., 2018). Each dot represents a single cell and is colored according to its cluster identity as indicated on the figure key. The 13 cluster identities were assigned based on marker gene expression. Right, SSX1 expression pattern projected on the tSNE plot. Red indicates high expression and gray indicates low or no expression.

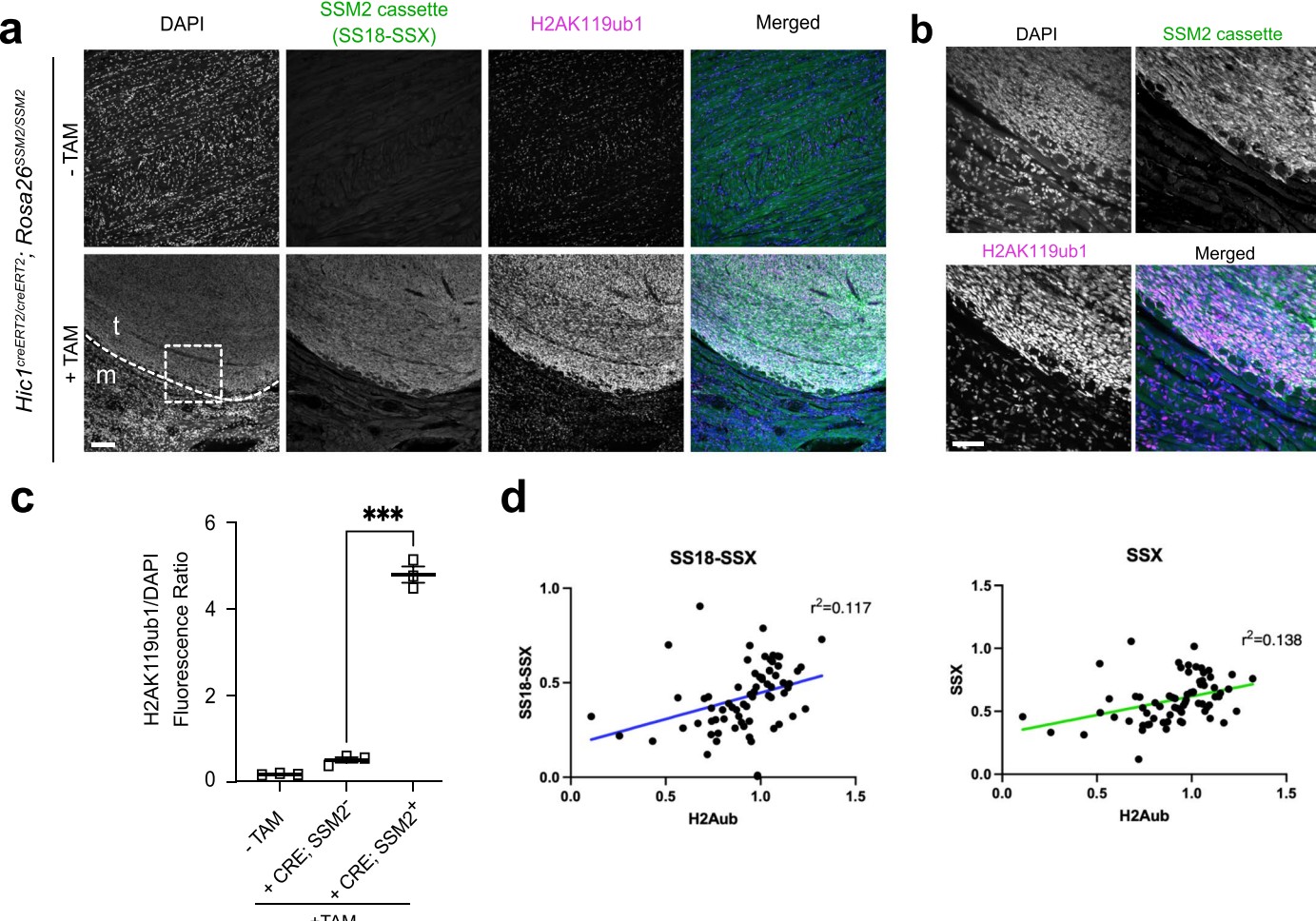

**Extended Data Fig. 5 | Murine and human synovial sarcomas exibit high levels of H2AK119ub1. a)** Immunofluorescence of *Hic1^{creERT2/creERT2}; Rosa26^{SSM2/SSM2}* mice at 16-week endpoint tongue tissue showing left, non-tamoxifen treated mice (-TAM) (upper panel) or tamoxifen treated mice expressing or not the SSM2 cassette (human SS18-SSX2) embedded in striated muscle +TAM; SSM2⁺ and +TAM; SSM2⁻ cells (lower panel). The cells are stained for DAPI, SSM2 and H2AK119ub1. The scale bar represents 100 um. **b)** Close-ups of images shown in the panel above, in the area delineated by the dashed square in (a). **c)** Quantification of H2AK119ub signal intensity normalised to DAPI signal intensity in 3 biological replicates (3 different mice) in non-tamoxifen treated mice (-TAM), or tamoxifen treated mice (+TAM) expressing or not the SSM2 cassette (human SS18-SSX2) and showing normal tongue muscle (+TAM; SSM2⁻) adjacent to synovial sarcoma tumours (+TAM; SSM2⁺). Asterisks represent p-values of paired one-tailed t-test between groups (p = 0.0006). **d)** Spearman correlation between SS18-SSX, left or SSX, right signals and H2AK119ub1 signals per sarcoma sample.

## Step1: imaging 3 channels

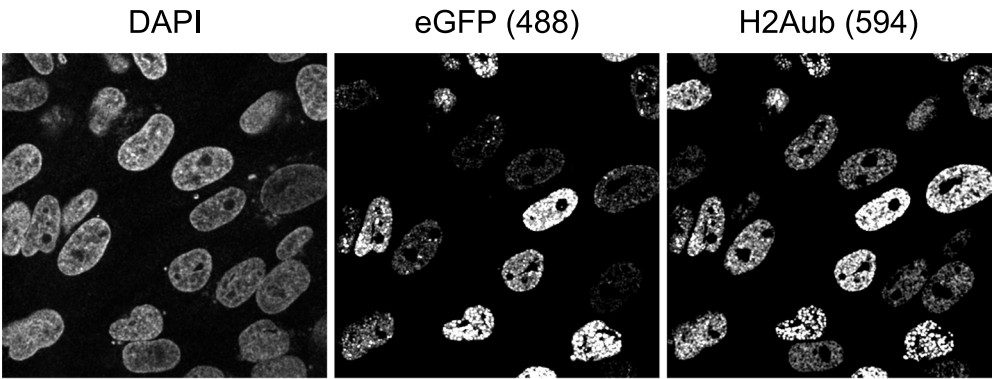

DAPI eGFP (488) H2Aub (594)

## Step2: nuclear signal intensities

Li threshold on DAPI selection region of interest (ROI)

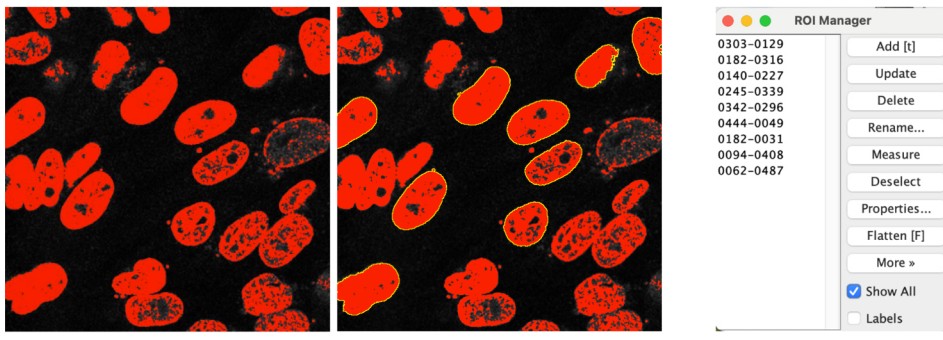

## Step3: normalization and computation

### keeping the mean

|       | DAPI    | eGFP 488 | H2Aub 594 |
|-------|---------|----------|-----------|
| Area1 | 315.976 | 315.976  | 315.976   |
| Mean1 | 75.18   | 67.296   | 32.749    |
| Min1  | 23      | 8        | 0         |
| Max1  | 226     | 255      | 156       |
| Area2 | 288.517 | 288.517  | 288.517   |
| Mean2 | 60.584  | 118.883  | 34.346    |
| Min2  | 16      | 9        | 0         |
| Max2  | 154     | 255      | 172       |
| Area3 | 354.577 | 354.577  | 354.577   |
| Mean3 | 72.479  | 250.371  | 195.101   |
| Min3  | 6       | 56       | 2         |
| Max3  | 187     | 255      | 255       |
| Area4 | 277.772 | 277.772  | 277.772   |
| Mean4 | 52.376  | 7.579    | 17.318    |
| Min4  | 21      | 7        | 0         |
| Max4  | 135     | 15       | 69        |

**Low eGFP**

| | H2Aub |
|--|--|
| 0.05149104 | 0.29868143 |
| 0.06808208 | 0.40636938 |
| 0.09196129 | 0.64836287 |
| 0.09983039 | 0.51407321 |
| 0.10063245 | 1.00849086 |
| 0.10288041 | 0.23055511 |
| 0.13002954 | 0.8308028 |
| 0.13762587 | 1.18001438 |
| 0.14470368 | 0.33064762 |
| 0.28838241 | 0.43993299 |
| 0.42878836 | 0.85378042 |
| 0.55650851 | 1.00357069 |
| 0.62883815 | 0.99125588 |
| 0.89513168 | 0.43560787 |
| 0.96369027 | 1.05995362 |
| 0.98493685 | 1.87623421 |

average → 0.75677083

**High eGFP**

| | |
|--|--|
| 1.26428195 | 1.28985462 |
| 1.61083217 | 0.52538257 |
| 1.69241229 | 1.33589608 |
| 1.81243932 | 1.32829739 |
| 1.96228377 | 0.56691536 |
| 2.09886952 | 2.29091991 |
| 2.31948764 | 1.90352586 |
| 2.37189316 | 2.01132146 |
| 2.38605461 | 1.87675987 |
| 2.86231157 | 2.18602477 |
| 2.86430846 | 1.97892566 |
| 3.30055099 | 2.25796716 |
| 3.35919341 | 2.73038797 |
| 3.45439369 | 2.69182798 |
| 3.48590207 | 2.83717028 |
| 3.5201983  | 2.26390533 |
| 3.55294699 | 2.66338974 |
| 3.64125381 | 2.93021706 |
| 4.17479505 | 2.40567345 |

average → 2.00391382

eGFP/DAPI H2Aub/DAPI

2.64797971 fluorescence ratio = High/Low

**Extended Data Fig. 6 | Protocol for eGFP-based imaging quantification of protein levels.** Step1 imaging on the confocal of the eGFP overexpressing constructs. Step2, using FiJi and Li thresholding, selecting and marking of the nuclei as Region of interest (ROI). Computing signal intensity for each ROI, the mean is kept and normalized for each channel to its corresponding DAPI value. Step 3 the values are separated in high eGFP (signal above 1) and low eGFP (signal below 1). The fluorescence ratio for a specific channel is computed by dividing its average in the high eGFP population on the low eGFP population.

# Reporting Summary

## Statistics

For all statistical analyses, confirm that the following items are present in the figure legend, table legend, main text, or Methods section.

| n/a | Confirmed | |
|---|---|---|
| ☐ | ☒ | The exact sample size (*n*) for each experimental group/condition, given as a discrete number and unit of measurement |
| ☐ | ☒ | A statement on whether measurements were taken from distinct samples or whether the same sample was measured repeatedly |
| ☐ | ☒ | The statistical test(s) used AND whether they are one- or two-sided<br>*Only common tests should be described solely by name; describe more complex techniques in the Methods section.* |
| ☒ | ☐ | A description of all covariates tested |
| ☒ | ☐ | A description of any assumptions or corrections, such as tests of normality and adjustment for multiple comparisons |
| ☐ | ☒ | A full description of the statistical parameters including central tendency (e.g. means) or other basic estimates (e.g. regression coefficient) AND variation (e.g. standard deviation) or associated estimates of uncertainty (e.g. confidence intervals) |
| ☐ | ☒ | For null hypothesis testing, the test statistic (e.g. *F*, *t*, *r*) with confidence intervals, effect sizes, degrees of freedom and *P* value noted<br>*Give P values as exact values whenever suitable.* |
| ☒ | ☐ | For Bayesian analysis, information on the choice of priors and Markov chain Monte Carlo settings |
| ☒ | ☐ | For hierarchical and complex designs, identification of the appropriate level for tests and full reporting of outcomes |
| ☒ | ☐ | Estimates of effect sizes (e.g. Cohen's *d*, Pearson's *r*), indicating how they were calculated |

*Our web collection on statistics for biologists contains articles on many of the points above.*

## Software and code

Policy information about availability of computer code

| | |
|---|---|
| Data collection | Imaging was done using the Leica Application Suite Software (4.1). Western blots were imaged using Amersham Imager 680. qRT-PCRs using Roche LightCycler480 II.  ChIP libraries were sequenced as 75 bp Single-Read on Illumina NextSeq 550 platform High-Output. SS18-SSX and H2AK119ub1 CUT&RUN libraries were sequenced as 75 bp Paired-End reads on Illumina NextSeq 550 platform Mid-Output. HA-eGFP-SS18, HA-eGFP-SS18-SSX1 and HA-eGFP-SSX-C CUT&RUN libraries and Native H2AK119ub1 calibrated ChIP were sequenced as 50bp Paired-End reads on NovaSeq 6K SP. RNA libraries were sequenced on a NovaSeq 6K Paired-End 100 S4 |
| Data analysis | For ChIP-Seq analysis: Raw reads were trimmed for quality and Illumina adapter sequences using Trim Galore! (Galaxy Version 0.6.7+galaxy0), then aligned to the human genome assembly Hg38 using Bowtie2 (Galaxy Version 2.4.2+galaxy0). ChIP signals were normalised to their respective inputs using the pileup function from  MACS2 callpeak (Galaxy Version 2.1.1.20160309.6) using corresponding input for background normalization. To visualize ChIP-Seq tracks, normalized bigWig files were generated with Wig/BedGraph-to-bigWig converter (Galaxy Version 1.1.1).<br><br>For the Cut and Run: Paired-end reads were aligned to the T2T or E.coli K12, MG1655 reference genome using Bowtie2 (Galaxy Version 2.4.2+galaxy0). Genome coverage files were generated using  bamCoverage (Galaxy Version 3.5.1.0.0).<br><br>Image analysis was done using the Fiji software (2.9.0).<br>Statistics and graphs were done using Excel (16.75.2) or Prism (9.4.0).<br>R (versions 3.6.0 to 4.2.1; https://www.r-project.org/).<br>Flow Cytometry analysis: FlowJo (v10.9.0)<br>Gene editing efficiency: Tide (v3.3.0)<br>RNA-seq analysis:IDEP.93 http://bioinformatics.sdstate.edu/idep93/ |

Gene Tiling screen: MAGeCK (0.5.9) and ProTiler: (1.0.2) were run on Python (3.7.0)

I used python 3.7.0

For manuscripts utilizing custom algorithms or software that are central to the research but not yet described in published literature, software must be made available to editors and reviewers. We strongly encourage code deposition in a community repository (e.g. GitHub). See the Nature Portfolio guidelines for submitting code & software for further information.

## Data

Policy information about availability of data

All manuscripts must include a data availability statement. This statement should provide the following information, where applicable:
- Accession codes, unique identifiers, or web links for publicly available datasets
- A description of any restrictions on data availability
- For clinical datasets or third party data, please ensure that the statement adheres to our policy

Re-analysed HA-SS18-SSX1 and KDM2B ChIP sequencing data originates from GEO accession number GSE108929. The GEO accession number for all data created in this paper is reported under GSE205955. Genome assembly used for ChIP and Cut&Run was hg38, for RNAseq hg19. Proteomic data is provided as a supplementary table.

## Human research participants

Policy information about studies involving human research participants and Sex and Gender in Research.

| Reporting on sex and gender | Sex and gender considerations where not relevant for the biological question addressed in this study. |
|---|---|
| Population characteristics | Patients who underwent surgical excision specimens in the Vancouver General Hospital to which material was archived between 2007 and 2020. |
| Recruitment | N/A |
| Ethics oversight | Tissue Microarray construction from anonymized patient primary surgical excision specimens was performed under protocols H18-00524 and H18-02391, approved by the Clinical Research Ethics Board of the University of British Columbia and BC Cancer. |

Note that full information on the approval of the study protocol must also be provided in the manuscript.

# Field-specific reporting

Please select the one below that is the best fit for your research. If you are not sure, read the appropriate sections before making your selection.

☒ Life sciences          ☐ Behavioural & social sciences          ☐ Ecological, evolutionary & environmental sciences

For a reference copy of the document with all sections, see nature.com/documents/nr-reporting-summary-flat.pdf

# Life sciences study design

All studies must disclose on these points even when the disclosure is negative.

| Sample size | The sample size was not predetermined statistically. The sample size, which is listed in the material and method section, was chosen based on expected variance of the experiments and technical limitations. |
|---|---|
| Data exclusions | No data exclusions was performed. |
| Replication | All experiments were repeated independently. The number of biological replicates are indicated in the figure legends. |
| Randomization | Not relevant because the samples were no grouped, as it includes only molecular assays performed in cell lines of known genotype. |
| Blinding | Blinding was not relevant for this study as there were no prior assumptions about experimental outcomes. All data was collected and processed uniformly regardless of treatment groups |

# Reporting for specific materials, systems and methods

We require information from authors about some types of materials, experimental systems and methods used in many studies. Here, indicate whether each material, system or method listed is relevant to your study. If you are not sure if a list item applies to your research, read the appropriate section before selecting a response.

## Materials & experimental systems

| n/a | Involved in the study |
|-----|----------------------|
| ☐ | ☒ Antibodies |
| ☐ | ☒ Eukaryotic cell lines |
| ☒ | ☐ Palaeontology and archaeology |
| ☐ | ☒ Animals and other organisms |
| ☒ | ☐ Clinical data |
| ☒ | ☐ Dual use research of concern |

## Methods

| n/a | Involved in the study |
|-----|----------------------|
| ☐ | ☒ ChIP-seq |
| ☒ | ☐ Flow cytometry |
| ☒ | ☐ MRI-based neuroimaging |

# Antibodies

| Antibodies used | anti-NanoLuc, R&D Systems, MAB100261 |
|-----------------|--------------------------------------|
| | BCoR (C10), Santa Cruz, sc-514576 |
| | EED (E4L6E) XP® Rabbit mAb, Cell Signaling, #85322 |
| | Ezh2 (D2C9) XP Rabbit mAb, Cell Signaling, #5246 |
| | GFP (D5.1) XP® Rabbit mAb, Cell Signaling, #2956 |
| | HA-tag, Abcam, #9110 |
| | HA-Tag (6E2) Mouse mAB, Cell Signaling, #2367S |
| | HA-Tag (C29F4) Rabbit mAb, Cell Signaling, #3724 |
| | Histone H3 (1B1B2), Cell Signaling, #14269 |
| | HP1 (E-6), Santa Cruz, sc- 515341 |
| | PCGF1 (E-8), Santa Cruz, sc-515371 |
| | SS18-SSX (E9X9V) XP, Cell Signaling, #72364 |
| | SS18/SSX Antibody, Cell Signaling, #70929 |
| | SSX (E5A2C), Cell Signaling,  #23855 |
| | ß-Actin HRP, Sigma, A3854 |
| | SYT (a-10), Santa Cruz, sc-365170 |
| | Tri-Methyl-Histone H3 (Lys27) (C36B11), Cell Signaling, #9733 |
| | Tri-Methyl-Histone H3 (Lys9) (D4W1U), Cell Signaling, #13969 |
| | Ubiquityl-Histone H2A (Lys119) (D27C4) XP® Rabbit mAb, Cell Signaling, #8240 |
| | Acetyl-Histone H3 (Lys27) (D5E4) XP, #8173 |
| | V5 Tag Monoclonal Antibody (2F11F7), Alexa Fluor 555, Thermofisher, 2F11F7 |
| | v5-Probe (E10), Santa Cruz, sc-81594 |
| | V5-Tag (E9H8O)  mAb, Cell Signaling, #80076 |
| | V5-Taq (D3H8Q) Rabbit mAb, Cell Signaling, #13202 |
| | Anti-TATA binding protein TBP antibody[mAbcam51841], Abcam, ab300656 |
| | SMARCC1/BAF155 (D7F8S) Rabbit mAb, Cell Signaling,  #11956 |
| | BRM (D9E8B) XP® Rabbit mAb, Cell Signaling, #11966 |
| | p300 (D8Z4E), Cell Signaling, #86377S |
| | Brg-1 (G-7), Santa Cruz, sc-17796 |
| | ARID1A/BAF250A (D2A8U) Rabbit, Cell Signaling, #12354 |
| | BCOR polyclonal antibody, Proteintech, 12107-1-AP |
| | |
| | Secondary antibodies |
| | ECL Anti-Mouse IgG |
| | ECL Anti-Rabbit IgG |

| Validation | All antibodies were validated by the manufacturers. Tag antibodies (HA, V5) have been extensively used in the literature. In addition, the majority of primary antibodies were further validated using target-specific knockouts (PCGF1 E-8, EZH2 D2C9, EEDE4L6E, BRM D9E8B) or over-expression of a tagged proteins (NanoLuc MAB100261, BCOR polyclonal antibody, Proteintech, 12107-1-AP, BCoR (C10), Santa Cruz, sc-514576). |
|------------|---|
| | Antibodies against SS18-SSX and SSX (SS18-SSX (E9X9V) XP, Cell Signaling, #72364 SS18/SSX Antibody, Cell Signaling, #70929 SSX (E5A2C), Cell Signaling,  #23855) have been validated by Baranov et al (PMID: 32141887). |
| | |
| | p300 (D8Z4E), Cell Signaling, #86377S, Brg-1 (G-7), Santa Cruz, sc-17796, ARID1A/BAF250A (D2A8U) Rabbit, Cell Signaling, #12354 have been used previously in many other studies. See for example PMID: 33651988 (EP300), PMID: 35732731 (BRG1) and PMID: 36435834 (ARID1A). |
| | |
| | Antibodies used for histone and  histone marks have been extensively used in the literature: |
| | Tri-Methyl-Histone H3 (Lys27) (C36B11), Cell Signaling, #9733 (1065 citations) |
| | Tri-Methyl-Histone H3 (Lys9) (D4W1U), Cell Signaling, #13969 (112 citations) |
| | Ubiquityl-Histone H2A (Lys119) (D27C4) XP® Rabbit mAb, Cell Signaling, #8240 (298 citations) |
| | Histone H3 (1B1B2), Cell Signaling, #14269 (137 citations) |
| | HP1 (E-6), Santa Cruz, sc- 515341 (more than 100 citations) |

# Eukaryotic cell lines

Policy information about cell lines and Sex and Gender in Research

| Cell line source(s) | Human synovial sarcoma cell lines: HS-SY-II (RRID:CVCL_8719) and SYO-1 (RRID:CVCL_7146) were obtained fom their original |
|---------------------|---|

| Cell line source(s) | source laboratories. Human osteosarcoma KHOS-240S (RRID:CVCL_2544) and Human Embryonic Kidney HEK293T (RRID:CVCL_0063) were purchased from the American Type Culture Collection (ATCC). ASC52telo, hTERT immortalized adipose derived Mesenchymal stem cells were purchased from ATCC (SCRC-4000). Drosophila SG-4 cell line used for calibrated ChIP was provided by Angelika Feldmann (German Cancer Research Center). |
|---|---|
| Authentication | HS-SY-II and SYO-I were authenticated via classical STR profiling with the company Multiplexion and by western blot for SS18-SSX1/2 detection. he remaining cell lines (MSCs, HEK293-T, KHOS-240S and SG-4) were obtained authenticated by the manufacturer (ATCC) and by morphology. |
| Mycoplasma contamination | Cell lines were monthly tested against mycoplasma contamination and remained negative. |
| Commonly misidentified lines (See ICLAC register) | No misidentified cell lines were used in this study. |

# Animals and other research organisms

Policy information about studies involving animals; ARRIVE guidelines recommended for reporting animal research, and Sex and Gender in Research

| Laboratory animals | This study used Mus musculus (C57BL/6J) with a conditional SS18-SSX2-IRES-eGFP allele knocked into the Rosa26 locus. Tamoxifen treatment was performed on 8 weeks old mice. Mice were housed under standard conditions (12 h light/dark cycle) and provided food and water ad libitum. Animals were maintained in a controlled environment of between 21-24o C and 40-60% humidity, and experimental protocols were conducted in accordance with approved and ethical treatment standards of the Animal Care Committee at the University of British Columbia. |
|---|---|
| Wild animals | No wild animals were used. |
| Reporting on sex | No sex specific data was used. |
| Field-collected samples | No field-collected samples were used. |
| Ethics oversight | Animals were maintained and experimental protocols were conducted in accordance with approved and ethical treatment standards of the Animal Care Committee at the University of British Columbia. |

Note that full information on the approval of the study protocol must also be provided in the manuscript.

# ChIP-seq

## Data deposition

☒ Confirm that both raw and final processed data have been deposited in a public database such as GEO.

☒ Confirm that you have deposited or provided access to graph files (e.g. BED files) for the called peaks.

| Data access links May remain private before publication. | https://www.ncbi.nlm.nih.gov/geo/query/acc.cgi?acc=GSE205955 |
|---|---|
| Files in database submission | GSM6235997  HSSY, eGFP, input<br>GSM6235998  HSSY, eGFP, HA<br>GSM6235999  HSSY_SSX_input<br>GSM6236000  HSSY_SSX_HA<br>GSM6236001  HSSY_SSXDeltaRD_input<br>GSM6236002  HSSY_SSXDeltaRD_HA<br>GSM6236003  HSSY, IgG<br>GSM6236004  HSSY, MacroH2A2<br>GSM6236005  HSSY, Control, H2Aub1<br>GSM6236006  HSSY, Control, HA (SS18-SSX1)<br>GSM6236007  HSSY, PCGF1 knockout, H2Aub1<br>GSM6236008  HSSY, PCGF1 knockout, HA (SS18-SSX1)<br>GSM6236009  SYOI, IgG<br>GSM6236010  SYOI, Control, H2Aub1<br>GSM6236011  SYOI, Control, SS18-SSX2<br>GSM6236012  SYOI, PCGF1 knockout, H2Aub1<br>GSM6236013  SYOI, PCGF1 knockout, SS18-SSX2 |
| Genome browser session (e.g. UCSC) | |

Here is the link to the updated bigwigs:
https://dl.dropboxusercontent.com/s/b9ordhxtpcac3gf/HSS_cChIP_EV_H2A.bigwig?dl=0
cCHIP_sgPCGF1_H2A
https://dl.dropboxusercontent.com/s/xlvs7fhu6uwrrk4/HSS_cChIP_sgPCGF1_H2A.bigwig?dl=0
HSS_CR_EV_H2Aub
https://dl.dropboxusercontent.com/s/zdv5a4l7zlkpw01/HSS_CR_EV_H2Aub.bigwig?dl=0
HSS_CR_EV_SS18SSX
https://dl.dropboxusercontent.com/s/htjf4d0n9czyfan/HSS_CR_EV_SS18-SSX.bigwig?dl=0
HSS_CR_sgPCGF1_H2Aub
https://dl.dropboxusercontent.com/s/yi8cjf4lejjxzr5/HSS_CR_sgPCGF1_H2Aub.bigwig?dl=0
HSS_CR_sgPCGF1_SS18SSX
https://dl.dropboxusercontent.com/s/8wfyomyzam0rnev/HSS_CR_sgPCGF1_SS18-SSX.bigwig?dl=0
HSS_CR_SSX_C_DOX
https://dl.dropboxusercontent.com/s/ecklriqyjtyjh68/HSS_CR_SSX-C_Dox.bigwig?dl=0
HSS_CR_SSX_C_NoDox
https://dl.dropboxusercontent.com/s/j4n6197q8p8e0yi/HSS_CR_SSX-C_NoDox.bigwig?dl=0
KHOS_CR_EV
https://dl.dropboxusercontent.com/s/01lv1a8shce6kaz/KHOS_CR_EV.bigwig?dl=0
KHOS_CR_SS18
https://dl.dropboxusercontent.com/s/j4kmdt4su6t9w31/KHOS_CR_SS18.bigwig?dl=0
KHOS_CR_SS18-SSX
https://dl.dropboxusercontent.com/s/mzqs2x99k4grdst/KHOS_CR_SS18-SSX.bigwig?dl=0
KHOS_CR_SSX
https://dl.dropboxusercontent.com/s/rqsxm3e18wuovvd/KHOS_CR_SSX.bigwig?dl=0
SYOI_CR_EV_H2Aub
https://dl.dropboxusercontent.com/s/rbop41e9on1l4im/SYOI_CR_EV_H2Aub.bigwig?dl=0
SYOI_CR_EV_SS18SSX
https://dl.dropboxusercontent.com/s/a022ih0b4jv4fj6/SYOI_CR_EV_SS18-SSX.bigwig?dl=0
SYOI_sgPCGF1_H2Aub
https://dl.dropboxusercontent.com/s/efiruldy66pvvyy/SYOI_CR_sgPCGF1_H2Aub.bigwig?dl=0
SYOI_sgPCG1_SS18SSX
https://dl.dropboxusercontent.com/s/hurzbom8lfdauhn/SYOI_CR_sgPCGF1_SS18-SSX.bigwig?dl=0

## Methodology

**Replicates**

The ChIP sequencing was performed in a unique biological replicate. Cut and Run sequencing of endogenous SS18-SSX and H2AK119ub1 was performed in two independent cell lines HS-SY-II and SYO-I. Native H2AK119ub1 calibrated ChIP was perfomed in biological triplicates. Cut and Run of overexpressed constructs was done in one replicate.

**Sequencing depth**

ChIP libraries were sequenced as 75 bp Single-Read on Illumina NextSeq 550 platform High-Output.
total reads.
eGFP input: 31465378
eGFP HA: 35710039
SSX-C input: 39821585
SSX-C HA: 35710039
DeltaRD input: 38916894
DetaRD HA: 27970228

CUT&RUN libraries were sequenced as 75 bp Paired-End reads on Illumina NextSeq 550 platform Mid-Output.
total reads:
HSS IgG: 13526884
HSS EV H2Aub: 19510962
HSS EV SS18-SSX: 11610526
HSS sgPCGF1 H2Aub: 15938066
HSS sgPCGF1 SS18-SSX: 13320300
SYOI IgG: 23814054
SYOI EV H2Aub: 23120676
SYOI EV SS18-SSX: 20301480
SYOI sgPCGF1 H2Aub: 25397092
SYOI sgPCGF1 SS18-SSX: 15248120

CUT&RUN for overexpression were sequenced as 50bp Paired-End reads on NovaSeq 6K SP.
total reads:
KHOS IgG 27488012
KHOS EV 55388126
KHOS SS18 45457932
KHOS SS18-SSX 41549600
KHOS SSX-C 46110250

Native H2AK119ub1 calibrated ChIP were sequenced as 50bp Paired-End reads on NovaSeq 6K SP.
total reads:
HSS EV R1 23109716
HSS EV R2 29247779:
HSS EV R3 29882136:
HSS sgPCF1 R1 29064591

HSS sgPCF1 R2 39902391
HSS sgPCF1 R3 49357058

| | |
|---|---|
| Antibodies | The ChIP sequencing has been done using the HA-tag antibody #9110 from Abcam.<br>For the Cut and Run we used HA-Tag (C29F4) Rabbit mAb #3724 Cell Signaling,<br>SS18-SSX (E9X9V) XP #72364 Cell Signaling, Ubiquityl-Histone H2A (Lys119) (D27C4) XP® Rabbit mAb #8240 Cell Signaling.<br>Native H2AK119ub1 calibrated ChIP was done using Ubiquityl-Histone H2A (Lys119) (D27C4) XP® Rabbit mAb #8240 Cell Signaling. |
| Peak calling parameters | For ChIP sequencing HA-SS18-SSX1 peaks (n=26805) were generated with the MACS2 function (with "--no model", "--qvalue 0.05", "--broad" options) and normalized to input.<br>For H2AK119ub1 (n=11099) and SS18-SSX2 (n= 27686) peak calling, the MACS2 callpeak function was used on the aligned BAM files and IgG as control (with "--nomodel", "--qvalue 0.01", "--broad" options, "--keep-dup all"). For HA peak calling in KHOS-240S, HA-SS18, HA-SS18-SSX1 and HA-eGFP-SSX1 were combined in MACS2 to compute all the HA peaks (n=58843). |
| Data quality | We assessed the quality of the sequencing using FastQC and Plotfingerprint. For Cut and Run, the percentage of reads mapped to E.Coli was also an indication of the success of assay. As an indication, histone H2Aub pull down has a percentage of E.Coli reads between 0.5 and 1. For SS18-SSX the percentage is between 1.5 and 3.5 % and IgG around 15%. Therefore a Cut and Run sequencing that shows abnormally high E.Coli content is a sign that the pull down did not work. |
| Software | We used Galaxy program for the analysis of the ChIP sequencing and the Cut and Run.<br>For ChIP sequencing: Raw reads were trimmed for quality and Illumina adapter sequences using Trim Galore! (Galaxy Version 0.6.7 +galaxy0), then aligned to the human genome assembly Hg38 using Bowtie2 (Galaxy Version 2.4.2+galaxy0). ChIP signals were normalised to their respective inputs using the pileup function from MACS2 callpeak (Galaxy Version 2.1.1.20160309.6) using corresponding input for background normalization. To visualize ChIP-Seq tracks, normalized bigWig files were generated with Wig/ BedGraph-to-bigWig converter (Galaxy Version 1.1.1).<br>For the Cut and Run: Paired-end reads were aligned to the T2T or E.coli K12, MG1655 reference genome using Bowtie2 (Galaxy Version 2.4.2+galaxy0). Genome coverage files were generated using bamCoverage (Galaxy Version 3.5.1.0.0). |

