## [Peer Review File · Nature Structural & Molecular Biology]

Peer Review Information

Manuscript Title: Aberrant gene activation in synovial sarcoma relies on SSX specificity and increased PRC1.1 stability

Corresponding author name(s): Ana Banito

Reviewer Comments & Decisions:

Decision Letter, initial version:
--

Message:

Nature Structural & Molecular Biology NSMB-A46664-T

22nd Nov 2022

Dear Dr. Banito,

Thank you for submitting your manuscript, "An autoregulatory feedback loop converging on H2A ubiquitination drives synovial sarcoma". Please accept my sincere apologies for the very long delay in sending you a decision on your study, it took longer than expected to receive the full set of referee reports.

The comments of 3 expert referees are below. You will see that all three reviewers have serious concerns regarding the conceptual advance the study represents for the field - I am afraid these reflect our initial concerns as well. Based on these comments, we cannot offer to publish the study in Nature Structural & Molecular Biology. We hope the referees' comments will be useful to you in revising the manuscript for submission elsewhere.

A revised version of your study may be interest for further consideration at Nature Communications - please let me know if you would wish for us to consult our colleagues at Nature Communications on your behalf.

I am sorry we could not be more positive on this occasion.

Sincerely,

Carolina

Carolina Perdigoto, PhD
Chief Editor
Nature Structural & Molecular Biology
orcid.org/0000-0002-5783-7106

Reviewer #1 (Remarks to the Author):

Dr Banito and colleagues present a paper, entitled "An autoregulatory feedback loop converging on H2A ubiquitination drives synovial sarcoma". They report that the synovial sarcoma SS18-SSX fusion binds to H2AK119ub via its C-terminal SSXRD domain, which they propose contributes to directing it to chromatin. Supporting this, they show that deletion of the variant PRC1.1 complex member PCGF1 leads to reduced SS18-SSX binding on H2AK119ub enriched regions. It's important to note that the physical association between SS18-SSX and PRC1 mediated H2AK119ub has been made previously (PMID: 32747783). Please see below my specific comments on the manuscript.

Major:

1. Overall, I feel the paper is pushing the narrative that vPRC1 is the primary functional element that drives oncogenic transformation by SS18-SSX in synovial sarcoma. However, other studies have clearly established that the BAF complex also has a key role. For example, BAF complex member SMARCA4 is clearly a strong genetic dependency in synovial sarcomas (See Figure 1a in PMID: 34836971); and several studies have demonstrated that ncBAF complexes containing the BRD9 protein (and SS18-SSX) are essential in these cells. Moreover, the work in a related manuscript by the Kadoch lab has already linked SS18-SSX, via the SSX portion of the protein with H2A/H2AK119ub binding (PMID: 32747783).

2. For Figure 1, and related to my comment above, it would be important to evaluate the full-length wild-type SS18-SSX1 and mutant SS18-SSX1 (delta RD), and not just the SSX-C. It's clear in the staining that full length SS18-SSX has a somewhat different binding pattern to SSX alone. This indicates that the SS18 portion of the fusion is likely directing binding of the fusion to chromatin, probably via integration into BAF complexes. Therefore, it remains important to include these proteins in the comparison. Perhaps also including a mutant previously established by the Kadoch lab; where mutation of the residue at position 169 of the fusion protein abrogated H2A binding. This will better delineate the relative contribution of the SSXRD domain to chromatin binding of the SS18-SSX1 fusion oncogene.

3. In Figures 1h, it is not clear if the ChIP-seq analyses is performed with spike-in or not. The authors should clarify in the legends whether ChIP and CUT&RUN analyses are spike controlled, as indicated in the methods. This is very important given the large global shifts in the abundance of their targets in these assays. It's well established that large global shifts in the abundance of targets (proteins/modifications) in chromatin mapping studies must be normalised with a reference spike-in. If this is not done, the reliability of the measured differences is questionable.

4. The authors report that PRC1.1 is essential for depositing macroH2A in chromatin. However, their experimentation to support this is lacking. The experiments performed have utilised sgRNA mediated knockdown/out of PRC1 members which associates with reduced macroH2A. However, it's very important to note that chromatin dynamics and

histone deposition is regulated in a highly dynamic manner. Low term genetic knockout of a PRC1 factor and associated reductions in macroH2A abundance cannot be directly linked. On these long experimental time courses, it is possible that these changes are not directly linked and instead correlative. If the authors wish to prove that the two events are linked, they should utilise an experimental approach using for example, targeted protein degradation of PRC1 members. This would allow an experimental manipulation on a time scale relevant to the processes dictating histone deposition and turnover. Without such an experimental approach, it is impossible to conclude that the loss of macroH2A is anything more than a downstream correlation to PRC1 member knockout.

5. In Figure 4, the authors knockout the vPRC1 component PCGF1 and show that the consequent reduction of H2AK119ub correlates with a reduction of SS18-SSX binding to chromatin, supporting their model. However, not all SS18-SSX is removed. While this might be a consequence of failure to remove all H2AK119ub1 (other vPRC1 complexes can presumably compensate), the authors should perform a more extensive analyses of their CUT & RUN data. For example, they could divide each of their SS18-SSX and H2AK119ub1 datasets into 3-5 categories of changing most, to changing least. This might help answer questions such as: does SS18-SSX remain on sites in which H2AK119ub1 is least changed? Vice versa, is H2AK119ub1 least changed at sites that SS18-SSX is least reduced in PCGF1 KO cells? Are there sites in which SS18-SSX binds that are weakly or not enriched for H2AK119ub, and again, vice versa, are there sites that are enriched for H2AK119ub1 that do not have SS18-SSX binding?

6. The authors report that high H2AK119ub1 levels are a feature of synovial sarcomas, but is it important? Does knockout of PCGF1 slow the proliferation of synovial sarcomas?

Minor:

1. Figure 1e lacks the necessary controls to confirm equal loading/extraction. They should also blot for other 1-2 endogenous chromatin proteins.

Reviewer #2 (Remarks to the Author):

Benabdallah and colleagues investigate the role of PRC1.1 in SS18-SSX driven synovial sarcoma. They perform a gene-tiling screen with a library of guideRNAs against the SS18-SSX fusion and demonstrate that the RD domain of SSX is required and sufficient for high affinity chromatin binding. They then utilize IP-MS of overexpressed eGFP compared to SSX-Cterminus-eGFP and SSX-C terminus-eGFP compared to SSX-C-E184*-eGFP (identified in McBride et al NSMB 2020) to identify association between the SSX-C terminus and histones, as well as MacrohistoneH2A1 and MacrohistoneH2A2. Referencing prior work from the Kadoch lab (McBride et al NSMB 2020), they compare the overlap of SS18-SSX with H2AK119Ub and find using BRET that the SSX RD domain and terminal acidic domain (E184*) are required for association with H2AK119Ub and MacroH2A1/A2. They make use of a synthetic biology strategy to retarget PRC1.1 component KDM2B to methylated CpG sites by fusing it with an MBD domain and deleting the CXXC domain in KDM2B. Using this approach, they find that SS18-SSX is recruited to MBD-KDM2B foci and the colocalization of these proteins is dependent on PRC1.1 subunits BCOR and PCGF1, but not MacroH2A. On chromatin, they find that knockdown of PCGF1 results in global

reduction of H2AK119Ub and SS18-SSX binding, which correlates with reduced affinity for chromatin (not found for knockdown of PCGF3). They further find that overexpression of SS18-SSX in mesenchymal stem cells leads to transcriptional upregulation of BCOR and PCGF1 as well as increased H2AK119Ub, while knockdown of SSX or PCGF1 in SS cells reduced H2AK119Ub and BCOR, PCGF1. Overexpression of SSX-C alone in SS cells results in more BCOR, SS18, and H2AK119Ub and increased affinity of BCOR/PCGF1 for chromatin. Finally, an animal model of SS shows kinetic increase in H2AK119Ub with tumorigenic progression and human SS samples show increased H2AK119Ub relative to other sarcomas or normal tissue.

This study is a nice followup of previous work from Banito and colleagues on the dependence of SS on KDM2B and other subunits of the PRC1.1 complex primarily through KDM2B-dependent recruitment of SS18-SSX to PRC1.1 binding sites (Banito Cancer Cell 2018). However, the work is largely incremental in light of work from Kadoch and colleagues identifying the RD domain of SSX and indeed the exact residues that encode high affinity binding of SSX to histones, H2AK119Ub, and chromatin (by ChIP) (McBride NSMB 2020). Thus, most of Figure 1 and Figure 2 is not new information (the MacroH2A results are new, but appear to have a relatively small contribution to the mechanism of SS18-SSX oncogenesis). This work essentially ties the knot between these prior findings showing that PRC1.1 recruits SS18-SSX through the deposition of H2AK119Ub, which is perhaps not surprising given the known role of PRC1.1 in depositing H2AK119Ub. Below are a few comments that would strengthen the novel aspects of the study.

1) The authors show that PCGF1 is responsible for much of the H2AK119Ub and increased affinity for chromatin, and not PCGF3 (Figure 4d,e). This is important because CERES dependency scores implicate PCGF3 and PCGF5, in addition to KDM2B. It would be good to include confirmation of PCGF3 knockdown in Figure 4d and H2AK119Ub C&R for sgPCGF3, as well as SS18-SSX salt extraction and H2AK119Ub C&R for knockdown of PCGF5. It seems possible that multiple complexes could contribute H2AK119Ub and despite global reduction in H2AK119Ub in sgPCGF1 cells, some H2AK119Ub remains, which could be dependent on other PRC1 variants. Do PCGF1/3/5 double/triple knockouts have even greater reduction in H2AK119Ub, MacroH2A, and SS18-SSX?

2) What I found potentially very interesting was the sufficiency of SSX alone to bind chromatin (1g, h). The statement by the authors, "These results indicate that the SSX c-terminus, via its SSXRD, binds specific regions in the genome and determines SS18-SSX localisation independently of SS18 and therefore of the mSWI/SNF complex". However, as far as I can tell, these constructs are being overexpressed in an SS cell line which already expresses SS18-SSX. This also is the case for Figure 6a-e. It's therefore hard to know whether SSX-C activity is in some part influenced by the expression of the fusion and the effect it has already had on chromatin. I wonder if SSX-C is simply going to where the fusion is and acting like an amplification event. Indeed, SSX-C increases SS18 expression. To be able to make this statement (which indeed is quite controversial), I would suggest the authors overexpress SSX-C and SSX-C-delta RD in mesenchymal stem cells and then perform ChIP and immunofluorescence for BCOR, H2AK119Ub, SS18 as in Figure 1g and Figure 6a-e. Furthermore, to definitively make this statement, the authors need to show that SSX-C binding to chromatin by ChIP is not diminished by knockdown of SMARCA4, or treatment with SMARCA4/A2 degraders.

3) Related, I found the activity / sufficiency of SSX to induce transcription of genes when

fused to EWSR1 or MN1 (Figure 1i, j) very interesting. These data suggest a new model for thinking about the molecular basis for this disease, especially in light of the identification of novel SSX fusions in SS. Using the EWSR1-SSX and MN1-SSX fusions in subsequent assays would further underscore the authors' proposed model for an autoregulatory loop – for example 1) showing EWSR1-SSX and MN1-SSX are recruited to MBD-KDM2B foci in a PCGF1-dependent mechanism and 2) showing increase BCOR and PCGF1 protein, H2AK119Ub levels in EWSR1-SSX and MN1-SSX expressing cells.

Reviewer #3 (Remarks to the Author):

This manuscript addresses the altered chromatin regulatory mechanisms arising from translocations of the SWI/SNF subunit SS18, which yield the hallmark SS18-SSX fusion proteins that drive synovial sarcoma. The authors' results are high quality, well-designed, and deepen the understanding of the broader chromatin regulatory systems that operate in synovial sarcoma. The most prominent issue that the authors need to correct is the overlap with major portions of work that was previously published on a similar topic (McBride et al., Nat Struct Mol Biol 2020, PMID 32747783). For example, in the McBride paper, the connections between the C-terminus of SS18-SSX and binding of H2AK119ub1 and PRC1 were already established and hence the novelty of these specific aspects in the current manuscript are weakened.

Despite the overlap with portions of the McBride paper, the current work does reveal a number of novel findings not found in previous works, including the existence and characterization of the feedback loop between SS18-SSX, H2AK119ub1, and PRC1.1, and observations about the role for the histone variant macroH2A. As a result, the authors' findings do extend the links between SSX and H2AK119ub1 beyond previous findings, including the lack of PRC2 involvement. As a result, the work is meritorious, but should be restructured to focus on these novel aspects, and I would suggest focusing on the mechanisms underlying the feedback loop. Authors should take care to further distinguish the work from the McBride paper.

Major concerns:

1. Portions of Figures 1 and 2 in the current work somewhat lack novelty as the result of Figure 4 and other results from McBride et al. The current work needs restructuring to highlight the novel aspects of the work and should move all the findings that strictly confirm McBride et al. out of the main figures to the supplement or remove them in order to highlight the novel aspects of their findings. I believe there are important novel findings in the current work that, if refocused and refined, would sufficiently differentiate it.
2. The autoregulatory feedback loop is perhaps the most novel aspect of this work, and authors should deepen some of the mechanisms underlying this feedback loop. For example, can the authors distinguish between increased expression (i.e. up-regulation of transcription/translation) of SS18 (and other SWI/SNF subunits), BCOR, and RYBP (and other PRC1.1 subunits) in contrast to increased stability/accumulation of these proteins? Analysis of RNA levels or proteasome inhibition studies could be helpful to reveal the mechanistic basis for accumulation of these proteins, and may potentially provide stronger evidence for a mechanism mediated by altered transcription.

3. Given that SWI/SNF ATPase activity is widely recognized to oppose Polycomb activity in most contexts (including PRC1; see e.g. PMIDs 27941795, 20951942, and many others), it would be somewhat surprising that targeting of SWI/SNF activity would promote Polycomb accumulation. How does SWI/SNF ATPase activity influence the accumulation of PRC1.1/H2AK119ub1 at SS18-SSX sites in this setting? To characterize the mechanisms of the feedback loop, authors could perform SWI/SNF inhibitor studies to assess the influence of SWI/SNF ATPase activity both on the level of survival/proliferation as well as PRC1.1/H2AK119ub1 (e.g. using ChIP-qPCR at a given target site).

4. The macroH2A knockout changes in Ext Figure 3 are modest, as the authors acknowledge. Yet the authors state “a very specific chromatin environment containing both H2AK119ub1-modified nucleosomes and histone variant MacroH2A underlies SSX C-terminus chromatin binding”; authors should perform a more conclusive experimental test to differentiate whether macroH2A is instructive vs. reflective of SS18-SSX targeting.

5. The authors’ in vivo analysis in Figure 6 and 7 show strong support for tight links between H2AK119ub1 and SS18-SSX, which goes beyond previous findings to implicate H2AK119ub1 as a key mechanistic feature of synovial sarcoma. But in my view, the findings are consistent with but are not explicitly constrained to an autoregulatory model. Instead, they are consistent with any potential model where H2AK119ub1 accumulates upon development of synovial sarcoma. Is there additional data that would strengthen the autoregulatory finding in vivo? For example, are other features of PRC1.1 increased in these tumors in vivo as seen in vitro? Alternatively, if the samples in Figure 7f were processed identically, and if an autoregulatory mechanism were at play, one would expect these intensity values to be correlated. Testing for a significant correlation between SS18-SSX and H2AK119ub1 would be strong support of their in vitro mechanisms, and go beyond the discovery that both H2AK119ub1 and SS18-SSX arise concomitantly in the malignancy. There may be other possibilities to more directly test the feedback of SS18-SSX fusions with H2AK119ub1 beyond their shared presence in synovial sarcoma tumors.

Minor concerns:

1. The Fig 1e results of the bands are quite variable and the results for SS18-SSX1 do not appear to agree with the quantification shown in Fig 1f (e.g. 300 mM measurement). Why is this? Interpretation would benefit from a more consistent results, especially for SS18-SSX1.

2. Fig 1g color signal and Fig 1h read depth metrics should be explicitly stated to ensure apples-to-apples comparisons.

3. Labeling in Fig 6d-e and potentially other figures could be improved to indicate that the constructs are EGFP fusions.

4. I do not understand the image pairing in Fig 7c; are these replicates? How many times was this performed? Do all targets contain the SSM2 construct? The analysis in Fig 7d suggests that samples without the SSM2 construct were also analyzed, and these should be presented as controls in the main figure.

5. Fig 7f-d and all other summary figures should have N values and it should be made

clear how many different animals these studies were performed in.

6. The summary figure in Figure 8 is confusing since the title describes an autoregulatory feedback loop, but little detail is provided for the feedback loop, and little information about the feedback loop is presented, other than the loop being depicted as “existing.” I suspect other readers will be confused by the message of this image, and this summary image should be clarified and if authors focus future work on this loop, this figure should be improved to focus on the newly discovered features of the autoregulatory loop.

7. Despite having read the Methods section describing it, the frequent use of “fluorescence ratio in high vs. low GFP” measurements is confusing and could be better explained or labeled. Authors should provide a supplemental figure and characterize for readers what high- and low-GFP means in this setting. Are the authors relying on variable expression levels within each condition to make these measurements? If so, authors should make clear that the EGFP intensity distribution is controlled for between constructs, or provide a clearer rationale for the use of this analytical approach.

** As a service to authors, Springer Nature Limited provides authors with the ability to transfer a manuscript that one journal cannot offer to publish to another journal, without the author having to upload the manuscript data again. To transfer your manuscript to another NPG journal using this service, please click on [redacted]

** For Springer Nature Limited general information and news for authors, see <http://npg.nature.com/authors>.

Author Rebuttal to Initial comments

NSMB-A46664-T: Response to reviewer's comments

We appreciate the reviewer's comments and suggestions on how to strengthen the main conceptual advances in our manuscript. Given the previous study by McBride et al., 2020 we understand the concerns regarding novelty. However, we also think that some of the conceptual advances in our paper did not come across clearly. Namely, previous studies including McBride et al., 2020 suggest a model where the contribution of the SSX tail to the oncogenic activity of SS18-SSX is to change BAF complex composition (SMARCB1 loss). This results in a SS18-SSX-specific conformation of BAF complexes, that includes a change in the orientation of the core module on the nucleosome thereby allowing it to bind H2AK119ub-rich regions. It is crucial to note that, in that model, SS18-SSX binding to chromatin is a consequence of altered BAF complex specificity. Here we show that the ability to bind Polycomb target regions enriched in H2AK119ub1, is an intrinsic property of SSX-C that can be exploited by fusion with other transcriptional activators. Therefore, SSX-C alone can phenocopy SS18-SSX chromatin occupancy and SS18-SSX recruitment occurs independently of its association with the BAF complex. The comments from the reviewers demonstrate that this is a topic that still needs clarification and the set of experiments suggested by the reviewers clearly made the difference in consolidating this point.

The current version of our manuscript includes several new experiments discussed in the point-by-point response to the reviewer's comments, and overall restructuring of the figures and the text. In summary:

1. The paper was restructured to be shorter and more concise. Data showing affinity of SSXRD domain to histones and in particular H2AK119ub1 is now presented in supplementary figures as suggested.
2. Figure 2 and Figure 3 are mostly new results that address two important points: the uncoupling between SS18 (BAF) and the SSX tail in determining oncofusion occupancy (Figure 2), and characterization of new SSX1 fusions in activating a synovial sarcoma signature in a BAF-independent manner (Figure 3).
3. Experiments that strengthen the feedback loop model were added as well. We show BCOR/H2AK119ub1 stabilization/increase in mesenchymal stem cells in response to all SSX fusions or SSX-C alone. Further we show that increased BCOR levels accompanies the increase in H2AK119ub1 *in vivo* during murine sarcomagenesis, and that SS18-SSX levels correlate with H2AK119ub1 levels in patient samples. We also changed the wording in the text and the title of the manuscript to better emphasize the SSX-C/PRC1.1 link that underlies SS18-SSX1 binding rather than SS18-SSX autoregulation which is experimentally very challenging to untangle and thus absolutely demonstrate.

Reviewer #1 (Remarks to the Author):

Dr Banito and colleagues present a paper, entitled “An autoregulatory feedback loop converging on H2A ubiquitination drives synovial sarcoma”. They report that the synovial sarcoma SS18-SSX fusion binds to H2AK119ub via its C-terminal SSXR domain, which they propose contributes to directing it to chromatin. Supporting this, they show that deletion of the variant PRC1.1 complex member PCGF1 leads to reduced SS18-SSX binding on H2AK119ub enriched regions. It’s important to note that the physical association between SS18-SSX and PRC1 mediated H2AK119ub has been made previously (PMID: 32747783). Please see below my specific comments on the manuscript.

Major:

1. Overall, I feel the paper is pushing the narrative that vPRC1 is the primary functional element that drives oncogenic transformation by SS18-SSX in synovial sarcoma. However, other studies have clearly established that the BAF complex also has a key role. For example, BAF complex member SMARCA4 is clearly a strong genetic dependency in synovial sarcomas (See Figure 1a in PMID: 34836971); and several studies have demonstrated that ncBAF complexes containing the BRD9 protein (and SS18-SSX) are essential in these cells. Moreover, the work in a related manuscript by the Kadoch lab has already linked SS18-SSX, via the SSX portion of the protein with H2A/H2AK119ub binding (PMID: 32747783).

We agree that the BAF complex plays a critical role in synovial sarcomas harboring an SS18-SSX fusion and that BAF components are strong dependencies. Indeed, both BAF and vPRC1 complex subunits are well described top dependencies in synovial sarcoma (Banito et al 2018, PMID 29502955; DepMap data shown in McBride et al, 2020 PMID:32747783). We rephrased parts in the manuscript to avoid any misunderstandings, see for example underlined text lines 85 and 174. What our study shows, is that it is the SSX tail, via PRC1.1-deposited H2AK119ub1 that determines fusion binding specificity independently of SS18 (and of BAF). This is important as so far, the model has concentrated on a BAF centric view where incorporation of SS18-SSX into BAF complex changes its composition, conformation and activity. Our study does not invalidate that BAF complex components are dependencies in synovial sarcoma, rather it shows that the SSX tail works as an anchor to H2A-rich regions, and defines a synovial sarcoma gene signature regardless of the fusion partner. Accordingly, SS18-SSX can be recruited in the absence of BAF core components or its catalytic activity. Importantly, the intrinsic property of SSX tail to bind H2AK119ub1 can be exploited by fusion to alternative transcriptional regulators (EWSR1 or MN1). That other transcriptional activators are fused to SSX1 in patient’s tumors and able to activate Polycomb target genes *in vitro*, clearly demonstrates that direct deregulation of BAF complex via introduction of the SSX tail is not essential to all synovial sarcomas. This point is reinforced in the revised manuscript in different ways by showing that:

- Ectopic expression of SS18-SSX and alternative SSX fusions in mesenchymal stem cells (MSCs) result in similar genome-wide transcriptional outputs. (Fig. 3c).
- Alternative SSX fusions interact with transcriptional activators such as EP300 and TBP (tata-binding protein) and result in H3K27ac deposition at H2AK119ub1-rich Barr bodies.
- As suggested by reviewer 2, we used a SMARCA2/4 degrader to show that BAF complexes are not required for: 1) Recruitment of SS18-SSX to H2Aub-rich foci (Fig. 2g-i). 2) The ability of EWSR1 or MN1-SSX fusions to deposit H3K27ac, interact with TBP and activate gene expression of target genes (Fig. 3h-j and Extended Fig. 3).

2. For Figure 1, and related to my comment above, it would be important to evaluate the full-length wild-type SS18-SSX1 and mutant SS18-SSX1 (delta RD), and not just the SSX-C. It's clear in the staining that full length SS18-SSX has a somewhat different binding pattern to SSX alone. This indicates that the SS18 portion of the fusion is likely directing binding of the fusion to chromatin, probably via integration into BAF complexes. Therefore, it remains important to include these proteins in the comparison. Perhaps also including a mutant previously established by the Kadoch lab; where mutation of the residue at position 169 of the fusion protein abrogated H2A binding. This will better delineate the relative contribution of the SSXRD domain to chromatin binding of the SS18-SSX1 fusion oncogene.

We understand the points raised. It should be mentioned that in the old Fig. 1h (now Fig 2 c, d) we show occupancy of endogenous SS18-SSX1 (HA-tagged SS18-SSX in HS-SY-II from Banito et al 2018) side-by-side with SSX-C overexpression in the same cells. A similar experiment to the one suggested –occupancy of SS18-SSX full length or with a deletion of the last 7 amino acids - has been performed before (McBride et al, Fig.2g PMID: 32747783). That experiment showed that deletion of this region results in a chromatin binding pattern similar to that of SS18. The point we want to make here is that SSX-C alone, without being fused to SS18 and therefore being incorporated into the BAF complex, is sufficient to reproduce SS18-SSX binding patterns. Therefore, SS18-SSX occupancy is a result of SSX-C specificity independently of it being fused to SS18 or not. As explained above this is an important point as the notion remains that the SSX tail mainly alters BAF complex conformation by incorporating into the complex via SS18.

We performed two experiments to further show that SS18-SSX chromatin binding is mediated by its SSX tail:

- To uncouple SSX-C binding patterns from SS18 in its own cellular and epigenetic background, we compared the ability of SSX-C to occupy its targets in the presence or absence of a validated shRNA against endogenous SS18-SSX in synovial sarcoma

cells. These results showed that SSX-C occupancy (which phenocopies that of SS18-SSX) remains unchanged upon SS18-SSX knockdown (Fig. 2e and Extended Fig. 2h). Therefore, SSX-C occupancy does not depend on the presence of SS18-SSX (and therefore of altered BAF) at those sites.

- To undoubtably demonstrate that SSX-C terminal occupancy overlaps with that of SS18-SSX, we overexpressed HA-tagged SS18, SS18-SSX1 and SSX1-C constructs side-by-side in SS18-SSX-negative cells osteosarcoma cells and profiled them using Cut&Run (Fig. 2f). While there is some overlap between all constructs, the characteristic broad binding patterns of SS18-SSX to Polycomb target genes was again phenocopied when SSX-C alone was expressed. Importantly, SS18-SSX demonstrated a stronger correlation with SSX-C compared to SS18 (Extended Fig. 2i).

We have performed some experiments using the SSX1 R169E mutant, and have seen that it behaves like the DeltaRD mutant - it loses nuclear localization which is consistent with McBride et al where the SS18-SSX^{R169E} mutant loses the ability to bind nucleosomes. Given the comments from the other reviewers, and the need to re-structure the paper to move experiments related to the McBride study to the supplementary materials, we thought further work on this mutant is not a priority for a revised manuscript.

3. In Figures 1h, it is not clear if the ChIP-seq analyses is performed with spike-in or not. The authors should clarify in the legends whether CHIP and CUT&RUN analyses are spike controlled, as indicated in the methods. This is very important given the large global shifts in the abundance of their targets in these assays. It's well established that large global shifts in the abundance of targets (proteins/modifications) in chromatin mapping studies must be normalised with a reference spike-in. If this is not done, the reliability of the measured differences is questionable.

The ChIP-seq shown in old Fig. 1h (now Fig. 2d) is not calibrated. However, the deltaRD mutant loses the ability to bind chromatin and resembles the empty vector control HA-eGFP (Extended Fig. 2b, c). Here, the differences in ChIP signal are so drastic that the use of calibrated ChIP would not change the conclusions. However, we now repeated the SSX-C overexpression with and without shSS18-SSX in HS-SY-II cells using Cut&Run which is normalized using *E.coli* spike-in as described in the Methods and confirmed that SSX-C phenocopies SS18-SSX chromatin binding profile (Fig. 2e). Cut&Run *E.coli* spike-in was also used to profile SS18-SSX upon PCGF1 removal (Fig. 1f-h). Again, the differences in SS18-SSX signal were quite strong. Of note this last data set was already present in the first version of the manuscript but we re-analysed it using the most recent genome build T2T.

Regarding H2AK119ub1 levels, as the changes are more subtle, we agree that to accurately compare global differences in histone marks a more quantitative method should be used. We performed calibrated ChIP for H2AK119ub1 as described in Fursova et al., 2019 and Fursova et al., 2021 (PMID: 31029541 and PMID: 33888563) and confirmed that PCGF1 knockout in synovial sarcoma cells results in reduced levels of H2AK119ub1 (Fig 1f-i).

4. The authors report that PRC1.1 is essential for depositing macroH2A in chromatin. However, their experimentation to support this is lacking. The experiments performed have utilised sgRNA mediated knockdown/out of PRC1 members which associates with reduced macroH2A. However, it's very important to note that chromatin dynamics and histone deposition is regulated in a highly dynamic manner. Low term genetic knockout of a PRC1 factor and associated reductions in macroH2A abundance cannot be directly linked. On these long experimental time courses, it is possible that these changes are not directly linked and instead correlative. If the authors wish to prove that the two events are linked, they should utilize an experimental approach using for example, targeted protein degradation of PRC1 members. This would allow an experimental manipulation on a time scale relevant to the processes dictating histone deposition and turnover. Without such an experimental approach, it is impossible to conclude that the loss of macroH2A is anything more than a downstream correlation to PRC1 member knockout.

The experiments in question were performed to confirm that SSX-C also binds regions containing macroH2A (as a validation of the mass spectrometry presented in the old Fig. 2a-c, now Extended Fig. 2e). However, we do not think that "PRC1.1 is essential for depositing macroH2A in chromatin". Indeed, the Cut&Run profiling presented in old Figure 2f showed that some domains that are enriched in MacroH2A2 are not occupied by PRC1.1. What we found was that PRC1.1 recruitment (in the absence of PRC2) results in MacroH2A deposition (old Fig 3f and showed by MBD-KDM2B assay). This was surprising given the current literature implicating PRC2 in this process. Although we think this is an interesting result in a well-controlled assay, understanding if MacroH2A deposition is a direct or indirect result of PRC1.1 recruitment is out of the scope of our study. We decided to remove these experiments from the restructured version of the paper as they distract, rather than add, to the main messages of our study.

5. In Figure 4, the authors knockout the vPRC1 component PCGF1 and show that the consequent reduction of H2AK119ub correlates with a reduction of SS18-SSX binding to chromatin, supporting their model. However, not all SS18-SSX is removed. While this might be a consequence of failure to remove all H2AK119ub1 (other vPRC1 complexes can presumably compensate), the authors should perform a more extensive analyses of their CUT & RUN data. For example, they could divide each of their SS18-SSX and H2AK119ub1 datasets into 3-5 categories of changing most, to changing least. This might help answer questions such as: does SS18-SSX remain on sites in which H2AK119ub1 is least changed? Vice versa, is H2AK119ub1 least changed at sites that SS18-SSX is least reduced in PCGF1 KO cells? Are there sites in which SS18-SSX binds that are weakly or not enriched for H2AK119ub, and again, vice versa, are there sites that are enriched for H2AK119ub1 that do not have SS18-SSX binding?

This is a very good suggestion. We performed these analyses and as seen in the figure below

sites displaying a stronger decrease in HA (endogenously tagged SS18-SSX1) upon sgPCGF1 expression (cluster 1) are those that exhibit higher levels of H2AK119ub1 in the control condition and that change the most in the PCGF1 knockout condition (left hand side). The same is true in the reverse analysis (right hand side). Sites with lower H2AK119ub1 changes exhibit low levels of SS18-SSX and show low magnitude differences in SS18-SSX upon PCGF1 knockout (cluster 3). We have also included a similar analysis in the manuscript performed using the H2AK119ub1 cChIP (Fig.1i). The figure shows that regions displaying a stronger decrease in SS18-SSX (cluster 1) are those that also exhibit a stronger decrease in H2AK119ub1. The regions where SS18-SSX changes the least and where some of it remains (cluster 3) are those where H2AK119ub1 is least changed. It is important to note that SS18-SSX heatmaps have changed slightly due to the re-analysis in the new genome build T2T.

Log2 HA = Log2 fold change between EV and sgPCGF1 for SS18-SSX Log2 H2Aub = Log2 fold change between EV and sgPCGF1 for H2Aub

Cluster 1 = strong decrease, Log2 below 1

Cluster 2 = intermediate decrease, Log2 between -0.25 and -0.5

Cluster 3 = weak decrease, Log2 negative but not stronger than -0.25

Cluster 1 = strong decrease, Log2 below 1

Cluster 2 = intermediate decrease, Log2 between -0.25 and -0.75

Cluster 3 = weak decrease, Log2 negative but not stronger than -0.25

Regarding the comment about incomplete removal of SS18-SSX. We should point out that even when we induce depletion of SS18-SSX itself (e.g., using an shRNA) there is still some SS18-SSX signal (see for example Banito et al 2018, PMID: 29502955, Fig 7C). So, we were not very surprised by the incomplete removal of SS18-SSX upon sgPCGF1 expression. Additionally, these are not clonal populations where all cells have a specific PCGF1 knockout mutation. These cells are transduced with sgPCGF1, selected with puromycin and evaluated by Cut&Run approximately 8 days later. sgPCGF1 cells cannot be expanded as clonal populations as they stop proliferating.

6. The authors report that high H2AK119ub1 levels are a feature of synovial sarcomas, but is it important? Does knockout of PCGF1 slow the proliferation of synovial sarcomas?

Yes, PCGF1 depletion using shRNAs or guide RNAs impacts proliferation of synovial sarcoma cell lines (Banito et al 2018 PMID: 29502955, see Fig4C-F). This is also shown in the current manuscript in Extended Fig. 1k.

Minor:

1. Figure 1e lacks the necessary controls to confirm equal loading/extraction. They should also blot for other 1-2 endogenous chromatin proteins.

Since the paper was restructured to move this figure to supplementary material, we thought that repeating the experiment to blot for control proteins was not a priority for a revised manuscript. However, we can repeat such experiments if this is still a point the reviewer considers important. Of note, the protein quantity in each fraction is internally normalized in each condition, and indeed in other publications (see for example in McBride et al Fig1 d) a loading control is not used.

Reviewer #2 (Remarks to the Author):

Benabdallah and colleagues investigate the role of PRC1.1 in SS18-SSX driven synovial sarcoma. They perform a gene-tiling screen with a library of guideRNAs against the SS18-SSX fusion and demonstrate that the RD domain of SSX is required and sufficient for high affinity chromatin binding. They then utilize IP-MS of overexpressed eGFP compared to SSX-Cterminus-eGFP and SSX-C terminus-eGFP compared to SSX-C-E184*-eGFP (identified in McBride et al NSMB 2020) to identify association between the SSX-C terminus and histones, as well as MacrohistoneH2A1 and MacrohistoneH2A2. Referencing prior work from the Kadoch lab (McBride et al NSMB 2020), they compare the overlap of SS18-SSX with H2AK119Ub and find using BRET that the SSX RD domain and terminal acidic domain (E184*) are required for association with H2AK119Ub and MacroH2A1/A2.

They make use of a synthetic biology strategy to retarget PRC1.1 component KDM2B to methylated CpG sites by fusing it with an MBD domain and deleting the CXXC domain in KDM2B. Using this approach, they find that SS18-SSX is recruited to MBD-KDM2B foci and the colocalization of these proteins is dependent on PRC1.1 subunits BCOR and PCGF1, but not MacroH2A. On chromatin, they find that knockdown of PCGF1 results in global reduction of H2AK119Ub and SS18-SSX binding, which correlates with reduced affinity for chromatin (not found for knockdown of PCGF3). They further find that overexpression of SS18-SSX in mesenchymal stem cells leads to transcriptional upregulation of BCOR and PCGF1 as well as increased H2AK119Ub, while knockdown of SSX or PCGF1 in SS cells reduced H2AK119Ub and BCOR, PCGF1. Overexpression of SSX-C alone in SS cells results in more BCOR, SS18, and H2AK119Ub and increased affinity of BCOR/PCGF1 for chromatin. Finally, an animal model of SS shows kinetic increase in H2AK119Ub with tumorigenic

progression and human SS samples show increased H2AK119Ub relative to other sarcomas or normal tissue.

This study is a nice follow up of previous work from Banito and colleagues on the dependence of SS on KDM2B and other subunits of the PRC1.1 complex primarily through KDM2B-dependent recruitment of SS18-SSX to PRC1.1 binding sites (Banito Cancer Cell 2018). However, the work is largely incremental in light of work from Kadoch and colleagues identifying the RD domain of SSX and indeed the exact residues that encode high affinity binding of SSX to histones, H2AK119Ub, and chromatin (by ChIP) (McBride NSMB 2020). Thus, most of Figure 1 and Figure 2 is not new information (the MacroH2A results are new, but appear to have a relatively small contribution to the mechanism of SS18-SSX oncogenesis). This work essentially ties the knot between these prior findings showing that PRC1.1 recruits SS18-SSX through the deposition of H2AK119Ub, which is perhaps not surprising given the known role of PRC1.1 in depositing H2AK119Ub.

Below are a few comments that would strengthen the novel aspects of the study.

We understand the concerns regarding novelty. Many of the experiments identifying the interaction of the SSXRD with histones and H2AK119ub1 have been performed prior to the publication by McBride et al using overlapping but yet complimentary experiments. Still, we agree that our manuscript should concentrate on further supporting the conceptual advances to clearly differentiate it from previous literature. What we show is that it is the SSX tail, via PRC1.1-deposited H2AK119ub1 that determines fusion specificity regardless of BAF complex deregulation. This is important as so far, the model has focused on a BAF centric view where incorporation of SS18-SSX into BAF complex changes its composition, conformation and activity. Our study does not invalidate that BAF complex components are key dependencies in synovial sarcoma, rather it shows that the SSX tail works as an anchor to H2A-rich regions, and defines a synovial sarcoma gene signature regardless of the fusion partner. Our data support a model where it is not de-regulation of BAF complex composition or conformation (eg SMARCB1 loss) that determines SS18-SSX occupancy but rather an intrinsic property of SSX proteins that can be exploited by fusion to alternative transcriptional regulators (EWSR1 or MN1). That other transcriptional activators are fused to SSX1 in patient's tumors and able to activate Polycomb target genes *in vitro*, clearly demonstrates that direct deregulation of BAF complex via introduction of the SSX tail is not essential to all synovial sarcomas.

To emphasize these novel points, and following the suggestion of reviewer 3, we restructured the manuscript to move data presented in old Figure 1 and 2 to supplementary material. Many of the experiments suggested here, which we think were excellent, were added in new main Figures 2 and 3. We think they really strengthen the main conclusions of the paper, specifically in showing that SS18-SSX can be recruited in the absence of BAF core components or its catalytic activity. This point is reinforced in the revised manuscript in different ways and mostly by experiments suggested in the review.

1) The authors show that PCGF1 is responsible for much of the H2AK119Ub and increased affinity for chromatin, and not PCGF3 (Figure 4d,e). This is important because CERES dependency scores implicate PCGF3 and PCGF5, in addition to KDM2B. It would be good to include confirmation of PCGF3 knockdown in Figure 4d and H2AK119Ub C&R for sgPCGF3, as well as SS18-SSX salt extraction and H2AK119Ub C&R for knockdown of PCGF5. It seems possible that multiple complexes could contribute H2AK119Ub and despite global reduction in H2AK119Ub in sgPCGF1 cells, some H2AK119Ub remains, which could be dependent on other PRC1 variants. Do PCGF1/3/5 double/triple knockouts have even greater reduction in H2AK119Ub, MacroH2A, and SS18-SSX?

This is an excellent point. We have compared the requirement of different PRC1 complexes across synovial sarcoma cell lines by targeting the various PCGF genes using sgRNAs. We observed that sgRNAs against PCGF1 or PCGF3 greatly impacted synovial sarcoma cell proliferation but not those against other PCGFs. Interestingly PCGF5 sgRNAs had no impact on cell proliferation or in transcriptional changes by RNA-seq and indeed PCGF5 levels are very low across several synovial sarcoma lines. This could, to some extent, explain the specific dependency on PCGF3 since they appear to be interchangeable in PRC1.3/PRC1.5 complexes.

Most importantly, we cannot detect a loss of H2AK119ub1 at fusion bound sites or loss of SS18-SSX chromatin binding genome-wide in response to PCGF3 knockout (as we see upon PCGF1 KO in the current manuscript). Clearly, PRC1.3 is a strong and specific dependency in these cells however via another gene regulatory mechanism. Experiments we are currently performing suggest that rather than mediating SS18-SSX fusion recruitment, PRC1.3 is required for the fine-tuning of SS18-SSX target genes. These experiments are still underway and not completed. Since the main message we want to emphasize here is that knockout of PCGF1 alone is sufficient to impact H2AK119ub1 levels at SS18-SSX bound regions and consequently SS18-SSX chromatin binding, we think that adding more experiments on a possible alternative role for PRC1.3 (PCGF3) will dilute the main message of this manuscript. Moreover, we think the differential role of PCGF3 merits a more in-depth characterization which we intend to pursue in more detail in a subsequent manuscript.

Of note we have added the effect of sgPCGF1 and of sgPCGF3 on cell proliferation (in HS-SY-II cells) and corresponding TIDE analysis to show efficient CRISPR/Cas9 editing in Extended Fig. 1j, k. These results clearly show that although PCGF3 does not deplete SS18-SSX from the chromatin fraction in these cells (Fig 1j), it has a great impact on cell proliferation.

2) What I found potentially very interesting was the sufficiency of SSX alone to bind chromatin (1g, h). The statement by the authors, “These results indicate that the SSX c-terminus, via its SSXRD, binds specific regions in the genome and determines SS18-SSX localisation independently of SS18 and therefore of the mSWI/SNF complex”. However, as far as I can tell, these constructs are being overexpressed in an SS cell

line which already expresses SS18-SSX. This also is the case for Figure 6a-e. It's therefore hard to know whether SSX-C activity is in some part influenced by the expression of the fusion and the effect it has already had on chromatin. I wonder if SSX-C is simply going to where the fusion is and acting like an amplification event. Indeed, SSX-C increases SS18 expression. To be able to make this statement (which indeed is quite controversial), I would suggest the authors overexpress SSX-C and SSX-C-delta RD in mesenchymal stem cells and then perform ChIP and immunofluorescence for BCOR, H2AK119Ub, SS18 as in Figure 1g and Figure 6a-e. Furthermore, to definitively make this statement, the authors need to show that SSX-C binding to chromatin by ChIP is not diminished by knockdown of SMARCA4, or treatment with SMARCA4/A2 degraders.

We agree that this is one of the points that should be emphasized and one we think is quite important. We performed several additional experiments that we think address these questions:

- To uncouple SSX-C binding patterns from SS18 in its own cellular and epigenetic background we compared the ability of SSX-C to occupy its targets in the presence or absence of a validated shRNA against endogenous SS18-SSX in synovial sarcoma cells. These results showed that SSX-C occupancy (which phenocopies that of SS18-SSX) remains unchanged upon SS18-SSX knockdown (Fig. 2e and Extended Fig. 2h). These results show that SSX-C occupancy does not depend on the presence of SS18-SSX (and therefore of altered BAF) at those sites.
- To undoubtedly demonstrate that SSX-C terminal occupancy overlaps with that of SS18-SSX, and not with SS18 WT, we overexpressed the three constructs side-by-side in SS18-SSX-negative cells and profile their occupancy in SS18-SSX osteosarcoma cells (Fig. 2f). The characteristic broad binding patterns of SS18-SSX to Polycomb target genes was again phenocopied when SSX-C alone was expressed.
- To show that recruitment of SS18-SSX is independent of BAF (or the "effect it has already had on chromatin") we used SMARCA2/4 degraders as suggested. Here we took advantage of SS18-SSX recruitment to the Barr bodies in HEK293T cells as well as the MBD-KDM2B assay in HS-SY-II cells which allows us to evaluate *de novo* recruitment of SS18-SSX to H2Aub-rich chromatin. Here, treatment with SMARCA2/4 degrader (ACBI1) abolished recruitment of SMARCC1 BAF complex core subunit to H2Aub-rich regions whilst *de novo* SS18-SSX recruitment to these regions was unaffected by the absence of the BAF complex (Fig. 2g-i). Importantly, ACBI1 treatment specifically resulted in depletion of SMARCA4 without affecting the total levels of SMARCC1 (Extended Fig 2j).
- We performed BCOR and H2AK119ub IF staining upon SSX-C expression in MSC which was already present in the old version in Extended Figure 6a. We have now

extended this analysis with RNAseq for SSX-C and SSX fusions in MSCs. Here we clearly see that SSX-C alone does activate BCOR mRNA levels (Fig. 4a), but results in higher BCOR and H2AK119ub1 levels (Fig. 4d-f). We have tried CUT&RUN in MSCs and did not obtain so far a good quality results with these cells. Additionally, to evaluate H2AK119ub levels would necessitate calibrated ChIP, which requires a larger number of cells and is difficult to achieve with SSX-overexpressing MSCs.

3) Related, I found the activity / sufficiency of SSX to induce transcription of genes when fused to EWSR1 or MN1 (Figure 1i, j) very interesting. There data suggest a new model for thinking about the molecular basis for this disease, especially in light of the identification of novel SSX fusions in SS. Using the EWSR1-SSX and MN1-SSX fusions in subsequent assays would further underscore the authors proposed model for an autoregulatory look – for example 1) showing EWSR1-SSX and MN1-SSX are recruited to MBD-KDM2B foci in a PCGF1-dependent mechanism and 2) showing increase BCOR and PCGF1 protein, H2AK119Ub levels in EWSR1-SSX and MN1-SSX expressing cells.

We agree that further exploring and comparing the activity of the new fusions is very interesting and shows that direct BAF deregulation is not an absolute requirement in synovial sarcoma. The following experiments further support model:

- RNAseq analysis of ectopic expression of alternative SSX fusions in mesenchymal stem cells (MSCs). This analysis shows that they activate similar expression signatures to SS18-SSX1 (but not EWSR1-FLI1 or SS18-NEDD4) (Fig.3c) and clearly supports a shared molecular basis for the three oncofusions containing an SSX tail.
- Like SS18-SSX1, EWSR1- and MN1-SSX1 are recruited to H2AK119ub1-rich Barr bodies in a manner that is independent on the presence of the BAF complex (Fig. 3d and Fig. 3j).
- EWSR1- and MN1-SSX1 engage with EP300 and TBP transcriptional activators and results in H3K27Ac in a manner that is independent of BAF complex catalytic activity (Fig. 3 e-j)
- Like SS18-SSX1, EWSR1- or MN1-SSX1 fusions result in elevated BCOR mRNA and protein levels in MSCs as well as increased H2AK119ub1 (Fig. 4a-f).

Reviewer #3 (Remarks to the Author):

This manuscript addresses the altered chromatin regulatory mechanisms arising from translocations of the SWI/SNF subunit SS18, which yield the hallmark SS18-SSX fusion proteins that drive synovial sarcoma. The authors' results are high quality, well-designed, and deepen the understanding of the broader chromatin regulatory systems that operate in synovial sarcoma. The most prominent issue that the authors need to

correct is the overlap with major portions of work that was previously published on a similar topic (McBride et al., Nat Struct Mol Biol 2020, PMID 32747783). For example, in the McBride paper, the connections between the C-terminus of SS18-SSX and binding of H2AK119ub1 and PRC1 were already established and hence the novelty of these specific aspects in the current manuscript are weakened.

Despite the overlap with portions of the McBride paper, the current work does reveal a number of novel findings not found in previous works, including the existence and characterization of the feedback loop between SS18-SSX, H2AK119ub1, and PRC1.1, and observations about the role for the histone variant macroH2A. As a result, the authors' findings do extend the links between SSX and H2AK119ub1 beyond previous findings, including the lack of PRC2 involvement. As a result, the work is meritorious, but should be restructured to focus on these novel aspects, and I would suggest focusing on the mechanisms underlying the feedback loop. Authors should take care to further distinguish the work from the McBride paper.

We appreciate the positive comments and the suggestions which we followed as explained below.

Major concerns:

1. Portions of Figures 1 and 2 in the current work somewhat lack novelty as the result of Figure 4 and other results from McBride et al. The current work needs restructuring to highlight the novel aspects of the work and should move all the findings that strictly confirm McBride et al. out of the main figures to the supplement or remove them in order to highlight the novel aspects of their findings. I believe there are important novel findings in the current work that, if refocused and refined, would sufficiently differentiate it.

We agree that the paper should be restructured. We have moved portions of Figure 1 and 2 to supplementary. Novel aspects of the study, including many experiments related with points raised by the review is now presented mostly in Fig 2, 3 and 4.

2. The autoregulatory feedback loop is perhaps the most novel aspect of this work, and authors should deepen some of the mechanisms underlying this feedback loop. For example, can the authors distinguish between increased expression (i.e. up-regulation of transcription/translation) of SS18 (and other SWI/SNF subunits), BCOR, and RYBP (and other PRC1.1 subunits) in contrast to increased stability/accumulation of these proteins? Analysis of RNA levels or proteasome inhibition studies could be helpful to reveal the mechanistic basis for accumulation of these proteins, and may potentially provide stronger evidence for a mechanism mediated by altered transcription.

We took advantage of a well characterized inducible SS18-SSX shRNA to better dissect these changes in PRC1.1 mRNA or protein levels. As you can see in Figure 4 b-c and Extended Fig 4c, the fusion is required to maintain high protein levels of PRC1.1 components. Downregulation of PRC1.1-specific subunits (KDM2B, PCGF1 and BCOR) is detected at the protein level as early as 12 hours after doxycycline administration while at this point only the BCOR mRNA levels are affected. Additionally, and probably the most important piece of data that argues in favor of regulation at the protein level, is that expression of the SSX-C alone (eGFP-SSX-C) does not induce BCOR mRNA levels in MSCs (Fig. 4a) but results in higher BCOR protein levels and increased H2AK119ub1 (Fig 4d, e). Likewise, SSX-C does not affect mRNA levels of SS18 (Extended Fig. 4i) and does not bind the SS18 locus (Extended Fig. 4j). These data support the notion that the rise in H2Aub levels by SSX-C expression, results in increased SS18-SSX protein levels (Fig. 4h) and not its expression.

3. Given that SWI/SNF ATPase activity is widely recognized to oppose Polycomb activity in most contexts (including PRC1; see e.g. PMIDs 27941795, 20951942, and many others), it would be somewhat surprising that targeting of SWI/SNF activity would promote Polycomb accumulation. How does SWI/SNF ATPase activity influence the accumulation of PRC1.1/H2AK119ub1 at SS18-SSX sites in this setting? To characterize the mechanisms of the feedback loop, authors could perform SWI/SNF inhibitor studies to assess the influence of SWI/SNF ATPase activity both on the level of survival/proliferation as well as PRC1.1/H2AK119ub1 (e.g. using ChIP-qPCR at a given target site).

It is counterintuitive that SS18-SSX promotes H2AK119ub1 and PRC1.1 whilst also recruiting BAF, which opposes Polycomb and induces transcription. This is a very good point but one that it is difficult to disentangle. While mSWI/SNF's ATPase activity recruited to Polycomb sites may to some extent oppose H2AK119ub1 deposition (as it does for PRC2 and H3K27me3), SSX-C is also present at the same sites where it stabilizes PRC1.1/H2AK119ub levels. Therefore, it is possible that SS18-SSX target sites are more resistant to PRC1.1 eviction, while PRC2 and H3K27me3 are efficiently opposed. That these sites retained high levels of H2AK119ub1 but are expressed, indicate that BAF complex recruitment (or other transcriptional activators) overrides H2AK119ub1's potential in controlling transcription bursting (PMID: 34608337).

Despite the SSX-C tail rendering SS18-SSX target sites more resistant to PRC1.1 eviction, one would expect that the BAF complex retains some ability to oppose PRC1 activity. In fact, when we overexpressed SSX-C or SS18-SSX in mesenchymal stem cells the increase in H2AK119ub1 was greater when SSX-C alone was expressed (Figure 4f). This suggests that SS18-SSX may have a lower potential than SSX-C to increase H2AK119ub1, possibly due to competition between BAF and PRC2/PRC1. However, further testing this mechanism is challenging as SMARCA2/4 removal will reinstate PRC2 which in turn will recruit canonical PRC1 thereby depositing H2AK119ub1.

If the reviewer thinks this is still an important experiment we can perform the ChIP-qPCR, however because of the complex scenario we are unsure if it will provide a definitive answer for this particular question.

4. The macroH2A knockout changes in Ext Figure 3 are modest, as the authors acknowledge. Yet the authors state “a very specific chromatin environment containing both H2AK119ub1-modified nucleosomes and histone variant MacroH2A underlies SSX C-terminus chromatin binding”; authors should perform a more conclusive experimental test to differentiate whether macroH2A is instructive vs. reflective of SS18-SSX targeting.

We agree. As we discussed and in response to a previous comment by reviewer 1 we performed the MacroH2A experiments to validate the results from the mass spec presented in old Figure 2a-c. While it seems that SS18-SSX is able to occupy regions decorated by MacroH2A this is not a key for SS18-SSX recruitment as knock out of both MacroH2A isoforms had a very mild effect in cell proliferation (old Extended Figure 3g) and did not impact SS18-SSX recruitment to MDB-KDM2B foci (old Figure 3f). We decided to remove these results and instead concentrate on the most novel and important aspects.

5. The authors' in vivo analysis in Figure 6 and 7 show strong support for tight links between H2AK119ub1 and SS18-SSX, which goes beyond previous findings to implicate H2AK119ub1 as a key mechanistic feature of synovial sarcoma. But in my view, the findings are consistent with but are not explicitly constrained to an autoregulatory model. Instead, they are consistent with any potential model where H2AK119ub1 accumulates upon development of synovial sarcoma. Is there additional data that would strengthen the autoregulatory finding in vivo? For example, are other features of PRC1.1 increased in these tumors in vivo as seen in vitro? Alternatively, if the samples in Figure 7f were processed identically, and if an autoregulatory mechanism were at play, one would expect these intensity values to be correlated. Testing for a significant correlation between SS18-SSX and H2AK119ub1 would be strong support of their in vitro mechanisms, and go beyond the discovery that both H2AK119ub1 and SS18-SSX arise concomitantly in the malignancy. There may be other possibilities to more directly test the feedback of SS18-SSX fusions with H2AK119ub1 beyond their shared presence in synovial sarcoma tumors.

This is a very good point that we tried to address before but had struggled with finding a BCOR antibody that worked in mouse sections. We have tested several antibodies against mouse BCOR to measure it in parallel to accumulation of H2A119ub in response to SS18-SSX expression *in vivo* and have now succeed. The staining was performed in samples from three different mice per time point and clearly shows increased BCOR staining early in tumorigenesis (Fig. 5d, e).

As suggested, we also determined if there was a correlation between SS18-SSX or SSX staining and H2AK119ub1 levels in patient samples. As presented in Extended Fig. 5d, there is a positive correlation between H2Aub and SS18-SSX average signals in each sample measured using an antibody against the SS18-SSX fusion or an antibody against SSX proteins. The correlation is not very high however these comparisons are challenging as the stainings were not performed in consecutive sections and not all cells within the tumor exhibit uniform SS18-SSX or H2AK119ub1 signal.

Please also note that, since the concept of an autoregulatory SS18-SSX/PRC1.1 feedback loop is difficult to achieve experimentally, we also changed the text and the title of the manuscript to toned down the auto-regulatory angle. Instead, we focused of the ability of SSX-C to increase PRC1.1 stability resulting in higher H2A ubiquitination levels which in turn further enable oncofusion binding.

Minor concerns:

1. The Fig 1e results of the bands are quite variable and the results for SS18-SSX1 do not appear to agree with the quantification shown in Fig 1f (e.g. 300 mM measurement). Why is this? Interpretation would benefit from a more consistent results, especially for SS18-SSX1.

There is some variability across the eGFP constructs regarding total protein levels expressed in HEK293T. This is partially a result of different protein stabilities and transfection efficiencies. Additionally, there is some variability in this assay across the fractions but we think it is clear that the eGFP-SS18-SSX, eGFP-SSX-C and the eGFP-SSXRD are enriched at the chromatin fraction while the eGFP-deltaRD are enriched in the soluble fractions. Regarding variability across constructs, the values presented in old Figure 1f (now Extended Fig 2) correspond to % of fraction at each salt concentration per condition (construct). So, they are internally normalized to total levels of protein per construct. This was calculated by dividing the value corresponding to intensity of the band per fraction for each construct, by the sum of the intensity of all bands across the different salt concentrations fractions for that construct. It should also be noted the quantification presented in Fig1f corresponds to the average of two independent experiments, one of them being the one presented in Fig1e. This also accounts for the fact that the quantification does not perfectly match the image shown in Figure 1e.

2. Fig 1g color signal and Fig 1h read depth metrics should be explicitly stated to ensure apples-to-apples comparisons.

The ChIP-seq for HA-SS18-SSX1 and KDM2B in HSSY-II cells (previously published, Banito et al 2018), and ChIP for HA-eGFP in HSSY-II cells is normalized using inputs. As we use mainly this figure to describe qualitatively SSX-C profile and not quantitatively, we think that normalizing to read depth may not matter as much as we normalized using input. Moreover,

as these different ChIPs where done using different antibodies, or endogenous vs construct overexpression, they will exhibit variable yields. For the ones performed side by side for the same target (HA-eGFP ChIP) the metrics are the same across the different conditions.

Of note, the other ChIP and Cut&Run sequencing present in the manuscript that we use for quantitative conclusions were further normalized using spike-ins.

3. Labeling in Fig 6d-e and potentially other figures could be improved to indicate that the constructs are EGFP fusions.

We agree and updated the labeling.

4. I do not understand the image pairing in Fig 7c; are these replicates? How many times was this performed? Do all targets contain the SSM2 construct? The analysis in Fig 7d suggests that samples without the SSM2 construct were also analyzed, and these should be presented as controls in the main figure.

The two images in the previous version corresponded to two representative images per time point, we appreciate that this was unnecessarily creating confusion and we selected only one image to represent the data.

The original data at 16 weeks in old Extended Fig. 7 (now Extended Fig 5a-c) was from 3 biological replicates but not the data representing the time-course in Fig 7c. We agree that the analysis should be performed from additional individual mice. We repeated the staining in additional samples from individual mice, and the data presented in Fig.5c-e corresponds now to three biological replicates (three mice per time point).

In all experiments presented the mice carry a SSM2 allele. However, in old Extended Fig. 7 a-c (now Extended Fig 5a-c) a control consisting in mice that were not treated with tamoxifen, and therefore do not express SS18-SSX in the muscle was additionally included. These samples are labeled -TAM. For the remaining samples labelled +TAM the SSM2+ and SSM2- cells correspond to SS18-SSX positive and negative cells within the same image (Extended Fig. 5a-c and Fig. 5c-d). These cells are labeled by GFP staining because after tamoxifen treatment the SS18-SSX positive cells also express a GFP reporter (see Fig. 5a). We adjusted the figures to make it clear.

5. Fig 7f-d and all other summary figures should have N values and it should be made clear how many different animals these studies were performed in.

We agree and we added all N numbers in the figure legends.

6. The summary figure in Figure 8 is confusing since the title describes an autoregulatory feedback loop, but little detail is provided for the feedback loop, and little information about the feedback loop is presented, other than the loop being depicted as “existing.” I suspect other readers will be confused by the message of this image, and this summary image should be clarified and if authors focus future work on

this loop, this figure should be improved to focus on the newly discovered features of the autoregulatory loop.

We agree and changed the title as the autoregulatory feedback loop model is challenging to undeniably prove experimentally. We think that the fact that the SSX-C alone stabilizes BCOR, and H2AK119ub1 in MSCs and in synovial sarcoma results in higher SS18-SSX levels support a feedback mechanism and a central role for the SSX-C tail. We simplified the model (Fig. 6) to emphasize that the SSX tail (red) binds to and promotes H2AK119ub deposition via PRC1.1 (purple). When fused to a transcriptional activator (blue) (SS18-BAF or other transcriptional regulator) it promotes aberrant gene activation of Polycomb target genes. We hope this simplified version is clearer and more representative of our findings.

7. Despite having read the Methods section describing it, the frequent use of “fluorescence ratio in high vs. low GFP” measurements is confusing and could be better explained or labeled. Authors should provide a supplemental figure and characterize for readers what high- and low-GFP means in this setting. Are the authors relying on variable expression levels within each condition to make these measurements? If so, authors should make clear that the EGFP intensity distribution is controlled for between constructs or provide a clearer rationale for the use of this analytical approach.

We apologize as this is not well explained in the methods. Yes, we are relying on GFP expression levels within each condition, to directly compare the staining signal of H2AK119ub or BCOR side by side in the same image, in GFP negative (or very low expressing cells) vs GFP-positive cells. This is possible because the constructs are EGFP fusions or are linked to GFP expression (in the case of the sgRNAs). The calculation was done by first measuring the mean nuclear intensity per nucleus in each channel: DAPI (405), green channel (488 nm), red channel (594 nm) and far read (647nm). Then, all individual nuclear intensities were normalized by dividing by their corresponding DAPI mean nuclear intensity value, which should be constant across different nuclei. eGFP high and eGFP low (or negative) cell populations were distinguished based on a threshold of normalized eGFP (488nm) intensity of >1 or <1 respectively. The ratio between the average of the mean nuclear intensity for the red channel or for the far-red channel (depending on the fluorophore used for detecting H2AK119ub, BCOR and SS18) in the each GFP population was plotted as shown in Extended Fig. 6.

Decision Letter, first revision:

Message: Our ref: NSMB-A46664A-Z

7th Jul 2023

Dear Dr. Banito,

Thank you for submitting your revised manuscript "Aberrant gene activation in synovial sarcoma relies on SSX specificity and increased PRC1.1 stability" (NSMB-A46664A-Z). It has now been seen by the original referees and their comments are below. The reviewers find that the paper has improved in revision, and therefore we'll be happy to in principle publish it in Nature Structural & Molecular Biology, pending revisions to satisfy the final requests and concerns of reviewer #3 (specific guidance as to how will be included in our follow-up detailed checks), and to comply with our editorial and formatting guidelines.

We are now performing detailed checks on your paper and will send you a checklist detailing our editorial and formatting requirements in about 7-10 days. Please do not upload the final materials and make any revisions until you receive this additional information from us.

To facilitate our work at this stage, it is important that we have a copy of the main text as a word file. If you could please send along a word version of this file as soon as possible, we would greatly appreciate it; please make sure to copy the NSMB account (cc'ed above).

Sincerely,

Dimitris Typas
Associate Editor
Nature Structural & Molecular Biology
ORCID: 0000-0002-8737-1319

Reviewer #1 (Remarks to the Author):

The authors have satisfactorily addressed all of my comments and suggestions.

Reviewer #2 (Remarks to the Author):

The authors have addressed my concern about novelty and added a considerable amount of new data directly addressing my curiosities. In my opinion, they responded to the reviewers' shared criticism concerning novelty well. They restructured the manuscript and provide data supporting a new model for SSX driven oncogenesis demonstrating BAF-independent sufficiency of SSX to bind H2AK119ub and drive transcription when fused to

SS18, EWSR1, or MN1. They also provide evidence for both SSX-dependent transcriptional and protein 'stabilization' of PRC1.1 components, which they propose induces a positive feedback loop. Although the mechanism by which SSX stabilizes PRC1.1 on chromatin is unclear, that is beyond the scope of this study.

Reviewer #3 (Remarks to the Author):

SUMMARY

This updated manuscript by Benabdallah and colleagues is an interesting improvement over the first submission, and it has successfully identified an interesting BAF-independent story related to the C-terminus of SSX1 in synovial sarcoma. However, some issues reduce my overall enthusiasm about it.

Strengths

- intriguing and exciting story that uncovers a BAF-independent role for the SSX C-terminus
- provocative discovery of SSX fusions that transform cells via presumably BAF-independent roles
- a topic of wide interest to many and an important cancer that currently lacks good therapeutic options
- thoughtful integration of many different types of data
- attractive figures
- analysis of human specimens

Weaknesses

- over-reliance on engineered/tagged, ectopic (over?-)expression of factors and simple colocalization using standard microscopy in a cell line
- largely conventional approaches
- reduced focus on endogenous machinery
- extensive and dense figures that can be difficult to follow to arrive at the authors' conclusions

Overall I feel the work has clearly arrived at a more novel conceptual finding, but as a paper, it is difficult to read, in part due to the complex figures. Additionally, I am less enthusiastic about the extensive use of tagged (over?-) expressed constructs to draw the conclusions, since these factors are unlikely to be at physiological levels or to undergo physiological regulation.

MAJOR CONCERNS

(1) I think it is important to see the authors' results validated using endogenous regulators rather than ectopically expressed tagged constructs.

(2) I am surprised about the authors' use of EWSR1-SSX to highlight BAF-independent roles, since it is known to directly interact with BAF. I am not sure I follow the authors' logic about whether this is truly BAF-independent. Along these lines, are the synovial sarcomas with non-SS18-SSX fusions insensitive to degradation of SMARCA4/2 or BRD9?

(3) The effects of the SSX C-terminus alone towards chromatin localization appear robust.

The mechanisms underlying this observation alone feel to me worthy of a visible publication. However, the increased stabilization of H2AK119ub1 is very modest and, despite one IF image showing a few cells in Figure 4, the effect appears inconsistent across cell contexts, and is not especially well quantified in the current manuscript. This stands in contrast to the strong, binary changes of H2AK119ub1 seen between normal and synovial sarcoma samples (e.g. Figure 5F). Hence stabilization of H2AK119ub1 is not the strongest aspect of the work. My suggestion would be to focus on the chromatin targeting, and perhaps highlight this in the manuscript title, which to me currently feels a little unfocused.

(4) More thought should be given to the clarity of the figure titles. For example, Figure 1 states that PRC1 and not PRC2 deposits H2AK119ub1 independent of PRC2 — it is already well-established that PRC1 deposits this mark, so it is a little unclear to me what truly novel point or contrast that the authors wish to make.

(5) In general, the model that the authors propose in Figure 6 is appealing, however the generality of the message about these activity being BAF-independent is undercut by the use of EWSR1-SSX, as stated above. If the authors wish to claim this as a general model, perhaps they could consider using synthetic transcriptional activator fusions (e.g. VP16, VP64, etc.), to demonstrate entirely synthetic recapitulation of the transforming activity independent of the natural fusions observed in disease.

(6) I also appreciate the message that there are BAF-independent roles of the SSX C-terminus, which I think is really interesting. SSX-C may stabilize H2AK119ub1, or do a lot of things regardless of BAF, and perhaps BAF is just one of many such transcriptional activators that could transform cells when linked to SSX-C. I think these is may be very important for the field to know. Still, I am left wondering whether these observations matter if H2AK119ub1 is insufficient to transform on its own and if SWI/SNF activity is the dominant transforming activity in synovial sarcoma. Greater clarity on this would increase the impact of the story.

Author Rebuttal, first revision:

NSMB-A46664A-Z: Response to reviewer's comments

Reviewer #1 (Remarks to the Author):

The authors have satisfactorily addressed all of my comments and suggestions.

Reviewer #2 (Remarks to the Author):

The authors have addressed my concern about novelty and added a considerable amount of new data directly addressing my curiosities. In my opinion, they responded to the reviewers' shared criticism concerning novelty well. They restructured the manuscript and provide data supporting a new model for SSX driven oncogenesis demonstrating BAF-independent sufficiency of SSX to bind H2AK119ub and drive transcription when fused to SS18, EWSR1, or MN1. They also provide evidence for both SSX-dependent transcriptional and protein 'stabilization' of PRC1.1 components, which they propose induces a positive feedback loop. Although the mechanism by which SSX stabilizes PRC1.1 on chromatin is unclear, that is beyond the scope of this study.

We thank both reviewers for the time reviewing our manuscripts and for the suggestions that clearly improved it.

Reviewer #3 (Remarks to the Author):

This updated manuscript by Benabdallah and colleagues is an interesting improvement over the first submission, and it has successfully identified an interesting BAF-independent story related to the C-terminus of SSX1 in synovial sarcoma. However, some issues reduce my overall enthusiasm about it.

Strengths

- intriguing and exciting story that uncovers a BAF-independent role for the SSX C-terminus
- provocative discovery of SSX fusions that transform cells via presumably BAF-independent roles
- a topic of wide interest to many and an important cancer that currently lacks good therapeutic options
- thoughtful integration of many different types of data
- attractive figures
- analysis of human specimens

Weaknesses

- over-reliance on engineered/tagged, ectopic (over?-)expression of factors and simple colocalization using standard microscopy in a cell line
- largely conventional approaches
- reduced focus on endogenous machinery
- extensive and dense figures that can be difficult to follow to arrive at the authors' conclusions

Overall I feel the work has clearly arrived at a more novel conceptual finding, but as a paper, it is difficult to read, in part due to the complex figures. Additionally, I am less enthusiastic about the extensive use of tagged (over?-) expressed constructs to draw the conclusions, since these factors are unlikely to be at physiological levels or to undergo physiological regulation.

MAJOR CONCERNS

(1) I think it is important to see the authors' results validated using endogenous regulators rather than ectopically expressed tagged constructs.

We have used endogenous factors whenever possible. For example, we determined differences in H2AK119ub1 and SS18-SSX1/2 occupancy upon knockout of PCGF1 in synovial sarcoma lines (Fig1f-i and Extended Fig1g-h). Of note, ChIP for HA-SS18-SSX1 in Fig.1f-i corresponds to an endogenously tagged SS18-SSX where the HA tag was inserted at the endogenous locus using CRISPR/Cas9 (Banito et al. 2018) (not exogenously overexpressed). We have used constructs for exogenous expression for other experiments where we expressed mutant versions of the protein or specific domains for functional studies (e.g. SSX-C or SSX-C^{deltaRD})(Fig. 2d-e), but even then we compared occupancy of overexpressed proteins with that of endogenous SS18-SSX - Fig. 2d and 2e shows endogenous SS18-SSX and KDM2B in comparison with eGFP-SSX constructs.

For experiments using the new SSX fusions – MN1-SSX1 and EWSR1-SSX1 - we agree that would be better to study endogenous proteins. However, there are no synovial sarcoma cell lines that harbour these translocations for functional studies. We needed to rely on ectopic expression in mesenchymal stem cells to determine if they activate the same expression signature as SS18-SSX (Fig.3a-c). We have included the following sentence in the discussion so that it is clear this is a limitation in our study: *“A limitation of our study is the use of ectopic expression of these new fusions for mechanistic studies, which may not reproduce the physiological levels observed in tumours. Future work will be needed to generate patient derived cell lines or murine models where the molecular activity of these alternative SSX fusions can be studied in more detail.”*

(2) I am surprised about the authors' use of EWSR1-SSX to highlight BAF-independent roles, since it is known to directly interact with BAF. I am not sure I follow the authors' logic about whether this is truly BAF-independent. Along these lines, are the synovial sarcomas with non-SS18-SSX fusions insensitive to degradation of SMARCA4/2 or BRD9?

Yes, the reviewer is correct that it has been shown that EWSR1 interacts with the BAF complex. We cited that study and referred to it in the results section (line 250) “In line with previous studies reporting an interaction of the BAF complexes with EWSR1 and MN1^{47,48}, we observed that all SSX fusions resulted in rewiring of the BAF to Barr bodies (Fig. 3e).”

However, as we also show that:

- EWSR1-SSX1 and MN1-SSX1, but not SS18-SSX, lead to the deposition of H3K27ac (**Fig. 3e, Extended Fig. 3a**).
- Treatments with ACBI1 (a SMARCA2/4 degrader) does not affect: 1) interaction of new fusions with other transcriptional regulators (TBP and EP300) (**Fig. 3i**), 2) Recruitment of both fusions to Barr bodies or consequent H3K27ac deposition (**Fig. 3j, Extended Fig. 3b**) or 3) gene activation by EWSR1-SSX1 or MN1-SSX1 in MSCs (**Extended Fig. 3c**).

Based on these results we think it is clear that aberrant gene activation by alternative SSX fusions can occur without direct BAF complex deregulation. The fact that the publication reporting these fusions (Yoshida et al 2022, PMID: 34504309) shows that the tumours retain SMARCB1 expression further supports the idea that BAF complex composition and activity is not affected in this case. Regarding sensitivity to BAF/BRD9 inhibition. This is an excellent question however, as mentioned in the first point, there are currently no synovial sarcoma lines harbouring these alternative fusions that could be used to address it.

(3) The effects of the SSX C-terminus alone towards chromatin localization appear robust. The mechanisms underlying this observation alone feel to me worthy of a visible publication. However, the increased stabilization of H2AK119ub1 is very modest and, despite one IF image showing a few cells in Figure 4, the effect appears inconsistent across cell contexts, and is not especially well quantified in the current manuscript. This stands in contrast to the strong, binary changes of H2AK119ub1 seen between normal and synovial sarcoma samples (e.g. Figure 5F). Hence stabilization of H2AK119ub1 is not the strongest aspect of the work. My suggestion would be to focus on the chromatin targeting, and perhaps highlight this in the manuscript title, which to me currently feels a little unfocused.

Not all cells exhibit a clear noticeable increase in H2AK119ub1 in the microscopy fields shown in Fig.4d and Fig.4e because the levels achieved by overexpression of eGFP-SSX tagged proteins are heterogeneous to start with. In fact, we took advantage that not all cells express high levels of eGFP-SSX construct to internally correlate their expression with H2A119ub1 levels. Still, the conclusions are made from the analysis and quantification of many nuclei from independent biological replicates (quantification shown in Fig.4f). See also Extended Fig 6 for a detailed explanation on how the quantification was done.

Regarding the “strong binary” changes in Fig 5f. The changes in SS18-SSX fusion IHC (top panels) are indeed binary because normal muscle, like other tissues or tumours do not express the oncofusion which is synovial sarcoma-specific (see also quantification in 5g). However, we do not see this binary pattern the reviewer mentions for H2AK119ub1. As one would expect, normal muscle is also positive for H2AK119ub1 as are other tissues and tumours. Synovial sarcoma exhibits overall higher levels in comparison (please see graph of the left in Fig 4g).

(4) More thought should be given to the clarity of the figure titles. For example, Figure 1 states that PRC1 and not PRC2 deposits H2AK119ub1 independent of PRC2 — it is already well-established that PRC1 deposits this mark, so it is a little unclear to me what truly novel point or contrast that the authors wish to make.

We changed the figure title so that is clearer. The main point of this figure is to show that SS18-SSX1 recruitment relies on an intact PRC1.1 complex but is independent from PRC2 activity. This is an important point because SS18-SSX binds Polycomb target genes, usually occupied by both PRC1 and PRC2 complexes.

(5) In general, the model that the authors propose in Figure 6 is appealing, however the generality of the message about these activity being BAF-independent is undercut by the use of EWSR1-SSX, as stated above. If the authors wish to claim this as a general model, perhaps they could consider using synthetic transcriptional activator fusions (e.g. VP16, VP64, etc.), to demonstrate entirely synthetic recapitulation of the transforming activity independent of the natural fusions observed in disease.

This is a very good suggestion, that we also thought about. We prioritized the fusions that occur in patients for RNAseq analysis and experiments described in the revised manuscript. But we have recently cloned a VP64-SSX1 construct that we plan to use in future experiments.

(6) I also appreciate the message that there are BAF-independent roles of the SSX C-terminus, which I think is really interesting. SSX-C may stabilize H2AK119ub1, or do a lot of things regardless of BAF, and perhaps BAF is just one of many such transcriptional activators that could transform cells when linked to SSX-C. I think these is may be very important for the field to know. Still, I am left wondering whether these observations matter if H2AK119ub1 is insufficient to transform on its own and if SWI/SNF activity is the dominant transforming activity in synovial sarcoma. Greater clarity on this would increase the impact of the story.

We appreciate the comments regarding the importance of BAF-independent roles of SSX-C. Regarding the insufficiency of H2AK119ub1 to transform cells. SSX-C alone binds H2AK119ub1-rich regions and further promotes H2AK119ub1 deposition, but does not activate a synovial sarcoma signature. This is not surprising, since SSX proteins are transcriptional repressors. Still, when SSX-C is fused to a transcriptional activator, the SSX/H2AK119ub1 connection is fundamental for specificity and consequent aberrant activation of Polycomb target genes. So indeed, in synovial sarcoma, increased H2AK119ub1 does not transform the cells, but acts as an interface for SSX fusions to remodel occupancy of transcriptional activators resulting in the specific synovial sarcoma gene signature. In others tumours that express wildtype SSX proteins, it remains to be determined if their role in enhancing H2AK119ub1 has an oncogenic role. We also added a sentence in the discussion so that it is clear that the latter remains undetermined.

Final Decision Letter:**Message** 15th Aug 2023

:

Dear Dr. Banito,

We are now happy to accept your revised paper "Aberrant gene activation in synovial sarcoma relies on SSX specificity and increased PRC1.1 stability" for publication as an Article in Nature Structural & Molecular Biology.

Your paper will be published online soon after we receive proof corrections and will appear in print in the next available issue. You can find out your date of online publication by contacting the production team shortly after sending your proof corrections. Content is published online weekly on Mondays and Thursdays, and the embargo is set at 16:00

London time (GMT)/11:00 am US Eastern time (EST) on the day of publication. Now is the time to inform your Public Relations or Press Office about your paper, as they might be interested in promoting its publication. This will allow them time to prepare an accurate and satisfactory press release. Include your manuscript tracking number (NSMB-A46664B) and our journal name, which they will need when they contact our press office.

About one week before your paper is published online, we shall be distributing a press release to news organizations worldwide, which may very well include details of your work. We are happy for your institution or funding agency to prepare its own press release, but it must mention the embargo date and Nature Structural & Molecular Biology. If you or your Press Office have any enquiries in the meantime, please contact press@nature.com.

Please note that *Nature Structural & Molecular Biology* is a Transformative Journal (TJ). Authors may publish their research with us through the traditional subscription access route or make their paper immediately open access through payment of an article-processing charge (APC). Authors will not be required to make a final decision about access to their article until it has been accepted. <https://www.springernature.com/gp/open-research/transformative-journals> Find out more about Transformative Journals

Authors may need to take specific actions to achieve [compliance](https://www.springernature.com/gp/open-research/funding/policy-compliance-faqs) with funder and institutional open access mandates. If your research is supported by a funder that requires immediate open access (e.g. according to [Plan S principles](https://www.springernature.com/gp/open-research/plan-s-compliance)) then you should select the gold OA route, and we will direct you to the compliant route where possible. For authors selecting the subscription publication route, the journal's standard licensing terms will need to be accepted, including [12](https://www.springernature.com/gp/open-research/policies/journal-

self-archiving policies. Those licensing terms will supersede any other terms that the author or any third party may assert apply to any version of the manuscript.

Sincerely,

Dimitris Typas
Associate Editor
Nature Structural & Molecular Biology
ORCID: 0000-0002-8737-1319
